# Multi-Integration of Labels Across Categories for Component Identification in Multi-trial Time Series

**Noga Mudrik** [1]   **Yuxi Chen** [2]   **Gal Mishne** [3]   **Adam S. Charles** [1]

## Abstract

Many fields collect large-scale temporal data through repeated measurements ('trials'), where each trial is labeled with a set of metadata variables spanning several categories. For example, a trial in a neuroscience study may be linked to a value from category (a): task difficulty, and category (b): animal choice. A critical challenge in time-series analysis is to understand how these labels are encoded within the multi-trial observations, and disentangle the distinct effect of each label entry across categories. Here, we present MILCCI, a novel data-driven method that i) identifies the interpretable components underlying the data, ii) captures cross-trial variability, and iii) integrates label information to understand each category's representation within the data. MILCCI extends a sparse per-trial decomposition that leverages label similarities within each category to enable subtle, label-driven cross-trial adjustments in component compositions and to distinguish the contribution of each category. MILCCI also learns each component's corresponding temporal trace, which evolves over time within each trial and varies flexibly across trials. We demonstrate MILCCI's performance through both synthetic and real-world examples, including voting patterns, online page view trends, and neuronal recordings.

[1]Biomedical Engineering, Kavli NDI, The Mathematical Institute for Data Science, Center for Imaging Science, The Johns Hopkins University, Baltimore, MD [2]Department of Neuroscience, The Johns Hopkins University, Baltimore, MD [3]Halıcıoğlu Data Science Institute, UCSD, San Diego, CA. Correspondence to: Gal Mishne <gmishne@ucsd.edu>, Adam S. Charles <adamsc@jhu.edu>.

*Proceedings of the $43^{rd}$ International Conference on Machine Learning*, Seoul, South Korea. PMLR 306, 2026. Copyright 2026 by the author(s).

## 1. Introduction

A key approach to understanding high-dimensional, temporally evolving systems (e.g., the brain) is analyzing time-series data from multiple repeated observations (hereafter *'trials'*). Each trial is typically labeled with a set of experimental metadata variables. Often, such metadata spans multiple *categories*; for example, each trial in neuronal recordings can be labeled with an attribute from category (a) task difficulty, and from category (b) animal's choice; each trial in weather measurements can be a time series of temperature over a day, labeled with (a) city, (b) humidity level, and (c) precipitation. We therefore refer to a trial's *label* as the tuple of its category values, e.g., '(easy task, correct choice)' or '(New York, 90% humidity, 1" snow)'. Notably, different trials can have similar or distinct labels. When a category changes between trials (e.g., task difficulty: easy vs. difficult), the corresponding *label entry* changes, while *label entries* corresponding to other categories may remain the same or also change.

Given such multi-trial, multi-label (*'multi-way'*) data, interpreting how the observations vary across the label space is complicated by the data's high dimensionality and trial-to-trial variability. A practical approach is to analyze the data in a lower-dimensional latent space (Ma & Zhu, 2013), where label-related structures become easier to interpret. In this space, the activity can be described by a small set of *components*—units that capture the dominant sources of variability across trials and labels. Existing dimensionality reduction methods for analyzing multi-way data often factorize a single, large tensor into such components (Harshman et al., 1970), but they typically overlook trial labels and require constraints on the data structure (e.g., equal-length trials). An alternative is to apply factorizations separately to each trial, which can accommodate varying trial structure but sacrifices information about cross-trial relationships.

Hence, there is a need for new flexible yet interpretable methods to (1) discover the underlying structure within high-dimensional multi-way data, (2) reveal how it captures label information, and (3) disentangle the effect of each category. This, in turn, demands leveraging the trial-to-trial relationships captured by the labels and understanding how these relationships govern the observations.

In this paper, we present MILCCI, a novel method to uncover the underlying structure of multi-way time-series data and disentangle how multi-category labels are embedded within it, both structurally and temporally. Our contributions include:

- We introduce MILCCI, a flexible model that discovers interpretable sparse components underlying multi-way data and reveals how they capture diverse label categories.
- We identify components that capture label-driven variability across trials and track how their activations evolve within individual trials, thereby encompassing the full spectrum of trial-by-trial variability.
- We validate MILCCI on synthetic data, showing it better recovers true components than other methods.
- We demonstrate MILCCI's ability to uncover interpretable, meaningful patterns in real data, including the discovery of voting trends across US states that match known events, patterns of online activity reflecting language and device, and neural ensembles supporting decision-making in multi-regional recordings.

## 2. Related Work

The naive approach for analyzing multi-way data is to apply dimensionality reduction individually per trial or jointly across all trials (by stacking multiple trials into a single matrix). This can be done, e.g., via linear matrix decomposition such as PCA, ICA (Hyvarinen et al., 2001), NMF (Lee & Seung, 1999)), sparse factorization for improved interpretability (e.g., SPCA (Zou et al., 2006)), or via non-linear embeddings (e.g., t-SNE (Maaten & Hinton, 2008)). However, per-trial analysis overlooks cross-trial relationships, while analyzing all trials with a single mapping ignores trial-to-trial variability in internal structure.

Demixed PCA (dPCA) (Kobak et al., 2016) isolates task-related neural variance into low-dimensional components, however it does not address missing data, different trial durations, and varying trial sampling rates, which hinders alignment across heterogeneous trials. Mudrik et al. (2024) recently introduced a unified cross-trial model that identifies building blocks encoding label information in multi-array data; however, their method handles only a single dimension of label change and thus cannot disentangle effects of multiple categories that change jointly or separately across trials. TDR and its extensions (Mante et al., 2013; Aoi & Pillow, 2018) capture multi-category labels via per-trial scalar reweighting of fixed matrices, but assume cross-trial variability arises only from linear reweighting of fixed temporal signals and are not tailored to capture variability across trials sharing the same label.

Tensor Factorization (TF), e.g., PARAFAC (Harshman et al., 1970) and HOSVD (Lathauwer et al., 2000), goes beyond individual trials by treating trials as an extra data dimension

in a multi-dimensional array. However, existing TF methods, including those incorporating Gaussian processes (Tillinghast et al., 2020; Xu et al., 2012; Zhe et al., 2016) or dynamic information (Wang & Zhe, 2022), are not designed to distinguish label-driven variability from other sources of variability and often produce components that are difficult to interpret. SliceTCA (Pellegrino et al., 2024) extends tensor factorization by simultaneously demixing neural, trial, and temporal covariability classes within the same dataset, allowing components to capture structure across different types of neural variability. However, sliceTCA does not incorporate explicit label information and forces a tensor structure on the data. Chen et al. (2015) enable flexibility in cross-trial representations based on meta-data information, but their assumption of component orthogonality prevents the model from capturing correlated or partially overlapping patterns, limiting their expressive power.

Unlike the methods above, MILCCI (1) identifies the structure of multi-trial, multi-label data that vary over sessions, (2) captures trial-to-trial variability, including within repeated measures of the same label, (3) disentangles how each category is encoded in the data.

*Table 1. Key notations.*

| Notation | Description |
|---|---|
| $m$ | Trial # |
| $(k) \in C$ | Category |
| $L^{(m)} = (L_{(a)}^{(m)}, L_{(b)}^{(m)}, \ldots)$ | Label of trial $m$ |
| $\boldsymbol{Y}^{(m)} \in \mathbb{R}^{N \times T^{(m)}}$ | Trial $m$'s observations |
| $\boldsymbol{\Phi}^{(m)} \in \mathbb{R}^{P \times T^{(m)}}$ | Trial $m$'s traces |
| $\mathcal{G}^{(k)}$ | Indices of category (k)'s traces |
| $\mathcal{A}^{(k)} \in \mathbb{R}^{N \times p^{(k)} \times |(k)|}$ | Category (k)'s components |
| $\boldsymbol{A}^{(L^{(m)})} \in \mathbb{R}^{N \times P}$ | Trial $m$'s components |

## 3. Our Component-identification Approach

**Problem formulation:**

Let $\boldsymbol{Y} = \{\boldsymbol{Y}^{(m)}\}_{m=1}^{M}$ be a set of $M$ time-series (trials), where each $\boldsymbol{Y}^{(m)} \in \mathbb{R}^{N \times T^{(m)}}$ represents measurements from $N$ channels across $T^{(m)}$ time points (Fig. 1, **I**). We assume that the identities of the $N$ channels are fixed across trials, while trial durations, $T^{(m)}$, can vary. Each trial $\boldsymbol{Y}^{(m)}$ is observed under trial-related metadata variables belonging to categories $C := \{a, b, \ldots, f\}$ (e.g., (a) task difficulty, (b) animal's choice, etc.), such that $|C|$ is the number of categories. We define the *label* of trial $m$ as the tuple $L^{(m)}$ containing its $|C|$ metadata variables, such that each *label entry* $L_i^{(m)}$ is the value of the $i$-th category in that trial (e.g., $L^{(m)} = (\text{task difficulty: easy}, \text{choice: correct})$). Notably, different trials may exhibit identical labels, partially overlapping labels, or entirely distinct labels. We aim to understand how these labels are encoded within the multi-way observations via their underlying components and traces.

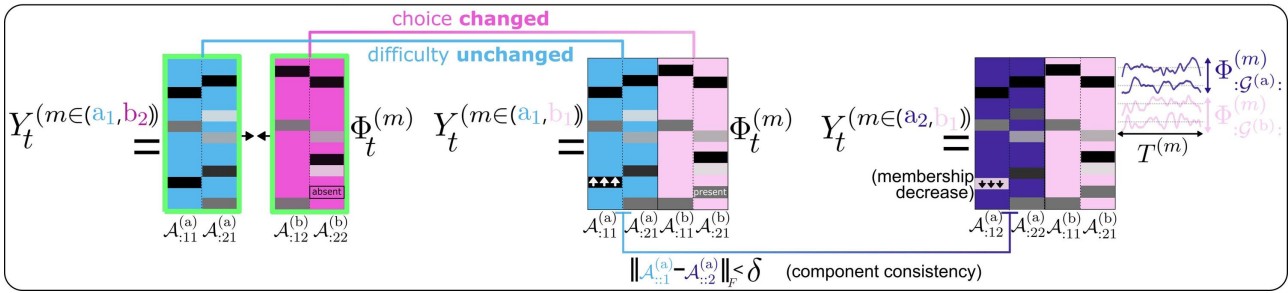

*Figure 1.* **Illustration. I:** Time-series (e.g., brain recordings) across $M$ trials of varying duration ($\{T^{(m)}\}_{m=1}^M$). Each trial $m$ is associated with a label $L^{(m)}$, which is a set of experimental variables spanning different categories (e.g., $L^{(m)} = (\text{easy task}, \text{correct choice})$). **II:** Each category (k)'s components are represented by a tensor $\mathcal{A}^{(k)}$, whose $i$-th variant ($\mathcal{A}^{(k)}_{::i}$) refers to the $i$-th option of that category (e.g., if the 2-nd option of category (b): correct choice, then $\mathcal{A}^{(b)}_{::2}$ are correct-choice components). **III:** Each trial $m$ is modeled via a sparse factorization, with its sparse components defined by selecting a variant (layer) from each category's tensor, based on that trial's label (green borders, **II**), and then concatenating all selected variants horizontally (green borders, **III**). This forms the loading matrix of that trial. Importantly: 1) trials with identical labels use identical loadings, 2) components can *subtly* adjust their composition under shifts in the respective category values to maintain consistency (e.g., same component under task difficulty 1 vs. 2: $\|\mathcal{A}^{(a)}_{::1} - \mathcal{A}^{(a)}_{::2}\|_F < \epsilon$), and 3) component temporal traces ($\{\Phi^{(m)}\}_{m=1}^M$) can vary flexibly across trials.

A parsimonious modeling strategy is to model observations $\boldsymbol{Y}^{(m)}$ in each trial $m$ as being linearly generated by a small set of $P$ core components $\boldsymbol{A}^{(m)} \in \mathbb{R}^{N \times P}$ with corresponding temporal activity $\boldsymbol{\Phi}^{(m)} \in \mathbb{R}^{P \times T^{(m)}}$, such that $\boldsymbol{Y}^{(m)} = \boldsymbol{A}^{(m)}\boldsymbol{\Phi}^{(m)} + \epsilon$, with $\epsilon$ representing e.g., *i.i.d.* Gaussian noise. Particularly, traditional approaches model the data either with per-trial $\boldsymbol{A}^{(m)}$ reflecting components that change freely over trials, or via a single $\boldsymbol{A}$ shared between trials (via matrix/tensor factorization on stacked trials) such that components are identical across all trials. Consequently, they cannot capture components that *subtly* adjust their composition under label changes across trials.

For example, consider a brain network (i.e., a neuronal 'component') that recruits additional neurons during a hard task but not during an easy task. Methods that linearly scale fixed components, if applied independently to trials across task difficulties, would fail to recognize the network as identical across these conditions. Alternatively, if applied to all trials stacked, they would force identical component structures for both easy and difficult tasks, which would misrepresent the additional recruited neurons and would also distort the corresponding traces tied to the components.

Hence, there is a need for methods capable of identifying

consistent components underlying multi-way data, understanding how they adapt based on label changes, and capturing their temporal-trace evolution within and across trials.

### 3.1. Our Model:
**Component Decomposition Approach:** We first assume that to capture category-specific information, each component in $\boldsymbol{A}$ corresponds to a single category (k). Thus, $\boldsymbol{A}$ is composed as a set of $|C|$ category-specific component matrices, $\{\boldsymbol{A}^{(k)}\}_{(k) \in C}$, where each $\boldsymbol{A}^{(k)} \in \mathbb{R}^{N \times p^{(k)}}$ consists of $p^{(k)}$ components associated with category $(k) \in C$, such that $\sum_{(k) \in C} p^{(k)} = P$. Each entry $A^{(k)}_{nj}$ captures channel $n$'s membership in the $j$-th component of category (k) (e.g., the extent to which neuron $n$ participates in that neuronal ensemble), and $A^{(k)}_{nj} = 0$ indicates non-membership (Fig. 1, **II**). We assume that component memberships are sparse, namely each channel $n$ belongs to *only a few* components. Hence, we place a Laplace prior on each entry: $A^{(k)}_{nj} \sim \text{Laplace}(0, \frac{1}{\gamma_1})$, where $\gamma_1$ is a sparsity-scaling parameter (App. C). Each of the $P$ components exhibits a time-varying trace within each trial (m), collectively represented by the rows of the per-trial traces matrix $\boldsymbol{\Phi}^{(m)} \in \mathbb{R}^{P \times T}$. Then, the observations in each trial are

modeled by $\boldsymbol{Y}^{(m)} \approx \sum_{(\mathrm{k}) \in C} \boldsymbol{A}^{(\mathrm{k})} \boldsymbol{\Phi}_{\mathcal{G}^{(\mathrm{k})}:}^{(m)}$, where $\mathcal{G}^{(\mathrm{k})}$ are the row indices of $\boldsymbol{A}^{(\mathrm{k})}$'s traces (length($\mathcal{G}^{(\mathrm{k})}$) = $p^{(\mathrm{k})}$).

**Extension to Label-dependent Components:** Our full model extends beyond a fixed-component structure; instead, we assume that not only are the components category-specific, the components of each category can exhibit compositional variants by subtly adjusting their membership under label changes. Thus, the representation of each category (k)'s components extends beyond a single matrix to a 3D tensor $\mathcal{A}^{(\mathrm{k})} \in \mathbb{R}^{N \times p^{(\mathrm{k})} \times |(\mathrm{k})|}$, where $|(\mathrm{k})|$ is the number of unique label options for that category (e.g., $|(\mathrm{k})| = 2$ for a binary choice vs. $|(\mathrm{k})| = \kappa$ options for task difficulty, Fig. 1, **II**). The $i$-th component variant of category (k) is the $i$-th slice along the third mode of the component tensor $\mathcal{A}^{(\mathrm{k})}$ and is denoted by $\mathcal{A}_{::i}^{(\mathrm{k})}$ (Fig. 1, **II**).

Thereby, for any trial $m$ with label $L^{(m)}$, category (k) contributes the component variant $\mathcal{A}_{::\arg(L_{(\mathrm{k})}^{(m)})}^{(\mathrm{k})}$, where $\arg(L_{(\mathrm{k})}^{(m)}) \in \mathbb{Z}_{\geq 0}$ maps a label value to its corresponding variant index within that category (e.g., $\arg(\text{choice: correct}) = 2$, as 'correct' is the 2nd option in the choice category).

While the variants of each component need not be identical, we assume they are structurally similar, with their similarity level proportional to the similarity of their label values. Thus, for distinct category options $(\mathrm{k})_{i'} \neq (\mathrm{k})_i$ within category (k), we assume that $\|\mathcal{A}_{::i'}^{(\mathrm{k})} - \mathcal{A}_{::i}^{(\mathrm{k})}\|_F^2 < \delta\left((\mathrm{k})_i, (\mathrm{k})_{i'}\right)$, for some small category-dependent distance $\delta$. Notably, this constraint promotes alignment of the components across variants, forcing the $\{\mathcal{A}^{(\mathrm{k})}\}$ tensors to capture meaningful synergies between different label combinations rather than overfit to individual trials, ultimately revealing shared yet adaptable patterns across labels.

The decomposition model is thereby extended to the following label-specific formulation for each trial $m$:

$$\boldsymbol{Y}^{(m)} = \sum_{(\mathrm{k}) \in C} \mathcal{A}_{::\arg(L_{(\mathrm{k})}^{(m)})}^{(\mathrm{k})} \boldsymbol{\Phi}_{\mathcal{G}^{(\mathrm{k})}:}^{(m)} + \boldsymbol{\epsilon}, \ \epsilon_{ij} \overset{\text{i.i.d.}}{\sim} \mathcal{N}(0, \sigma^2). \quad (1)$$

### 3.2. Model Fitting Procedure:
Learning the component compositions and their traces directly from $\{\boldsymbol{Y}^{(m)}\}$ is hindered by the mixing of all labels' effects across categories. We address this through a three-stage procedure:

- *Stage 1: Pre-computing label similarity graphs.* To accommodate both categorical and ordinal labels, MILCCI pre-computes a label similarity graph $\boldsymbol{\lambda}^{(\mathrm{k})} \in \mathbb{R}^{|(\mathrm{k})| \times |(\mathrm{k})|}$ for each category (k) (for details see App. A.2). These graphs can be integrated into *Stage 3* to ensure that compositional adjustments reflect label-to-label distances for ordinal labels.
- *Stage 2: Initialization.* We initialize the components and

traces following Appendix A.1.
- *Stage 3: Iterative optimization.* We update $\{\mathcal{A}^{(\mathrm{k})}\}_{(\mathrm{k}) \in C}$, $\{\boldsymbol{\Phi}^{(m)}\}_{m=1}^M$ until convergence as detailed below.

**Inferring** $\{\mathcal{A}^{(\mathbf{k})}\}_{(\mathbf{k}) \in \mathbf{C}}$: For each unique category (k)'s option, $(\mathrm{k})_i$ (e.g., choice: correct), we infer the $i$-th variant $\mathcal{A}_{::i}^{(\mathrm{k})}$ using all trials observed under $(\mathrm{k})_i$ $(\widetilde{M} = \{m \in \{1, \ldots, M\} \mid L_{(\mathrm{k})}^{(m)} = (\mathrm{k})_i\})$, regardless of those trials' values in other categories (e.g., all trials where choice: correct, regardless of trial difficulty).

Since each trial $m$ is modeled via a decomposition of components from multiple categories (Eq. 1), to learn this variant's ($\mathcal{A}_{::i}^{(\mathrm{k})}$) structure, we need to separate its contribution from the contributions of the other categories' components. Hence, for each $m \in \widetilde{M}$, we first calculate the residual matrix $\widetilde{\boldsymbol{Y}}^{(m,\mathrm{k})}$, which represents the difference between the observations $\boldsymbol{Y}^{(m)}$ and the partial reconstruction based on all components excluding $\mathcal{A}_{::i}^{(\mathrm{k})}$. Specifically: $\widetilde{\boldsymbol{Y}}^{(m,\mathrm{k})} := \boldsymbol{Y}^{(m)} - \sum_{(\mathrm{k}') \neq (\mathrm{k})} \mathcal{A}_{::\arg(L_{(\mathrm{k}')}^{(m)})}^{(\mathrm{k}')} \boldsymbol{\Phi}_{\mathcal{G}^{(\mathrm{k}')}:}^{(m)}$. We then infer $\mathcal{A}_{::i}^{(\mathrm{k})}$ via LASSO (Tibshirani, 1996):

$$\widehat{\mathcal{A}}_{::i}^{(\mathrm{k})} = \arg \min_{\mathcal{A}_{::i}^{(\mathrm{k})}} \|\widetilde{\boldsymbol{Y}}^{(m,\mathrm{k})} - \mathcal{A}_{::i}^{(\mathrm{k})} \boldsymbol{\Phi}_{\mathcal{G}^{(\mathrm{k})}:}^{(m)}\|_F^2$$
$$+ \gamma_1 \|\mathcal{A}_{::i}^{(\mathrm{k})}\|_{1,1} + \gamma_2 \sum_{i' \neq i} \lambda_{i',i}^{(\mathrm{k})} \|\mathcal{A}_{::i'}^{(\mathrm{k})} - \mathcal{A}_{::i}^{(\mathrm{k})}\|_F^2 \quad (2)$$

where $\lambda_{i',i}^{(\mathrm{k})}$ promotes similarity among same-category (k) variants (App. A.2), and $\gamma_1, \gamma_2$ are hyperparameters controlling sparsity and variant-to-variant similarity. Notably, the model also supports applying a non-negativity constraint on the components ($\mathcal{A}_{nji}^{(\mathrm{k})} \geq 0 \, \forall n, j, i$), useful in applications where positive values are expected.

Collectively, Equation 2 balances data fidelity (1st term), component sparsity (2nd term), and consistency between corresponding components proportional to their label similarity by $\boldsymbol{\lambda}^{(\mathrm{k})}$ (3rd term). Each component is then normalized to a fixed sum to avoid scaling ambiguity with $\boldsymbol{\Phi}^{(m)}$.

**Updating** $\{\boldsymbol{\Phi}^{(\mathbf{m})}\}_{m=1}^M$: Since traces vary independently of labels across trials (i.e., are unsupervised), they can be learned per trial $m$ using the label-driven loading matrix of realized components in that trial. This loading matrix, notated by $\boldsymbol{A}^{(L^{(m)})} \in \mathbb{R}^{N \times P}$, is an auxiliary variable constructed by selecting one slice from each category tensor $\{\mathcal{A}^{(\mathrm{k})}\}_{(\mathrm{k}) \in C}$ according to the trial's label $L^{(m)}$, and horizontally concatenating these slices (Fig. 1, **III**). In each trial $m = 1 \ldots M$, we can then update $\boldsymbol{\Phi}^{(m)}$ by:

$$\widehat{\boldsymbol{\Phi}}^{(m)} = \arg \min_{\boldsymbol{\Phi}^{(m)}} \underbrace{\|\boldsymbol{Y}^{(\mathrm{m})} - \boldsymbol{A}^{(L^{(m)})} \boldsymbol{\Phi}^{(m)}\|_F^2}_{\text{data fidelity}} \quad (3)$$

$$+ \gamma_3 \underbrace{\sum_{t=1}^{T^{(m)}} \|\boldsymbol{\Phi}_{:,t}^{(m)} - \boldsymbol{\Phi}_{:,t-1}^{(m)}\|_2^2}_{\text{temporal smoothness}} + \gamma_4 \underbrace{\|(\boldsymbol{C} \odot (\boldsymbol{1} - \boldsymbol{I}_P)) \odot \boldsymbol{D}\|_{1,1}}_{\text{within-trial trace decorrelation}},$$

where $\gamma_3, \gamma_4$ are hyperparameters, $\odot$ is element-wise multiplication, $\mathbf{C} := \mathrm{Gram}(\boldsymbol{\Phi}^{(m)})$, and $\boldsymbol{D} \in \mathbb{R}^{p \times p}$ is for normalization $(D_{j,j'} := \|\boldsymbol{\Phi}^{(m)}_{:j}\|_2^{-1} \|\boldsymbol{\Phi}^{(m)}_{:j'}\|_2^{-1})$. Eq. 3 overall promotes data fidelity (1st term), encourages smoothness (2nd term), and penalizes correlations between traces of different components (3rd term). See algorithm, notations, and illustration in Alg.1, Tab.1, and Fig. 1.

## 4. Experiments

We validate MILCCI on synthetic data and also demonstrate its effectiveness on four real-world datasets.

**MILCCI Recovers True Components from Synthetic Data:** We generated synthetic data arising from $P = 4$ sparse components with time-varying traces ($T = 500$ time points; $M = 250$ trials). We defined two categories: (a) 'task difficulty' (5 options), and (b) 'choice' (2 options), such that $p^{(a)} = p^{(b)} = 2$ ensembles adjust with changes in (a) and (b) (Fig. 6B). Each trace was generated as a Gaussian process with parameters varying across components and trials: some reflect task difficulty, some choice, and some vary each trial (Fig. 2, B, Fig. 6, A, App. D). We ran MILCCI on this data, comparing to (1) Tucker (Tucker, 1966), (2) PARAFAC (Harshman et al., 1970), (3) non-negative PARAFAC (Shashua & Hazan, 2005), (4) SVD, (5) SiBBlInGS (Mudrik et al., 2024), and (6) sliceTCA (Pellegrino et al., 2024); all using the same $P = 4$ components (Fig. 2, F,G, Sec.H). Since TF is invariant to component permutation, we used linear sum assignment to align the components of each method with the ground truth.

MILCCI recovered the true components (Fig. 2, C) and traces (Fig. 2, D), with high correlations to the ground truth for both (Fig. 2, E). Compared to other methods (Fig. 2, F,G), MILCCI achieved the highest similarity to the ground truth. While SiBBlInGS attains comparable correlation, it produces blurred traces (e.g., Fig. 2, F) and lacks MILCCI's interpretability in distinguishing the contribution of each category.

**MILCCI Reveals State-Level Voting Patterns by Party and Office:** We next tested MILCCI on voting data from (Data & Lab, 2017a;b;c) consisting of voting counts for $N = 51$ US states (including DC) across $T = 23$ years (sampled every 2 or 4 years) for different parties (Democrat, Republican, Libertarian; Tab. 3) and offices (Presidency, Senate, House), such that each $\boldsymbol{Y}^{(m)} \in \mathbb{R}^{51 \times 23}$ represents the voting counts of all states over years. We first preprocessed the data, including handling missing values due to differing election schedules across states (Fig. 9, App. E).

We applied MILCCI using $P = 8$ components, with categories (a) party, and (b) office ($p^{(a)} = p^{(b)} = 4$ each). MILCCI discovers components capturing state-specific voting patterns that vary by office, party, and time. For example,

component $\mathcal{A}^{(\mathrm{party})}_{:1:}$ (Fig. 3, B.1) highlights Montana (MT) and Pennsylvania (PA) having increased membership in the 'Other' category, primarily driven by the Independent Party (Fig. 17, right) and the Constitution Party (Fig. 17, left), respectively. This aligns with MT's historical emphasis on individualism (Kitayama et al., 2010) and PA hosting the Constitution Party headquarters. The same component identifies Oregon's increased Libertarian membership, matching the 2001 law that eased ballot access for minor parties (*Oregon Political Party Manual*) and reflecting that the Libertarian Party of Oregon was among the earliest state branches. Notably, its trace ($\boldsymbol{\Phi}_{:\mathcal{G}_1^{(\mathrm{party})}}$, Fig. 3, B.1, top-left) shows overlapping Democrat–Republican activations diverging $\sim$2004, with Democrat activity rising and Republican activity decreasing, reflecting long-term partisan realignment driven by national political shifts of that period (e.g., Iraq War, 2003).

In component $\mathcal{A}^{(\mathrm{party})}_{:3:}$ (Fig. 3, B.1, right), MILCCI groups AK, OK, AL, AZ, MS, MT together. This grouping matches the legislative similarities between these states, e.g., strict voter ID laws (National Conference of State Legislatures, 2025), demonstrating MILCCI's effectiveness in recovering underlying trends directly from observations. Temporally, its trace $\boldsymbol{\Phi}_{:\mathcal{G}_3^{(\mathrm{party})}}$ (Fig. 3, B, bottom-left) shows opposing trends between Democrat and Republican activations, with Republican activation rising, and Libertarian activity emerging around 2016. This trend reflects a rise in Republican votes, a decline in Democratic votes, and a possible shift of some Democratic support toward the Libertarian party in these states. These and other patterns identified by MILCCI (App. E.2) demonstrate its ability to uncover state–party–office–dependent patterns. We further validated (App. E.3) that MILCCI's discovered voting components capture genuine structures, as individual states show meaningful contributions to reconstruction (Fig. 10, A-C), and the model significantly outperforms permuted null models ($p < 0.001$). Additional voting insights are in App. E.2.

Notably, components from other TF methods (Fig. 11, 14) are dense (PARAFAC), include negative values (SVD, PARAFAC, Tucker), or, in SiBBlInGS, fail to disentangle party from office, which hinder interpretability (Fig. 13). Moreover, due to their restrictive tensor structure, PARAFAC and Tucker do not capture compositional adjustments and cannot flexibly vary their traces to capture trial-to-trial temporal variability.

**MILCCI Finds Wikipedia Page Clusters Across Devices and Languages:** Next, we extracted Wikipedia pages (Wikipages) Pageview counts (Meta, 2022) (Oct. 20'–24', $T = 1482$ days) for $N = 32$ random Wikipages from different fields. We tracked views under 3 conditions: (a) agent: user/spider (i.e., web crawler), (b) platform: desktop/web/app, and (c) language: en/ar/es/fr/he/hi/zh. Each $\boldsymbol{Y}^{(m)} \in \mathbb{R}^{N \times T}$ contains daily Pageview counts for all 32

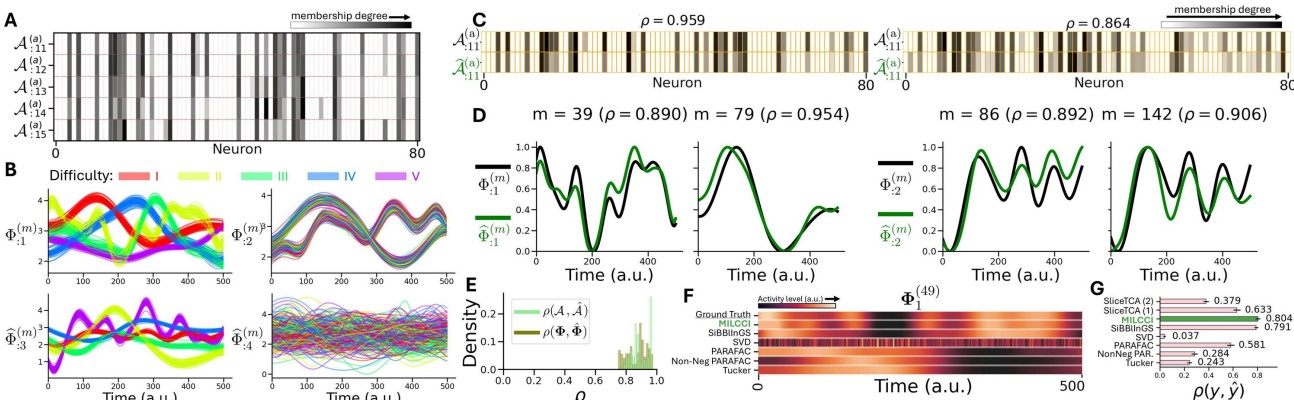

*Figure 2.* **MILCCI Recovers True Representations in Synthetic Data. A-B**: Generated synthetic data (full data in Fig. 6). Ground-truth components (examples in panel **A**) vary slightly across labels but remain fixed across same-label trials (rows). Ground-truth traces vary across trials (**B**, colored by difficulty). **C-D**: Identified vs. ground-truth components (**C**) and time-traces (**D**) for random trials. **E**: Histogram of correlations between identified components and traces vs. their true counterparts. **F-G**: Comparison of MILCCI to other methods (limited to the same 4-component dimension) based on traces (random trial, **F**) and reconstruction performance (**G**, baselines details in Sec. H).

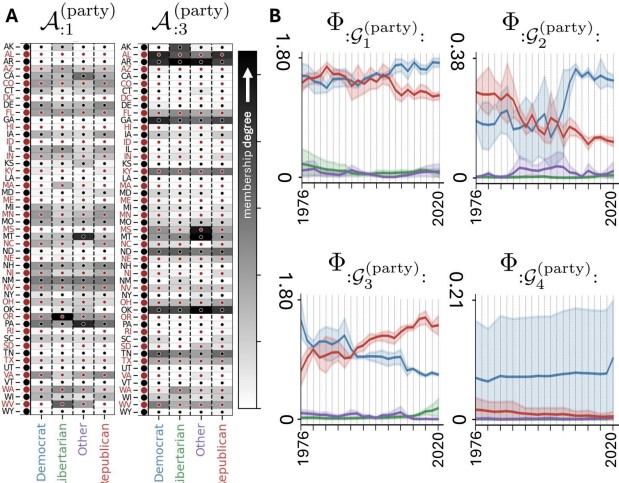

*Figure 3.* **Voting Results:** Identified example components (**A**) and traces (**B**, Mean and 80% confidence interval). See full in Fig. 9.

Wikipages over time under e.g. (English, desktop, user). We thus seek to identify components capturing Wikipages with similar interest patterns and to understand how these patterns vary across these different conditions (data and pre-processing details in App. F.1). We ran MILCCI on the data with $p^{(k)} = 4$ components per category, and identified components that cluster related Wikipages together and vary across categories (Fig. 4, A), with some pages (e.g., unsupervised learning) appearing in more than one component, emphasizing MILCCI's ability to capture multi-meaning terms.

Particularly, we identified, e.g., $\mathcal{A}_{:1:}^{(agent)}$ grouping Learning Theory (in Psychology) related Wikipages (Fig. 4, A, red arrows, list in App. F.2); $\mathcal{A}_{:2:}^{(agent)}$ grouping social-media pages (purple arrows); and component $\mathcal{A}_{:4:}^{(platform)}$ grouping computer science basics (blue arrows). Interesting patterns

emerge when exploring how component compositions adjust to e.g., user↔spider and desktop↔web↔app.

For $\mathcal{A}_{:1:}^{(agent)}$ (Learning theory in Psychology), MILCCI finds small differences across agents (spider vs. user, Fig. 4, A, left). For instance, the Bobo Doll Experiment's Wikipage (a psychological experiment on social learning theory) appears under 'spider' but not 'user'. This is consistent with it being less familiar to the average person than other psychology terms in the cluster, while spiders are linked to it through the actual Wikipedia links connecting this page to other related terms. Accordingly, other Wikipages, like classical and operant conditioning, which are foundations in psychology, show higher membership magnitudes in 'user'. 'Unsupervised learning' and 'embedding' also show small membership in this component, higher in 'user' than 'spider'. Interestingly, these actual Wikipages refer to CS terms (not psychology), but since they also carry meaning in psychological learning, the higher 'user' membership compared to 'spider' matches human behavior: users enter the page but leave upon realizing the term refers to a different field, whereas spiders follow predictable navigation patterns. These findings highlight MILCCI's ability to reveal distinct human vs. spider behaviors within the same component, and also underscore the importance of allowing compositional adjustments to capture subtle component adaptations. This component's trace ($\mathbf{\Phi}_{:\mathcal{G}_1^{(agent)}:}$, Fig. 4, B top-left) captures its fluctuations and higher activation in English compared to other languages. This aligns with professional terms being more elaborate in English, often using jargon not fully defined/used in other languages, and with non-English native speakers possibly preferring to read professional material in English (Miquel-Ribé & Laniado, 2018).

The social-media component $\mathcal{A}_{:2:}^{(agent)}$ shows small structural

**A**

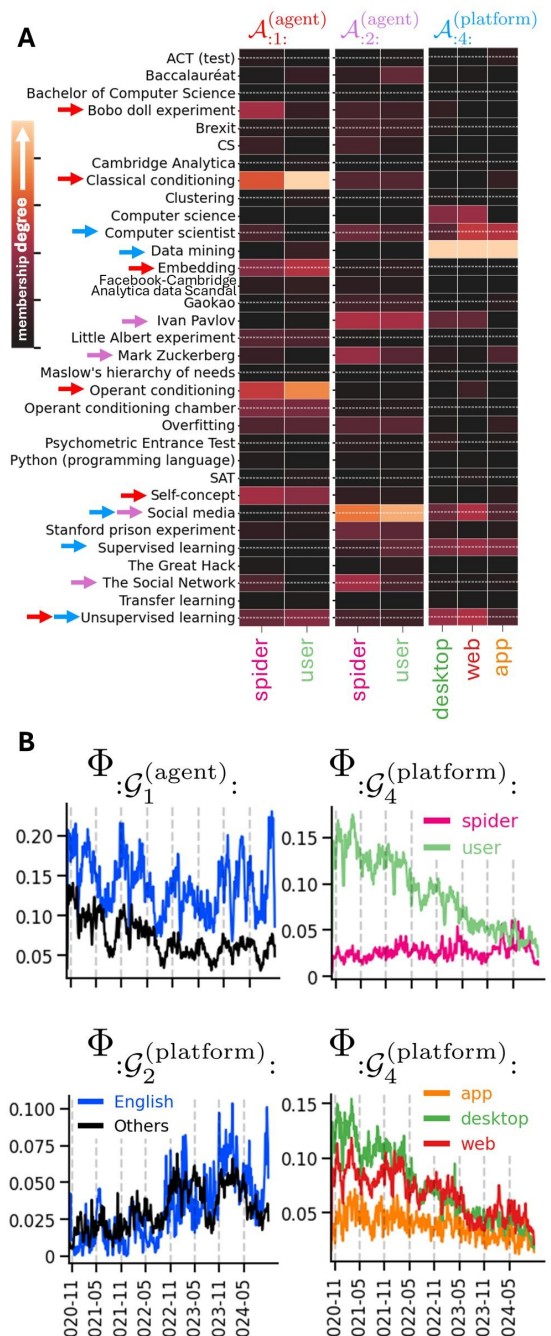

**B**

$$\Phi_{:\mathcal{G}_1^{(\text{agent})}:}$$

$$\Phi_{:\mathcal{G}_4^{(\text{platform})}:}$$

$$\Phi_{:\mathcal{G}_2^{(\text{platform})}:}$$

$$\Phi_{:\mathcal{G}_4^{(\text{platform})}:}$$

*Figure 4.* Identified Wiki page-view example components (**A**) and traces averaged by different categories (**B**).

adjustments user↔spider: 'Mark Zuckerberg' is higher in 'spider', while 'social media' is higher in 'user', possibly reflecting reduced human interest in figures versus common terms like 'social media'. Its trace ($\Phi_{:\mathcal{G}_2^{(\text{agent})}:}$) rises until a peak in Mar. 2024 in both English and non-English (Fig. 4 B, bottom-left), matching the general increase in social media popularity over the years and possibly related to, e.g., Florida's Social Media Ban in Mar. 2024, which is

mentioned on the corresponding 'social media' Wikipage captured by this component.

$\mathcal{A}_{:4:}^{(\text{platform})}$ (computer science basic terms) captures membership adjustments to platform, with, e.g., computer scientist (a short Wikipage without math/graphs) showing higher membership under app/web compared to under desktop. This contrasts with, e.g., unsupervised learning, whose page is more complex (includes math and graphs) and shows lower membership in the app compared to the other platforms. This may match users' preference to read 'easy' terms on the app and more complex terms on desktop, and shows MILCCI's ability to reveal interpretable processing differences across platforms. This component's temporal trace ($\Phi_{:\mathcal{G}_4^{(\text{platform})}}$) shows higher activity in English (Fig. 20), with access dominated by desktop and mobile web (Fig. 4, B, bottom-right). The temporal trace is mostly driven by users rather than spiders (Fig. 4, B, top-right); the user trace decreases over time, aligning with these terms being basic ('old') in CS (compared to newer trends, like LLMs). This highlights MILCCI's power in discovering platform-specific engagements and behavioral fingerprints. Some more interesting patterns include, e.g., $\mathcal{A}_{:2:}^{(\text{platform})}$ that captures terms related to Cambridge Analytica; its trace peaks around 2020, mainly in English, and has decreased since, aligning with the timeline of this case. See Figs. 20, 23, 22, 24, 25 for comparisons and full traces. We also performed cross-validated decoding from temporal traces across all methods, where MILCCI achieves the highest decoding accuracy for both agent and platform categories (Fig. 32).

**MILCCI Finds Neural Ensembles underlying Multi-Region Brain Data:** We then apply MILCCI to neuronal activity patterns from multi-regional, single-cell-resolution recordings of mice in a decision-making task (Laboratory et al. 2025; Angelaki et al. 2025; App. G). In this experiment, mice reported the location of a visual grating with varying contrast by turning a wheel left or right (Fig. 5, A). We used data from a random session, extracted the available spike time data, and estimated firing rates by applying a Gaussian convolution. We removed inactive neurons and split into $M = 1,011$ trials, such that each $\mathbf{Y}^{(m)} \in \mathbb{R}^{N \times T}$ with $N = 270$ and $T = 137$ represents the firing rates of neurons over time for trial $m$ (e.g., Fig. 35). We defined $p^{(\text{k})} = 2$ components ('neuronal ensembles') per category: (a) trial number, (b) prior (expected) stimulus side, (c) applied stimulus side, and (d) fixed components across trials. This setup (1) enables distinguishing representational drift (Rule et al., 2019; Driscoll et al., 2022) potentially related to learning or attention, from task variables, and (2) demonstrates MILCCI's ability to simultaneously support both non-adjusting and adjusting components, across both categorical and ordinal categories.

We found neuronal ensembles selective for diverse task

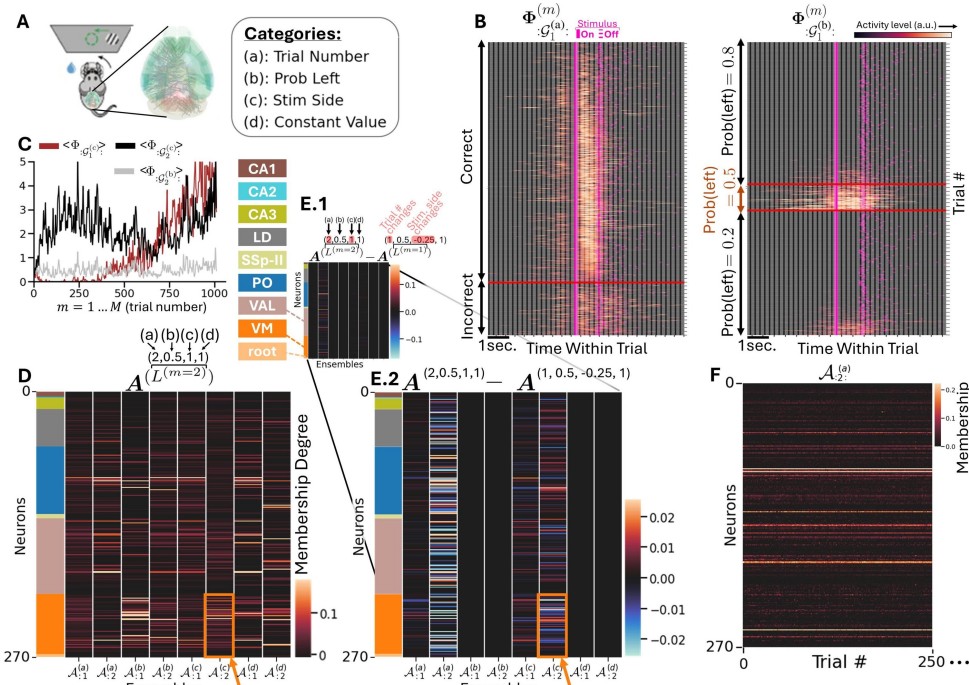

Figure 5. **MILCCI identifies meaningful neuronal ensembles in real-world brain data. A**: Experimental setting (from (Angelaki et al., 2025)). **B**: Component traces (all 1011 trials) sorted horizontally by trial correctness (left panel) and Prob(left) (right panel). **C**: Average within-trial values of exemplary traces reveal varying degrees of temporal drifts over trials. **D**: Ensembles identified (example trial). **E**: Differences in ensemble composition across trials. **F**: Trial-adjusted ensemble compositions over first 250 trials.

variables (Fig. 5). For example, $\Phi^{(m)}_{:\mathcal{G}^{(a)}_1}$ is tuned to choice correctness, with activity surging during stimulus presentation (Fig. 5, B left) under *correct* choice only. Interestingly, MILCCI also found an ensemble sensitive to *incorrect* decisions, activated just after stimulus presentation ($\Phi_{:\mathcal{G}^{(b)}_2:}$, Fig. 36). Notably, these components can provide insight as to how neuronal ensembles integrate stimulus information to support correct decision-making, and to relate these traces to the specific neurons involved (Fig. 38). In another example, $\Phi^{(m)}_{:\mathcal{G}^{(b)}_1:}$ is mostly active during trials with a random-prior (i.e., $p(\text{left}) = 0.5$), throughout before, during, and after the stimulus, and is largely inactive in trials where the prior favors one side (Fig. 5, B right). This highlights MILCCI's effectiveness in suggesting involvement of priors in decision-making, and its capability of exposing similarities of similar-condition traces.

Moreover, MILCCI's ability to isolate components with characteristics that track temporal drifts ($\Phi^{(m)}_{:\mathcal{G}^{(c)}_1:}$, Fig. 5, C brown; 37) reveals how neuronal coding evolves over trials, which can be the result of learning, attention, or adaptation. MILCCI reveals how ensemble compositions capture distinct local and widespread structures across regions (Fig. 5, D; colors on the left mark neurons' brain regions). For example, $\mathcal{A}^{(c)}_{:2:}$ captures many neurons in the Ventral Medial nucleus of the thalamus (VM, orange arrow in Fig. 5D), suggesting it may be involved in arousal regulation and motor coordination. We can also identify interesting patterns via the ensemble compositions (Fig. 5, D), and how they change between trials (e.g., between two example trials with

differing stimulus sides; Fig. 5, E.1, zoom-in on changes in Fig. 5, E.2). These reveal patterns in ensemble composition adjustments: the first ensemble of $\mathcal{A}^{(a)}$ (Fig. 5, E.2, left column) adjusts minimally, while the second ensemble exhibits distributed adjustments across areas. In stimulus-adjusting ensembles (columns 5, 6), adjustments occur around VM (e.g., $\mathcal{A}^{(c)}_{:2:}$, Fig. 5), suggesting an adaptive VM composition. We also performed cross-validated decoding and reconstruction analysis across all methods, where MILCCI achieves the highest decoding accuracy and competitive reconstruction quality (Fig. 40).

**MILCCI Identifies Arousal- and Frequency-Dependent Neuronal Ensembles:** Finally, in App. J, we show that MILCCI identifies components capturing arousal and stimulus frequency, with robustness to hyperparameters.

## 5. Discussion, Limitations, and Future Steps

We presented MILCCI, a data-driven method for analyzing multi-trial, multi-label time series. MILCCI identifies interpretable components underlying the data, captures cross-trial relationships, and integrates label information, while accounting for trial-to-trial variability beyond label effects. MILCCI allows components to adjust their composition with label changes, which uncovers similarities that could remain hidden under fixed-component factorizations. Unlike other approaches, MILCCI maintains interpretability while modeling cross-trial variability, integrating labels, and disentangling label-driven from non-label-driven sources of variation. Another strength of MILCCI is its ability to simul-

taneously handle multiple label types within a single dataset, including categorical, non-continuous ordinal, continuous, and trial-varying labels (e.g., as in the IBL experiment). We validated MILCCI on synthetic data, where it outperformed alternatives, and demonstrated its effectiveness on real-world data: (1) exposing voting patterns aligned with real events; (2) recovering interpretable Wikipedia view components with memberships varying across languages, platforms, and agents; and (3) revealing neuronal ensembles adapting across trials and task variables. Notably, MILCCI's architecture (low-rank, sparse components) naturally constrains the model to learn patterns that are flexibly reused across trials. Another feature of MILCCI is its modular design, which supports substitution of our current MSE metric (reflecting normal distribution assumptions) with alternative application-specific cost metrics in the future. Notably, while here we focused on time series for clarity, MILCCI is applicable to modalities beyond temporal data.

One potential concern is the scaling ambiguity between components and traces, which MILCCI addresses by normalizing components after each iteration while allowing traces to flexibly vary across trials to capture trial-specific amplitudes. Although component dimension per category is a hyperparameter, MILCCI naturally handles this, as sparsity drives redundant components to zero. Future directions include extending the linear assumption, central to interpretability, to non-linear structures (e.g., via kernelization); parallel optimization or batch processing for large datasets; or dynamic priors over component time evolution (e.g., App. I, J).

## Acknowledgments

N.M. was funded as a fellow by the Kavli NeurData Discovery Institute. Y.C. was funded by grant P01 AG009973 from the National Institute on Aging of the U.S. Public Health Service. G.M. was funded by NSF grant number CCF-2217058 and NSF EFRI 2223822. A.S.C. was supported in part by NSF CAREER award number 2340338. This work was partially carried out at the Advanced Research Computing at Hopkins (ARCH) core facility (rockfish.jhu.edu), which is supported by the National Science Foundation (NSF) grant number OAC1920103.

We thank Prof. James Knierim for his neuroscience insights on the method's utility and for valuable discussions on its applications.

## Code and Data

The code for MILCCI is publicly available over GitHub at
`https://github.com/NogaMudrik/MILCCI`.
The voting data are publicly available through the Harvard Dataverse (Data & Lab, 2017a;b;c). The Wikipedia pageview data are publicly available through Wikimedia (Meta, 2022). The IBL data (Angelaki et al., 2025) are publicly available through the DANDI portal in NWB format (Rübel et al., 2022).

## Impact Statement

This work advances machine learning methods for analyzing multi-view data across diverse domains. We do not anticipate negative societal consequences from this methodological contribution.

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

# Appendix

## Contents

# A. Additional Fitting Details:

### A.1. Initialization:

We initialize the components and traces using dictionary learning (Mairal et al., 2009) with sparsity on the components, using Sklearn.decomposition (Pedregosa et al., 2011) DictionaryLearning. Within the model, sparsity is applied through PyLops (Ravasi & Vasconcelos, 2020) SPGL1's solver (Van Den Berg & Friedlander, 2009).

### A.2. Details about state similarity graph calculation:

MILCCI allows disentangling of categorical, continuous, and non-continuous ordinal categories. It further supports allowing the components to adjust across trials with label change in a way that captures the degree of similarity between the corresponding labels for each category. For example, if we assume neuronal ensembles gradually present compositional shifts over the course of learning, this requires capturing the gradual / ordinal order of trials. Another example is when a certain category represents a continuous variable (e.g., x-position of a stimulus), where again we would like to capture label relationships (i.e., distance between labels).

Hence, MILCCI augments the model with a set of label-driven graphs that are pre-calculated before the beginning of the iterative optimization process and are reused across iterations for smoother cross-component regularization that maintains label similarity. For each category (k), we build the graph $\boldsymbol{\lambda}^{(k)} \in \mathbb{R}^{\#k \times \#k}$, where #k is the number of unique options observed under category (k) (e.g., if category "choice" can be correct / incorrect, then #k=2). This graph captures the degree of similarity between its possible values.

**For categorical labels**, we use a constant value for the graph (e.g., $\lambda_{i,i'}^{(k)} = 1 \ \forall i, i'$).

**For ordinal labels**: We use a Gaussian kernel $\lambda_{i,i'}^{(k)} = e^{\frac{\|k_i - k_{i'}\|_2^2}{2\sigma^2}} \ \forall i, i'$, where $k_i$ and $k_{i'}$ are the $i$-th and $i'$-th option of category (k) (e.g., task difficulty 1 vs. 5). Notably, MILCCI supports integration of diverse graph calculation distance metrics, so one can easily use a different distance metric (not Gaussian) if they assume similarities between labels are captured differently.

After the graph calculation, we recommend normalizing the graph by the per-row absolute sum of 1 to ensure that different labels are regularized to the same degree:

$$\lambda_{i,:}^{(k)} \leftarrow \frac{\lambda_{i,:}^{(k)}}{\|\lambda_{i,:}^{(k)}\|_1} \ \forall i, i'.$$

An $i$-th row of zeros in $\boldsymbol{\lambda}^{(k)}$ means that the $i$-th option of category (k) is not regularized to be consistent with the others. This can be used if there is some intention to create completely trial-varying components that vary flexibly between trials, which is another feature MILCCI offers.

### A.3. About sparsity and consistency hyperparameters

Like most machine learning models, MILCCI includes hyperparameters that control model behavior, though notably fewer than complex models such as deep networks. Two key hyperparameters in MILCCI control component behavior: $\gamma_1$ (sparsity) and $\gamma_2$ (cross-label consistency).

$\gamma_1$ controls the $\ell_1$ regularization on component memberships (Eq. 2). Higher $\gamma_1$ values produce sparser components with fewer non-zero entries, which is crucial for interpretability but potentially missing weaker relationships. Lower $\gamma_1$ values allow denser components that capture more subtle patterns but may include noise. $\gamma_2$ promotes similarity between component variants within the same category via $\ell_2$ regularization on their distances (Eq. 2). Higher $\gamma_2$ values force variants to remain nearly identical, losing the ability to capture label-driven adjustments. Lower $\gamma_2$ values allow more flexibility but risk components diverging too much across labels.

We recommend selecting $\gamma_1$ by examining component sparsity and interpretability (e.g., inspecting the number of non-zero entries and their meaningfulness), and by testing information criteria (e.g., AIC, BIC) post-training. For $\gamma_2$, we recommend analyzing the distribution of pairwise distances between same-category component variants to ensure they remain similar yet allow meaningful adjustments. When domain knowledge is available (e.g., in neuroscience, we may have an estimate of

how many neurons form a group based on the amount of data we recorded), this can further guide hyperparameter selection.

## B. Running Details

### B.1. Data and Code Availability

All real-world data used in this paper are publicly available online, with sources cited in the corresponding sections. Code for the model implementation and synthetic data generation will be made available upon publication.

### B.2. Versions

We trained the model using Python 3.10.4 (conda-forge) with matplotlib 3.8.2, scikit-learn 1.0.2, seaborn 0.11.2, numpy 1.23.5, pandas 1.5.0, PyLops 1.18.2, and SPGL1 0.0.2.

## C. Elastic Net Prior of Components

In Section 3.1, we specified that the component matrices follow a Laplace distribution:

$$\boldsymbol{A}_{nj}^{(k)} \sim \text{Laplace}(0, \frac{1}{\gamma_1}) = \frac{\gamma_1}{2} \exp\left(-\gamma_1 |\boldsymbol{A}_{nj}^{(k)}|\right)$$

When we later extended the model to multiple component variants for each category k with constrained $\ell_2$ distances between them (i.e., $\|\mathcal{A}_{n:i}^{(k)} - \mathcal{A}_{n:i'}^{(k)}\|_2 < \epsilon$), we essentially employ a hierarchical Bayesian framework. The variant-specific components $\mathcal{A}_{n:i}^{(k)}$ thus follow an elastic net prior:

$$p(\{\mathcal{A}_{n:i}^{(k)}\}) \propto \exp\left(-\gamma_1 \sum_{n,j} |\mathcal{A}_{nji}^{(k)}| - \gamma_2 \sum_{i' \neq i} \lambda_{i',i}^{(k)} \|\mathcal{A}_{::i'}^{(k)} - \mathcal{A}_{::i}^{(k)}\|_2^2\right)$$

The first term corresponds to the $\ell_1$ penalty inherited from the Laplace prior, promoting sparsity. The second term introduces an $\ell_2$ penalty on variant differences, encouraging similarity within variant groups. This combination yields elastic net regularization, emerging naturally from the hierarchical structure where variants share statistical strength through their common base component.

## D. Additional Information—Synthetic Experiment

We generated synthetic datasets with **80 channels**, **4 components** (2 categories, $\times$ 2 components adjusting per category), **500 time points per trial**, and **250 trials**. Each trial received one label per axis: category (a) (difficulty)'s labels were sampled from the set $\{I, II, III, IV, V\}$ (5 levels) and category (b) (choice) labels from $\{I, II\}$ (2 levels), yielding up to **10 unique label combinations**. Component-to-neuron maps were initialized for the reference trial with values in $[0.5, 1.0]$, then updated across label variants according to a trial-similarity graph calculated based on trial labels and thresholded at the **60th percentile** to enforce sparsity on the component compositions.

Temporal activity for each label pair was drawn from a Gaussian-process prior with an RBF kernel scaled by a per-sample amplitude: the amplitude was drawn per sample in $\approx [0.2, 1.533]$, and the kernel length scale was drawn per sample in $[0.05, 0.2]$ in normalized time units (0–1), which corresponds roughly to **25–100 time points** given 500 samples per trial. A white-noise term of $1 \times 10^{-8}$ was included in the kernel. For each label we drew one GP sample and then generated multiple similar trial traces by adding multivariate-normal perturbations with covariance scaled by $\sigma^2$, where $\sigma = 0.15$ ($\sigma^2 = 0.0225$), so trials that share a label exhibit correlated dynamics.

One component was designated as a random (trial-varying) component. Component activations were shifted to be nonnegative and rescaled so their 98th percentile matched the 98th percentile of the component maps. The observed data were produced

**Algorithm 1** MILCCI Algorithm

**Input:** Observed trial data $\{Y^{(m)}\}_{m=1}^{M}$, with associated multi-category labels $\{L^{(m)}\}_{m=1}^{M}$.
**Pre-calculate:** Label-similarity graph $\lambda^{(k)}$ for each category (k) (App. A.2).
**Initialize:** Sparse components $\{\mathcal{A}^{(k)}\}_{k\in\text{Categories}}$ and traces $\{\Phi^{(m)}\}_{m=1}^{M}$ (App. A.1)
**repeat**
    **for** each category (k) **do**
        **for** each label value $k_i$ **do**
            Compute residuals for trials with label $k_i$
            Solve for $\mathcal{A}^{(k)}_{::i}$ with cross-label consistency and sparsity via LASSO ( (2))
            Normalize each component to sum to 1 ($\mathcal{A}^{(k)}_{:ji} \leftarrow \frac{\mathcal{A}^{(k)}_{:ji}}{\|\mathcal{A}^{(k)}_{:ji}\|_1}$) to prevent scaling ambiguity with $\Phi$
        **end for**
    **end for**
    **for** each trial $m$ with label $\ell$ **do**
        Build the stacked component matrix $A^{(\ell)}$ by selecting a variant from each $\mathcal{A}^{(k)}$
        Update traces $\Phi^{(m)}$ to minimize data fidelity, smoothness and de-correlation ( (3))
    **end for**
**until** converged

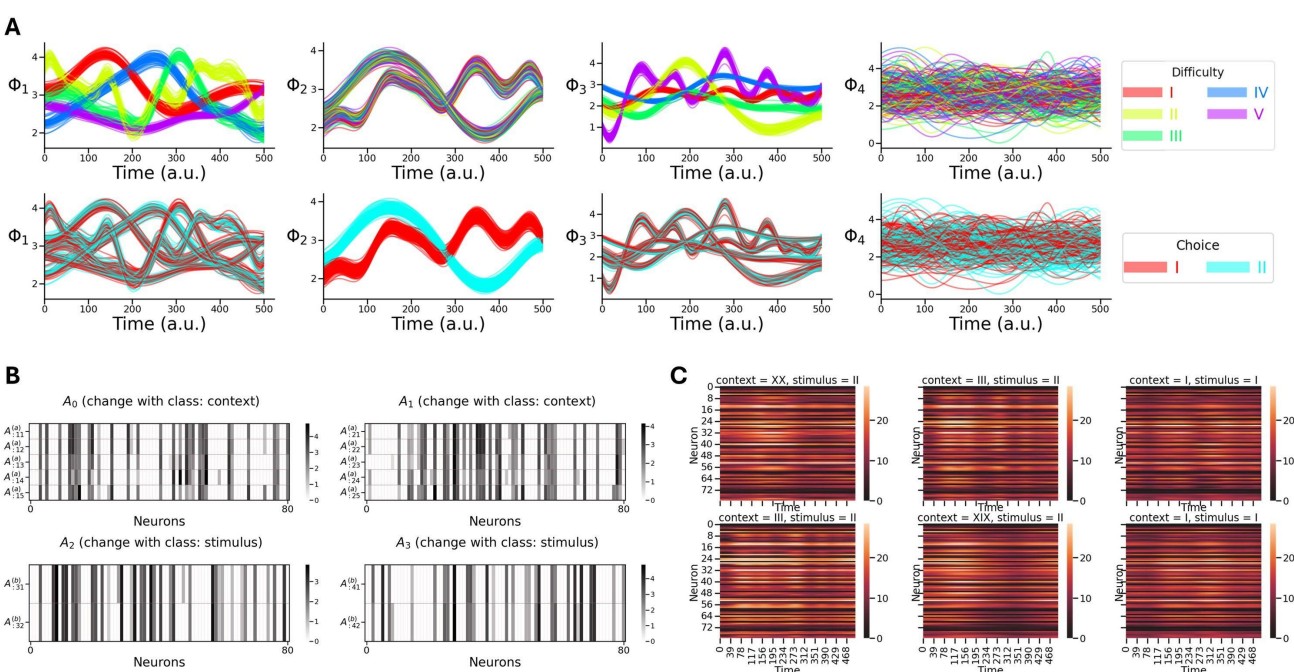

*Figure 6.* **Generated Synthetic Data. A:** Generated traces, colored by difficulty (top) or choice (bottom). **B:** Generated components. Each subplot shows one component and how it varies over the labels of each category (changes across rows). In other words, each subplot corresponds to $\mathcal{A}^{(k)}_{:j:}$ for some component $j$. **C:** Random example generated synthetic trials $\{Y^{(m)}\}$.

by multiplying each trial's component-to-neuron map by that trial's temporal activations, yielding data of shape (**neurons** $\times$ **time** $\times$ **trials**) = (**80** $\times$ **500** $\times$ **250**).

# E. Additional Information—Voting Experiment

## E.1. Voting Data Pre-Processing

Data were acquired from (Data & Lab, 2017a;b;c), which included vote information for presidential, senate, and house elections in 51 states, including Washington, DC. The datasets cover the years 1976 to 2020 for presidential and senate

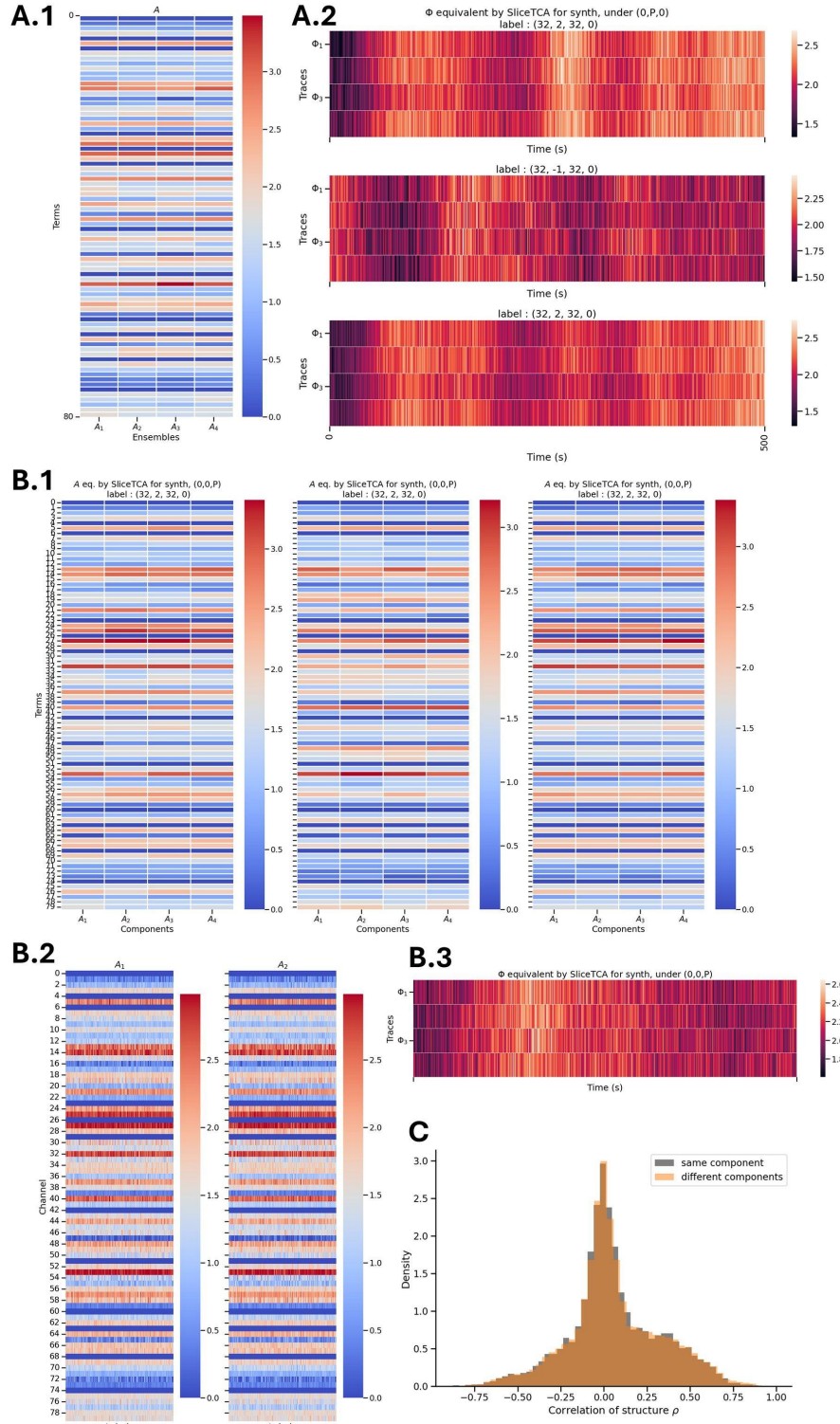

*Figure 7.* **Components and Traces Identified by SliceTCA on synthetic data. A** Results for configuration 1 (Sec. H.2), where **A.1** represents the components and **A.2** represents the corresponding traces from sliceTCA's $A$ matrix. **B** Results for sliceTCA's configuration 2. **B.1** Identified components for 3 example trials; each subplot represents all components of one trial. **B.2** Shows how identified components vary over trials. **B.1 & B.2** together show that sliceTCA identifies components that are very close to each other, with cross-component variability similar to the variability of the same component over trials. This suggests that components are not necessarily matched over trials in terms of identity, as seen in panel **C** which shows the correlation distribution between same components and different components. **B.3** The temporal traces obtained by sliceTCA for configuration 2, shared across all trials.

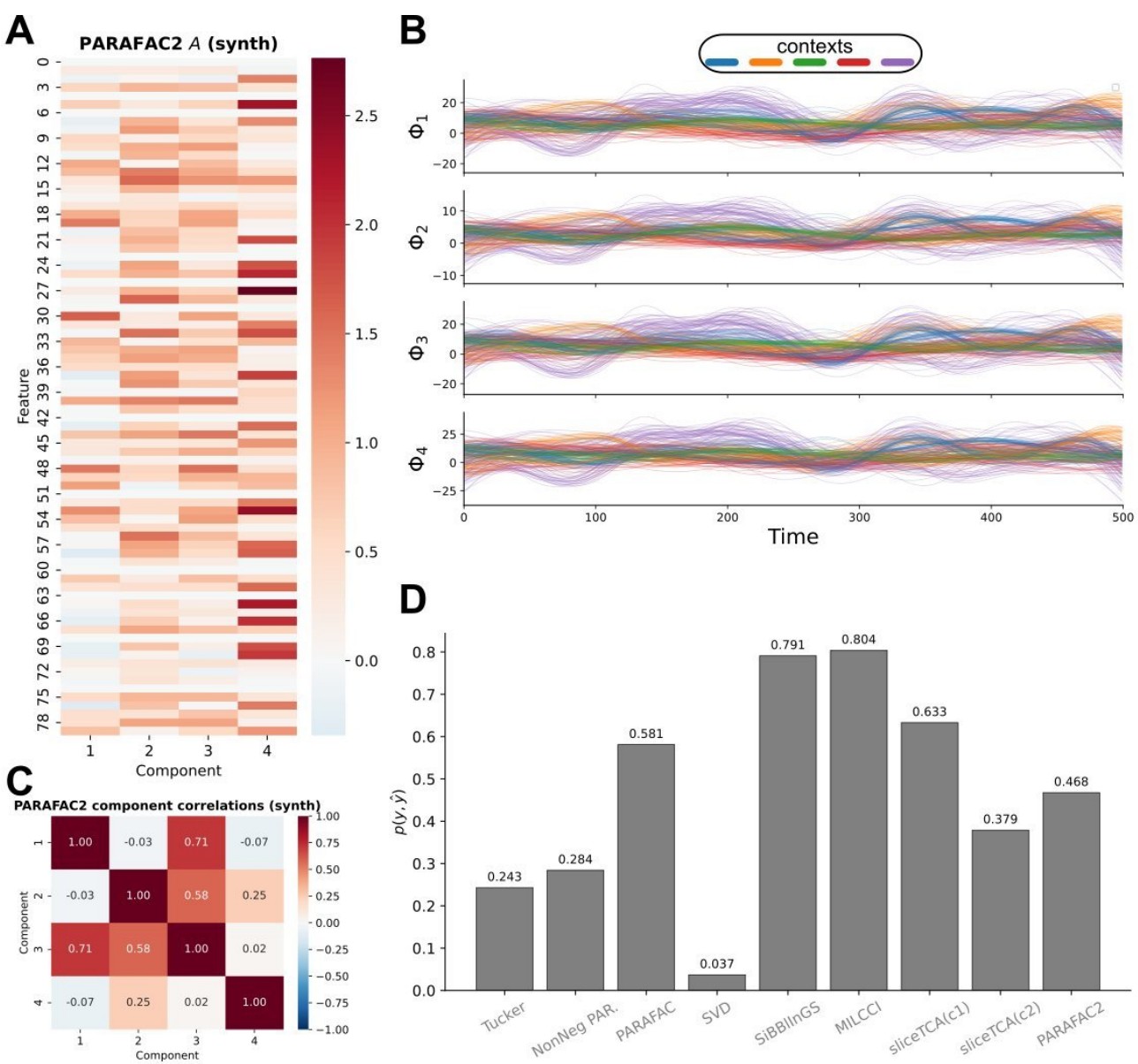

*Figure 8.* **PARAFAC2 for Synthetic Data. A:** Component matrix $A$ identified by PARAFAC2 (rank 4). **B:** Temporal traces colored by context label (5 levels), with mean trace in red. All four components exhibit similar temporal patterns, reflecting PARAFAC2's structural constraint that limits per-component differentiation. **C:** Cross-component correlation matrix showing high correlations between components (up to 0.71), indicating partial degeneracy. **D:** Reconstruction quality $p(y, \hat{y})$ across methods. PARAFAC2 achieves 0.468, below MILCCI (0.804) and SiBBlInGS (0.791).

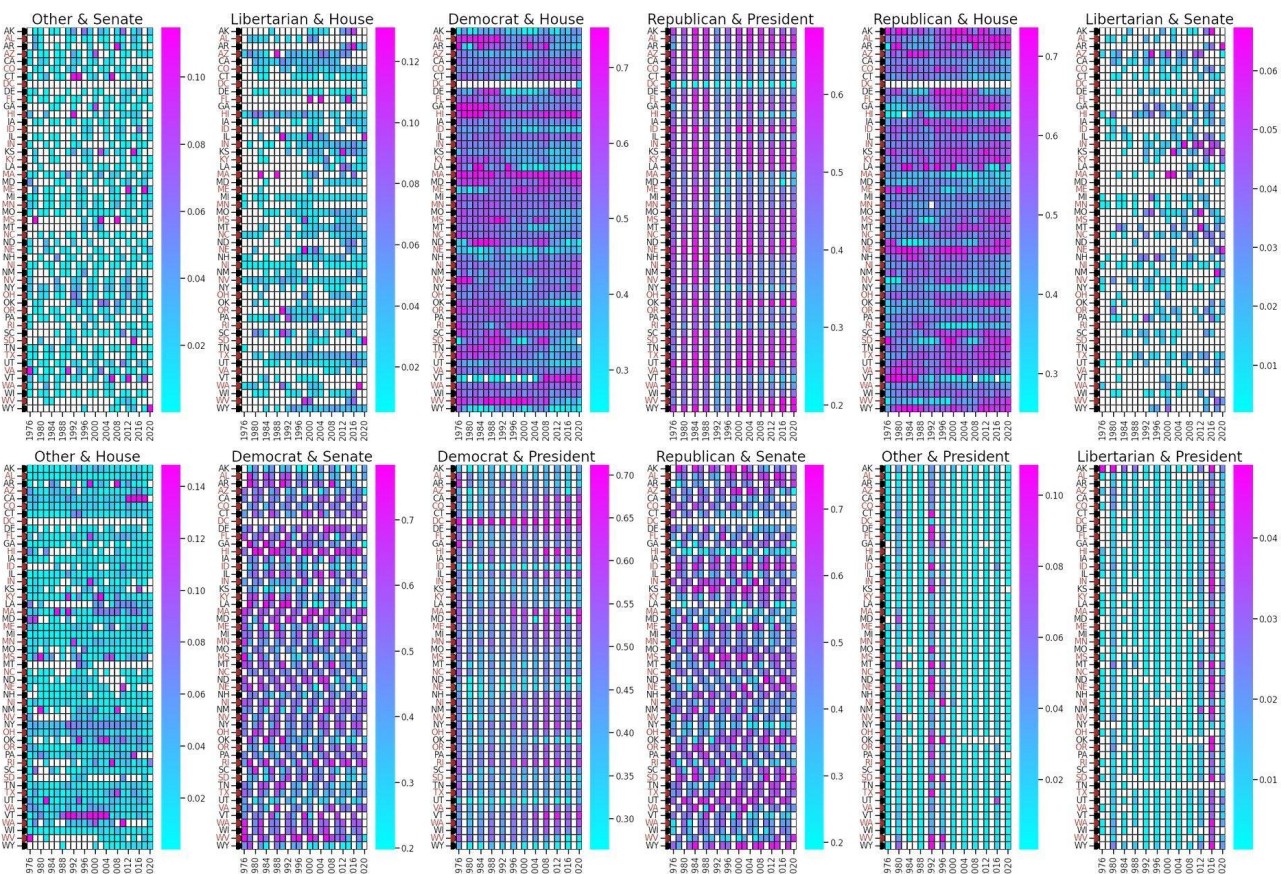

*Figure 9.* **Voting Data**: $(\{\boldsymbol{Y}^{(m)}\}_{m=1}^{M})$ Show Diverse Data Structures

elections, and 1976 to 2022 for house elections. For our analysis, we used the range 1976 to 2020 to ensure that all office types were included, resulting in 23 time points for house and senate, and 12 time points for presidential elections. For each year and each state, we took the total number of votes received by each party category (democrat, libertarian, republican, other) and divided each by the total number of votes cast in that state for that year's election. We excluded special elections. We designed the model to capture two following categories: 1) party (4 options), and 2) office (3 options). We ran MILCCI with $p^{(k)} = 4$ components for each category, that can structurally adjust per class, resulting in $p = 8$ total components. Due to the positive-only nature of the data, we applied a non-negativity constraint on both the unit-to-component memberships and the temporal traces.

### E.2. Additional Findings—Voting Data

Some more interesting voting patterns, beyond these discussed in the main text, can be found in Figs. 16 and 12.

In component $\mathcal{A}_{:4:}^{(\text{party})}$, Washington DC appears as its own distinct cluster. This likely reflects its highly unique political profile as the nation's capital, overwhelmingly Democratic, with very high voter turnout, which differs significantly from other states. In $\mathcal{A}_{:2:}^{(\text{office})}$ (i.e., the 6-th component overall), which unlike previous components changes its composition depending on the office, West Virginia (WV) is included for House and Senate voting but excluded for the Presidency. This component reflects strong Democratic dominance in those legislative elections. The distinction for WV may align with its historical voting pattern of supporting Democrats more in local and state-level offices (House and Senate), while trending more Republican in presidential elections, reflecting a split in voter behavior based on the office. When exploring all traces colored by party (Fig. 16 top), Democrat vs. Republican is the dominant axis, consistently driving the most prominent distinctions over time. In some traces ($\boldsymbol{\Phi}_{:\mathcal{G}_1^{(\text{party})}}$, $\boldsymbol{\Phi}_{:\mathcal{G}_2^{(\text{party})}}$, $\boldsymbol{\Phi}_{:\mathcal{G}_3^{(\text{party})}}$), a noticeable divergence emerges around the year 2000, suggesting that the two parties began to separate more sharply around that period. This trend may correspond to the aftermath of the 2000 presidential election dispute (Bush vs Gore), which catalyzed a lasting partisan split, possibly further

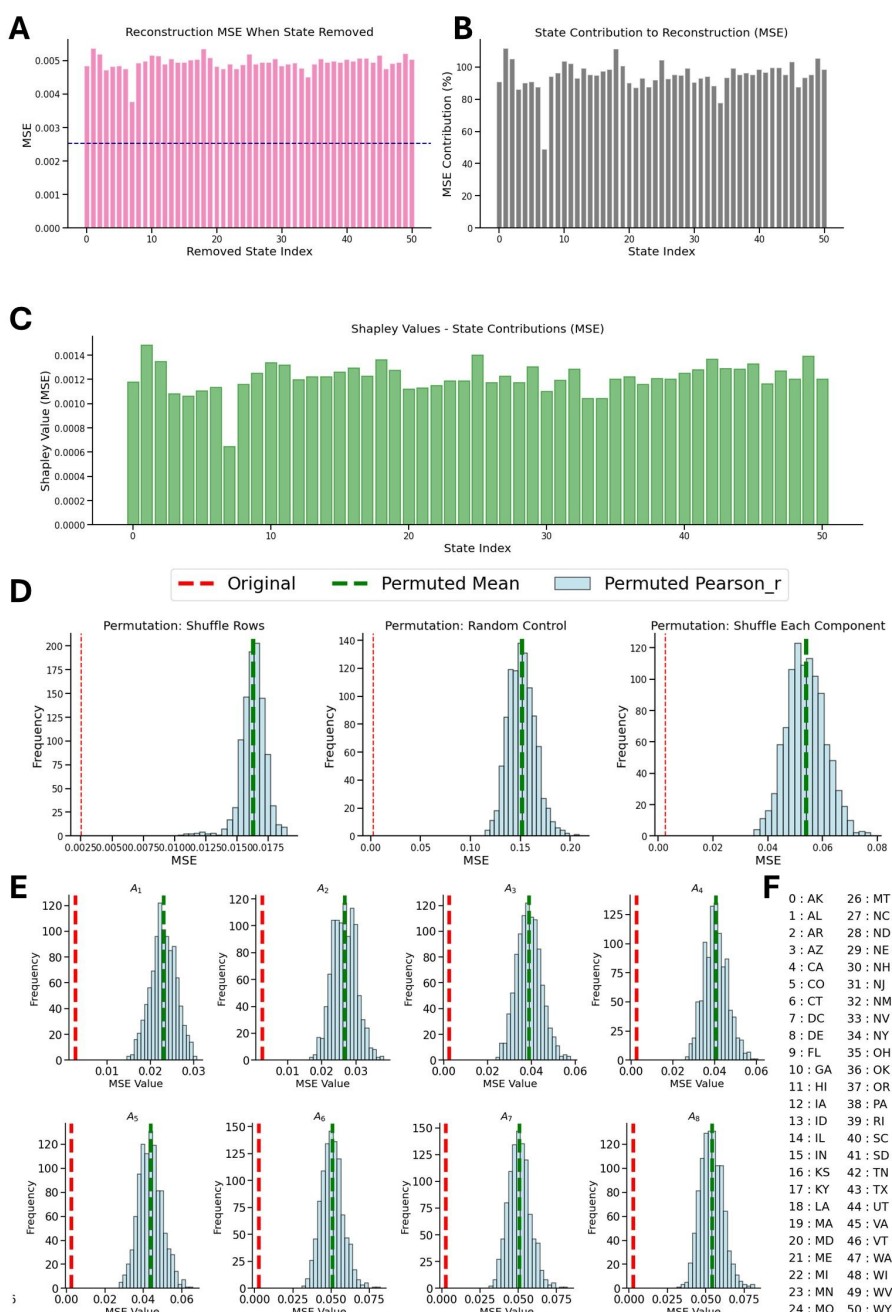

*Figure 10.* Post-hoc validation of MILCCI's discovered voting components (App. E.3). **A:** Leave-one-out analysis showing reconstruction MSE when each state is individually removed, with baseline MSE (dashed line) indicating model performance without omissions. **B:** Individual state contributions to reconstruction, how much each state's removal degrades performance ($100 * \frac{\text{MSE}_{\text{with omission}} - \text{MSE}}{\text{MSE}}$). **C:** Shapley values measuring each state's contribution to overall reconstruction. **D:** Permutation tests comparing original reconstruction error (red line) against three null hypotheses: shuffling state assignments between rows (left), replacing data with random noise (middle), and shuffling states within each component dimension (right). All tests show p < 0.001. **E:** Per-component permutation results demonstrating that discovered components are individually robust. **F:** State abbreviation key.

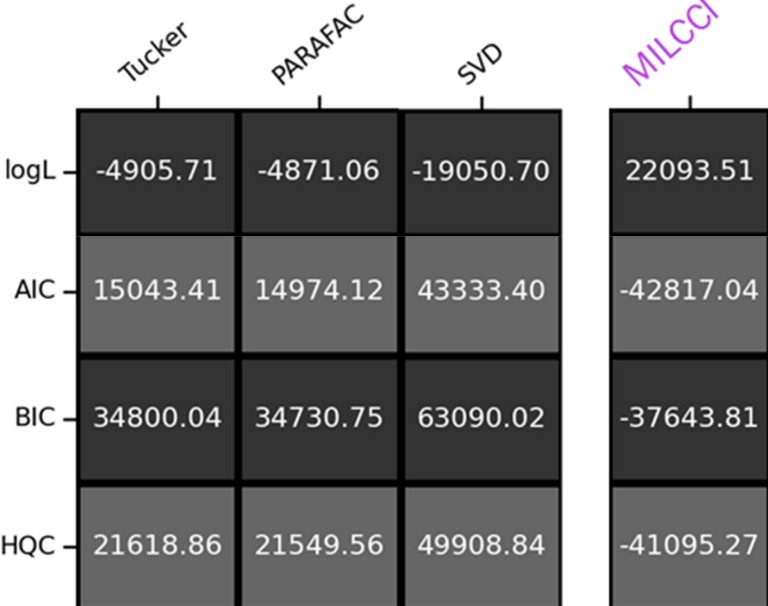

*Figure 11.* **Voting Experiment.** comparison to baselines in reconstruction and information criteria. Comparison to Tucker, PARAFAC, and SVD in terms of reconstruction and information criteria, including 1) log-likelihood of the observations given the identified components, and 2) information criteria that balance reconstruction and model complexity: AIC (Akaike Information Criterion), BIC (Bayesian Information Criterion), and HQC (Hannan-Quinn Criterion). Lower values indicate better performance.

intensified by the ideological polarization following 9/11. Other traces, such as $\mathbf{\Phi}_{:\mathcal{G}_2^{(office)}:}$ and $\mathbf{\Phi}_{:\mathcal{G}_4^{(office)}}$, show broader fluctuations over time in opposite directions, which may reflect deeper, long-standing historical or structural differences between the parties. Trace $\mathbf{\Phi}_{:\mathcal{G}_3^{(office)}}$ appears to capture short-term variation or noise, including year-to-year peaks in party voting behavior. In contrast, the projections based on the office (Fig. 16) exhibit far fewer separations. The traces overlap substantially and show wide confidence intervals, suggesting that electoral behavior is less differentiated by office type than by party affiliation.

### E.3. Post-hoc Validation Analysis on Voting Data

To validate that MILCCI discovers genuine voting structure rather than spurious correlations, we perform comprehensive statistical analyses that examine individual state contributions and test against multiple null hypotheses (Fig. 10).

We remove each state from the component matrix by zeroing out that state's contribution and measure reconstruction degradation. For each of the N states, we calculate reconstruction MSE with the modified components where the target state is excluded. Fig. 10A shows the reconstruction MSE when each state is removed, with the baseline MSE (dashed line) that represents performance with all states included. Fig. 10B displays each state's contribution calculated as the percentage change in MSE relative to baseline performance.

We calculate each state's fair contribution with game-theoretic Shapley values, which consider all possible combinations of states. Due to computational constraints with N=51 states (which require $2^{52}$ calculations), we use approximation with 500 random combinations. For each combination, we include only combination members in the components matrix while we zero non-combination states, then calculate each state's marginal contribution as the difference in reconstruction quality when that state is added to the combination. Fig. 10C shows the Shapley values, which quantify each state's individual contribution to overall reconstruction quality.

We test three null hypotheses by comparison of original reconstruction performance against randomized versions:

**Shuffle Rows** (Fig. 10D, left):
We randomly reassign complete state voting profiles to different state positions. This tests whether the specific correspondence between geographic states and their voting behavioral patterns is meaningful.

**Random Control** (Fig. 10D, middle):
We replace the entire components matrix with random values drawn from a normal distribution with the same global mean

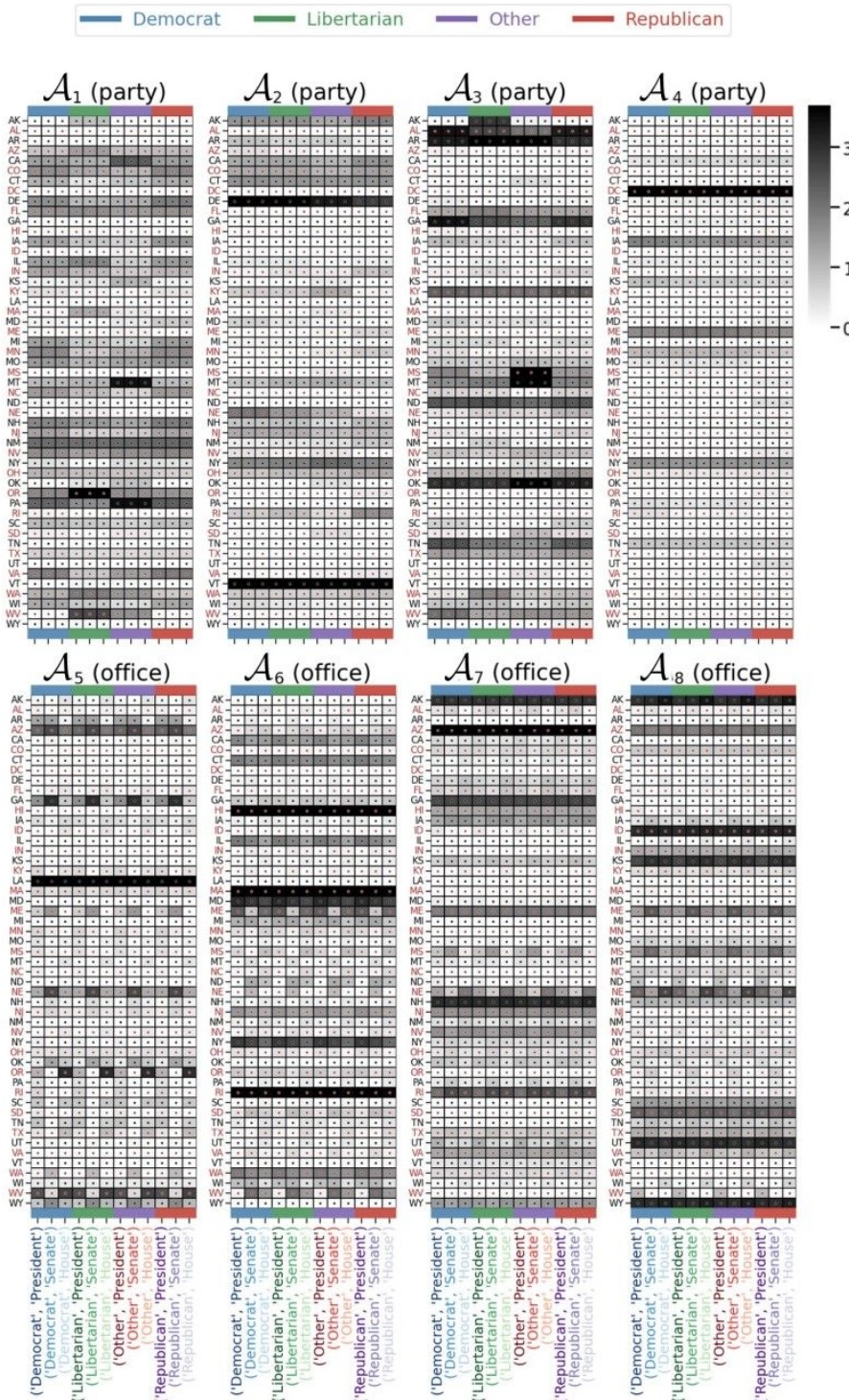

*Figure 12.* Identified Ensembles for Voting Experiment.

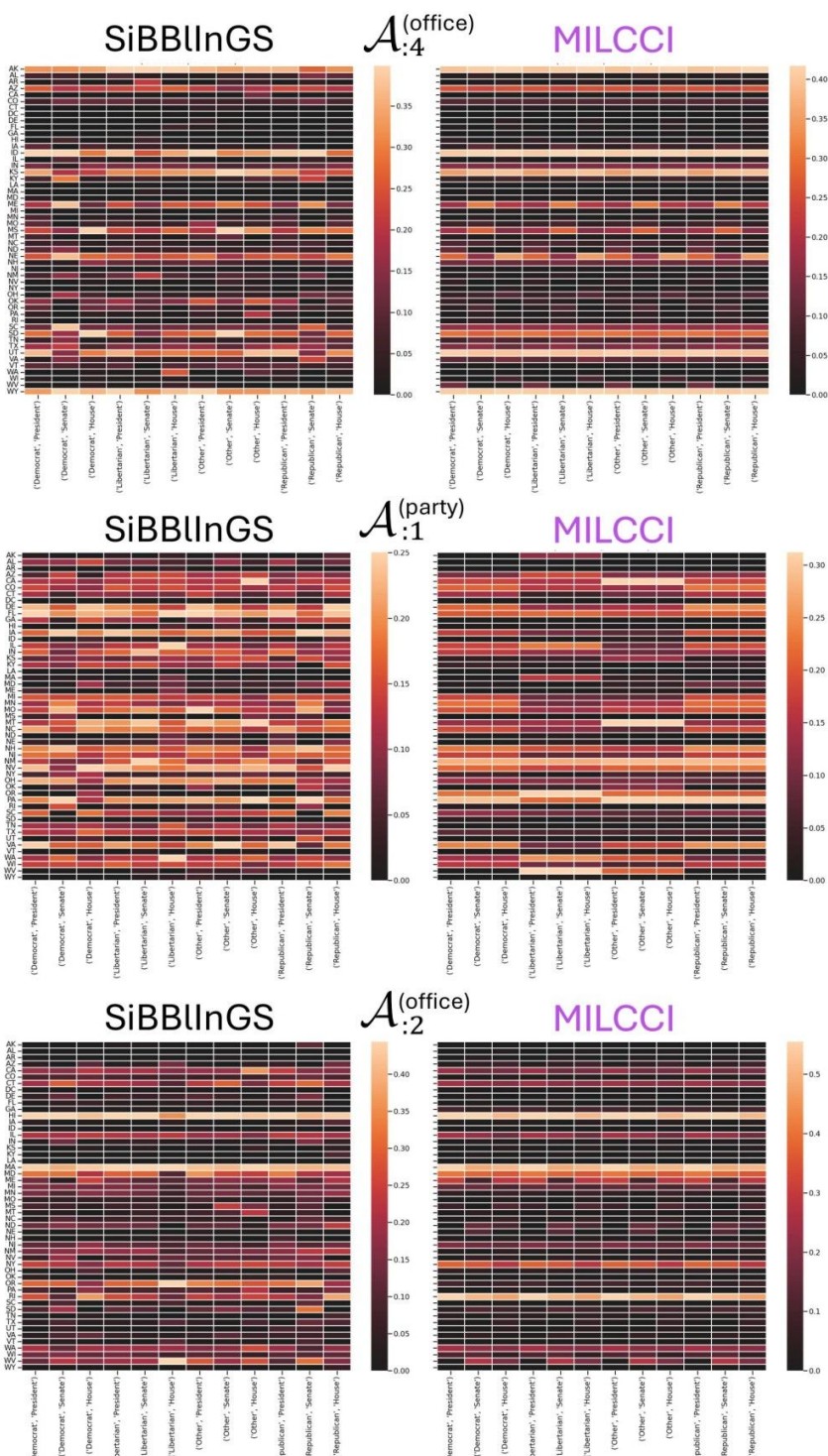

*Figure 13.* Comparison of components identified by MILCCI and SiBBlInGS for three example labels under the same parameters and seed. MILCCI components were duplicated to match the x-tick labels of SiBBlInGS. SiBBlInGS shows uninterpretable changes across every label, even when parts are shared, whereas MILCCI disentangles them.

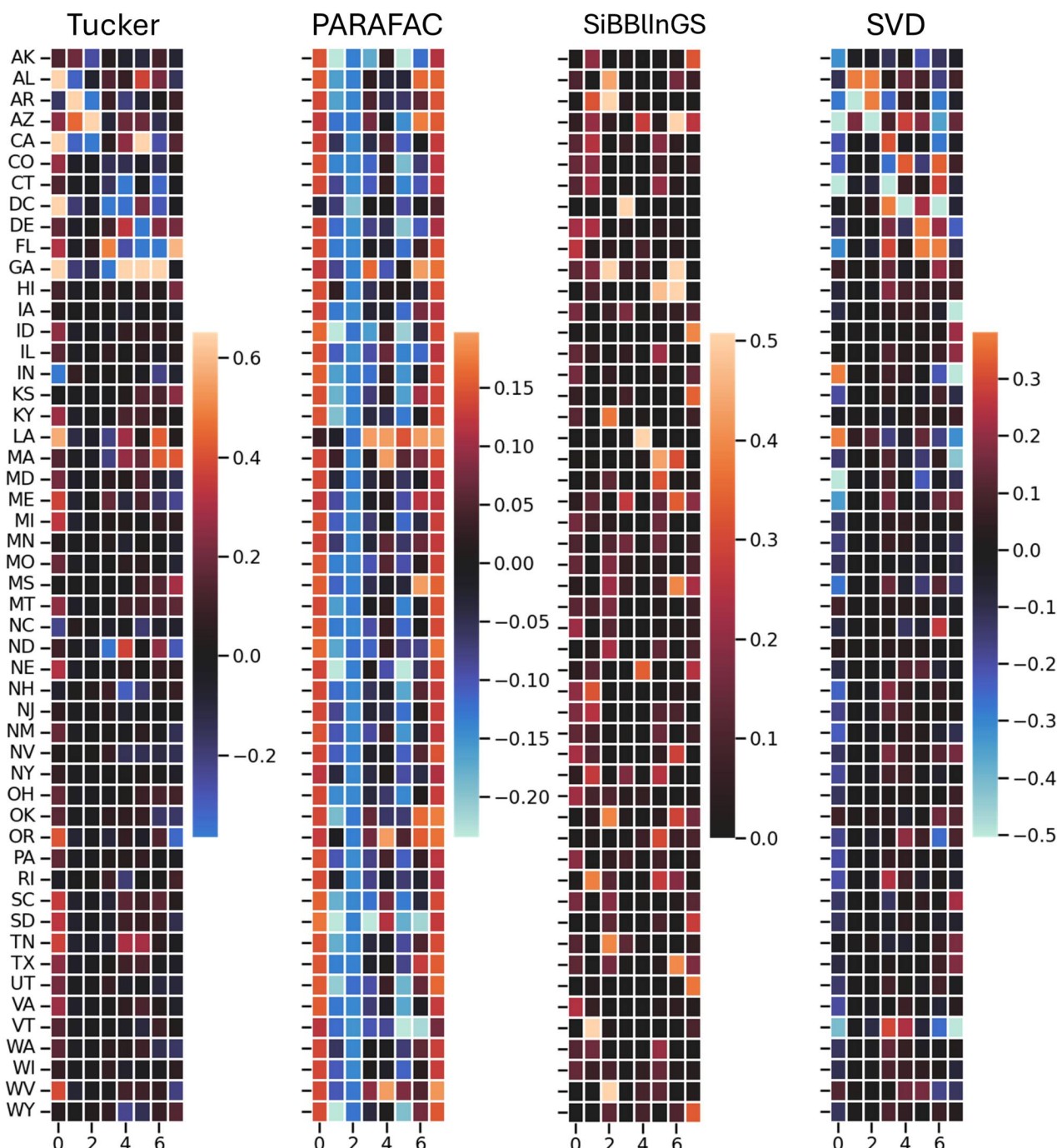

*Figure 14.* **Voting Experiment Baseline Comparison**. Components identified by the following baselines: 1) Tucker Decomposition (HOSVD), 2) PARAFAC, 3) SiBBlInGS (for a single random label entry: (Democrat, President)), 4) SVD (on all concatenated trials).

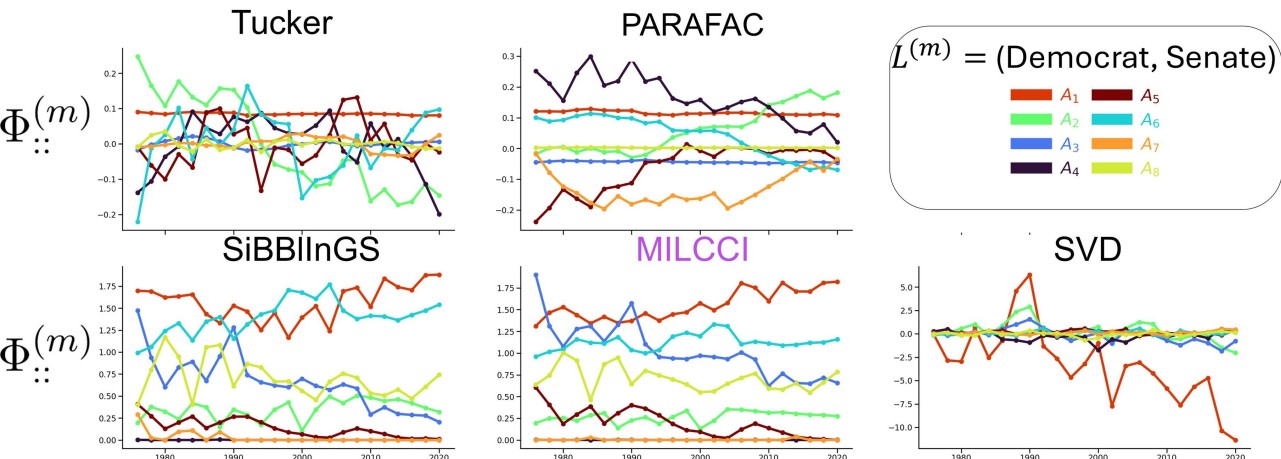

*Figure 15.* **Voting Experiment**. Traces Identified by MILCCI compared to the other baselines for example random trial.

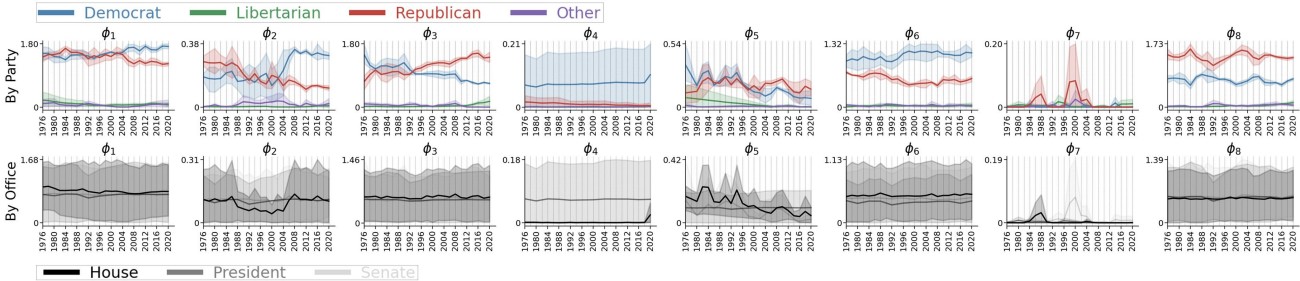

*Figure 16.* **Voting Traces.** Top: Colored by Party. Bottom: Colored by Office.

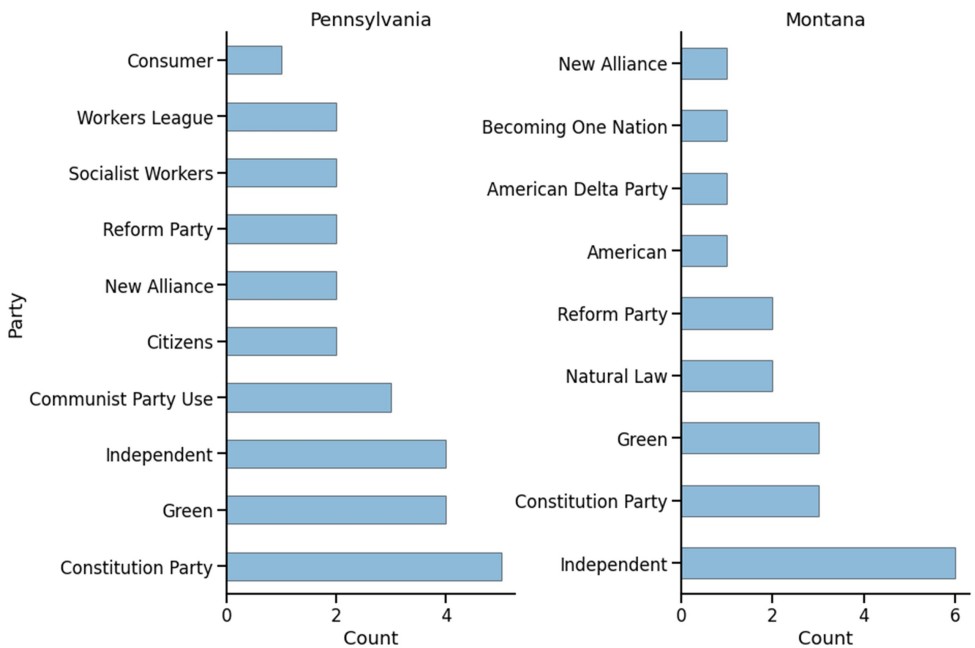

*Figure 17.* Top "Other" parties instance counts for Montana and Pennsylvania

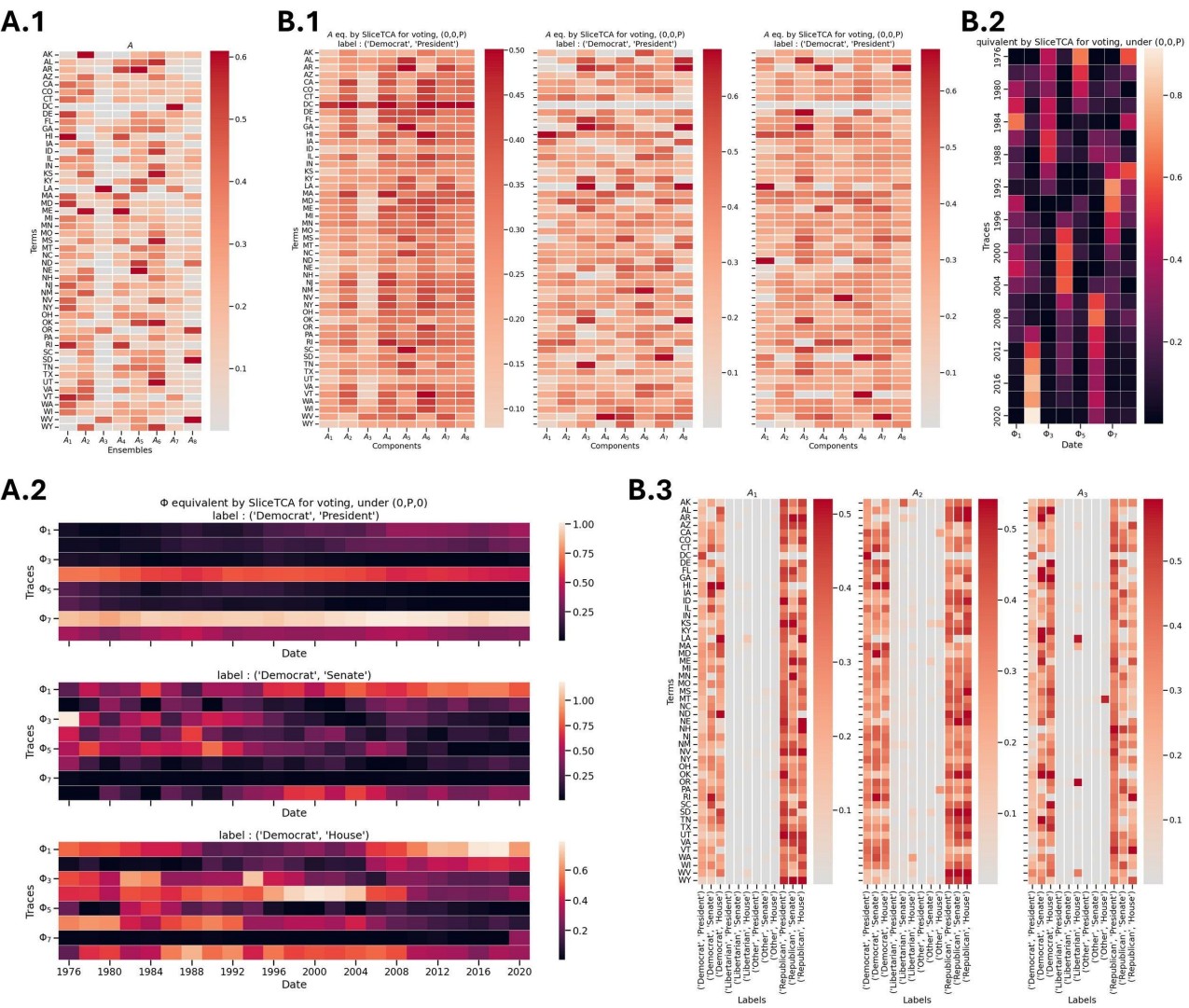

*Figure 18.* **Voting experiment: SliceTCA comparison.** **A.1** Components from the fixed component case (i.e., from u). **A.2** Traces from the fixed component case (i.e., from A). **B.1** Components from the varying component case (i.e., B). Each subplot represents a trial and columns show its different components. **B.2** Components from the varying component case (i.e., v). **B.3** Components from the varying component case. Each subplot represents one component and columns represent trials. We can observe that there is no consistency within the same component (B.3) across trials compared to across components.

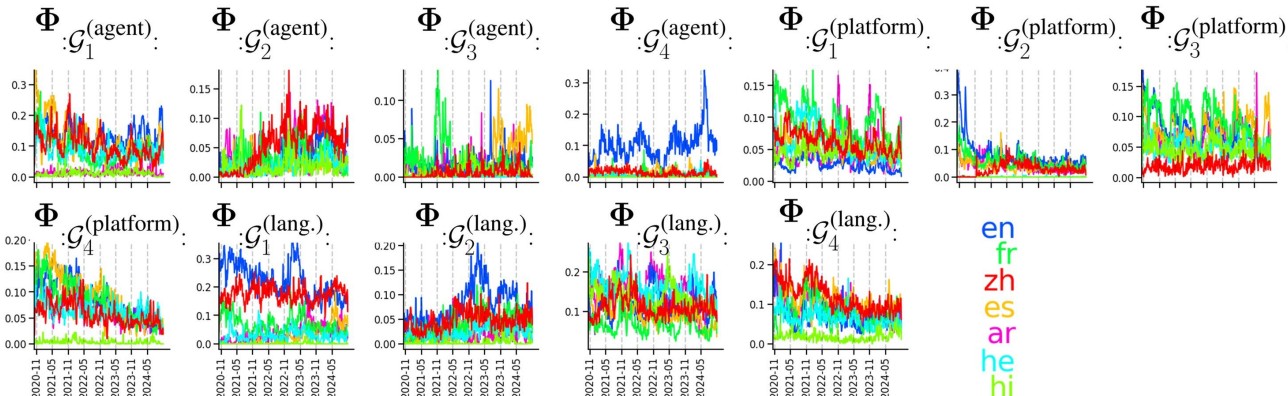

**A:** PARAFAC2 A (voting)

**B:** $\Phi_1$, $\Phi_2$, $\Phi_3$, $\Phi_4$

**C:** PARAFAC2 component correlations (voting)

**D:** Voting: Log-Likelihood

**E:** Voting: Information Criteria Comparison

*Figure 19.* **PARAFAC2 for Voting Data. A:** Component matrix $A$ identified by PARAFAC2 (rank 8), representing state-level loadings. **B:** Temporal traces per trial (gray, with markers) and mean (red), shown for 4 of 8 components. With only 23 time points and 12 trials, traces show limited temporal variation. **C:** Cross-component correlation matrix reveals moderate to high correlations between components. **D:** Log-likelihood comparison; MILCCI substantially outperforms all baselines. **E:** AIC and HQC comparison; MILCCI achieves the lowest (best) values, while PARAFAC2 performs comparably to Tucker and PARAFAC.

$\Phi_{:\mathcal{G}_1^{(\text{agent})}:}$  $\Phi_{:\mathcal{G}_2^{(\text{agent})}:}$  $\Phi_{:\mathcal{G}_3^{(\text{agent})}:}$  $\Phi_{:\mathcal{G}_4^{(\text{agent})}:}$  $\Phi_{:\mathcal{G}_1^{(\text{platform})}:}$  $\Phi_{:\mathcal{G}_2^{(\text{platform})}:}$  $\Phi_{:\mathcal{G}_3^{(\text{platform})}:}$

$\Phi_{:\mathcal{G}_4^{(\text{platform})}:}$  $\Phi_{:\mathcal{G}_1^{(\text{lang.})}:}$  $\Phi_{:\mathcal{G}_2^{(\text{lang.})}:}$  $\Phi_{:\mathcal{G}_3^{(\text{lang.})}:}$  $\Phi_{:\mathcal{G}_4^{(\text{lang.})}:}$

en fr zh es ar he hi

*Figure 20.* **Wikipedia Pageview Experiment**. Traces colored by Language (Arabic, English, Spanish, French, Hebrew, Hindi, Chinese) Across All Ensembles.

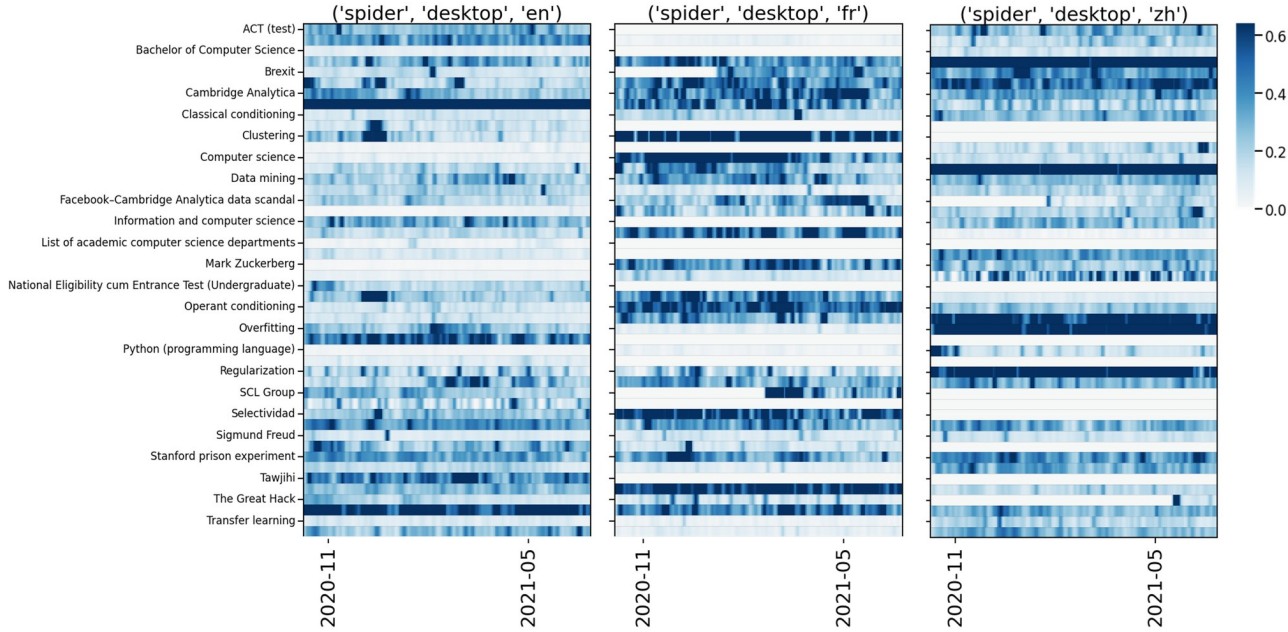

*Figure 21.* Wikipedia Pageview Data, example Trials

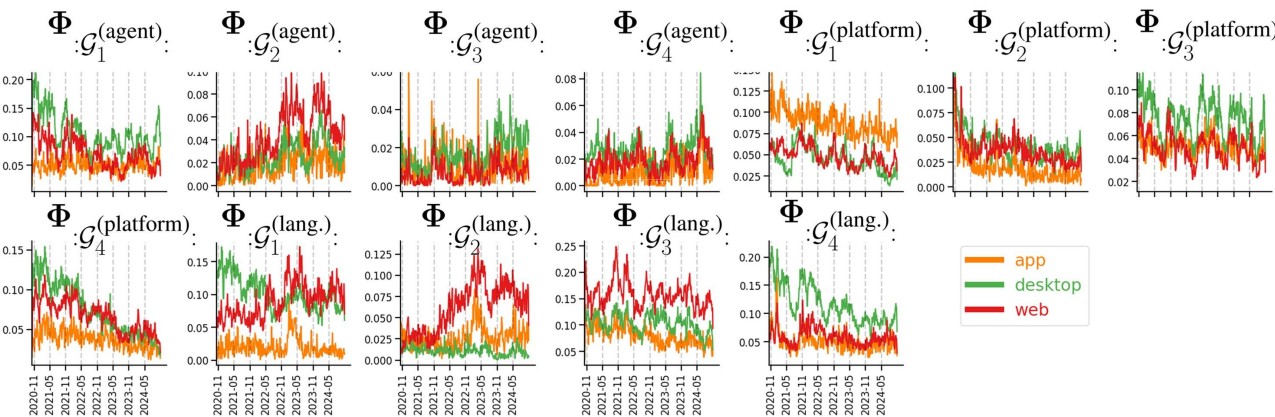

*Figure 22.* **Wikipedia Pageview Experiment**. Traces colored by Platform (desktop, mobile web, mobile app) Across All Ensembles

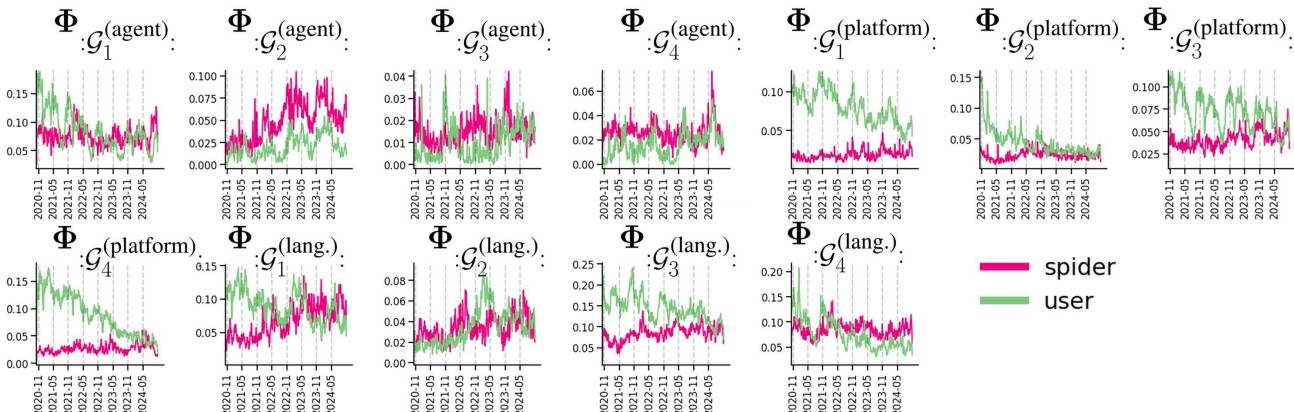

*Figure 23.* **Wikipedia Pageview Experiment**. Traces colored by agent (Spider vs. User) Across All Ensembles

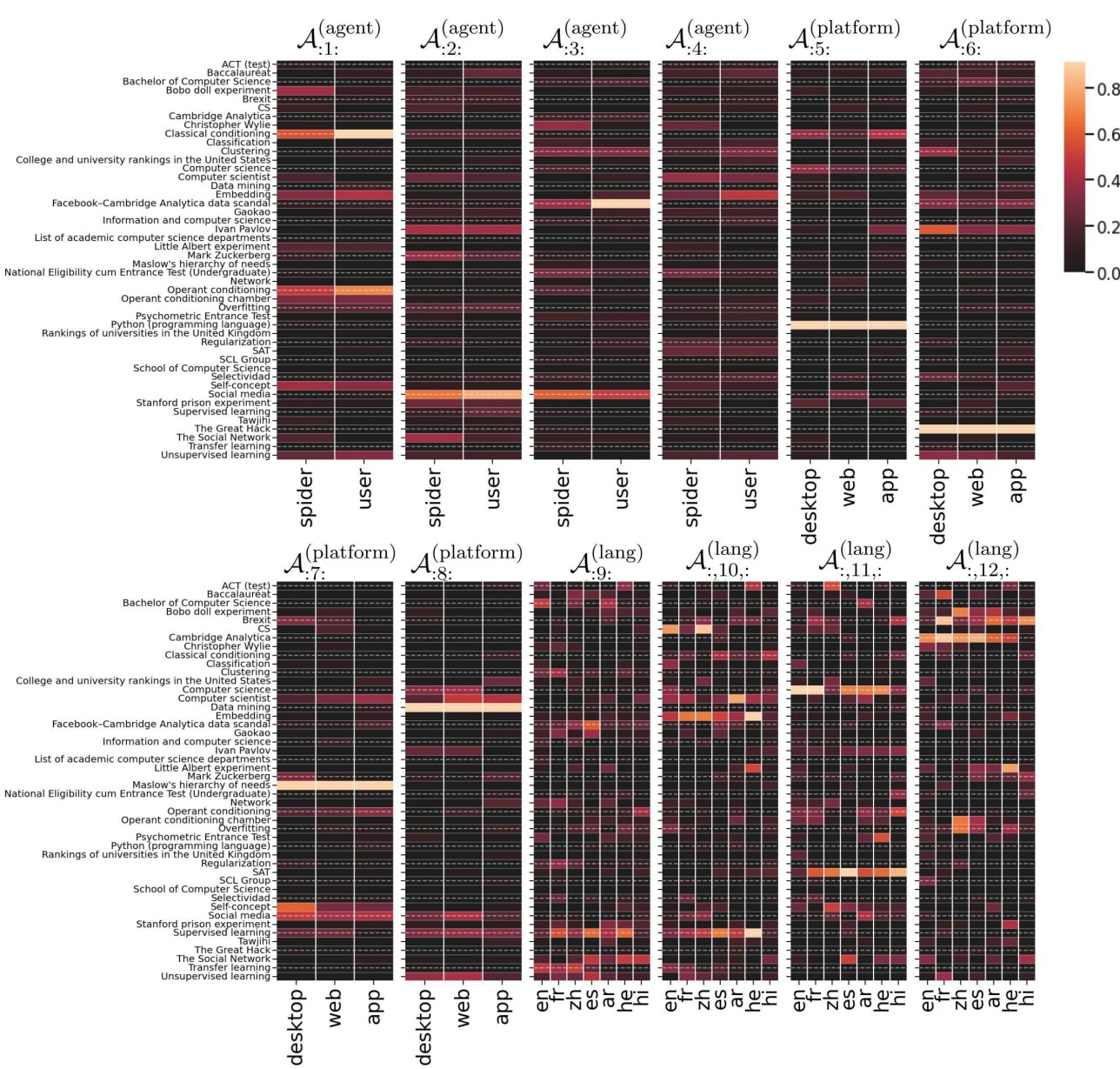

*Figure 24.* Components identified for Wikipedia Pageview Experiment.

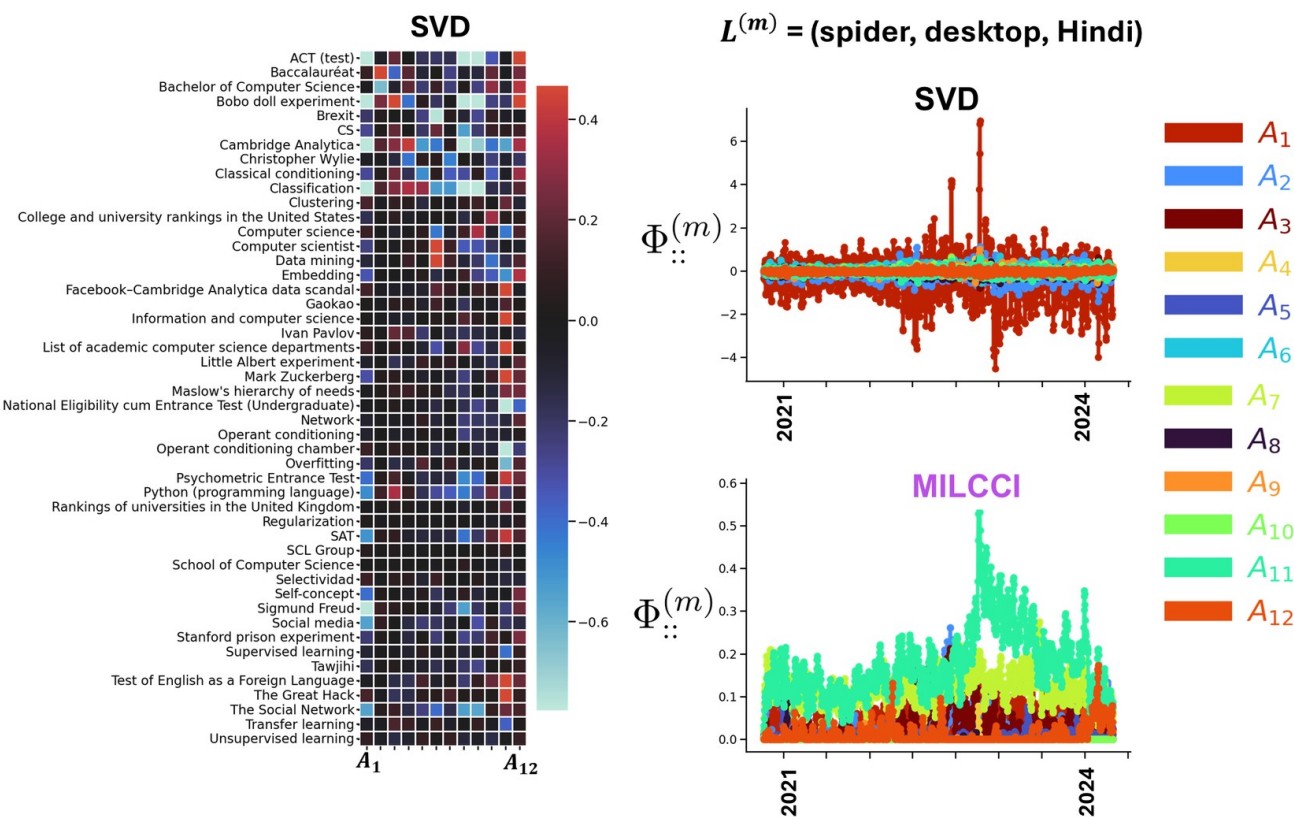

*Figure 25.* **Wikipedia Experiment Compared to SVD**. Components identified by SVD (compositions on the left, example trial traces on the right).

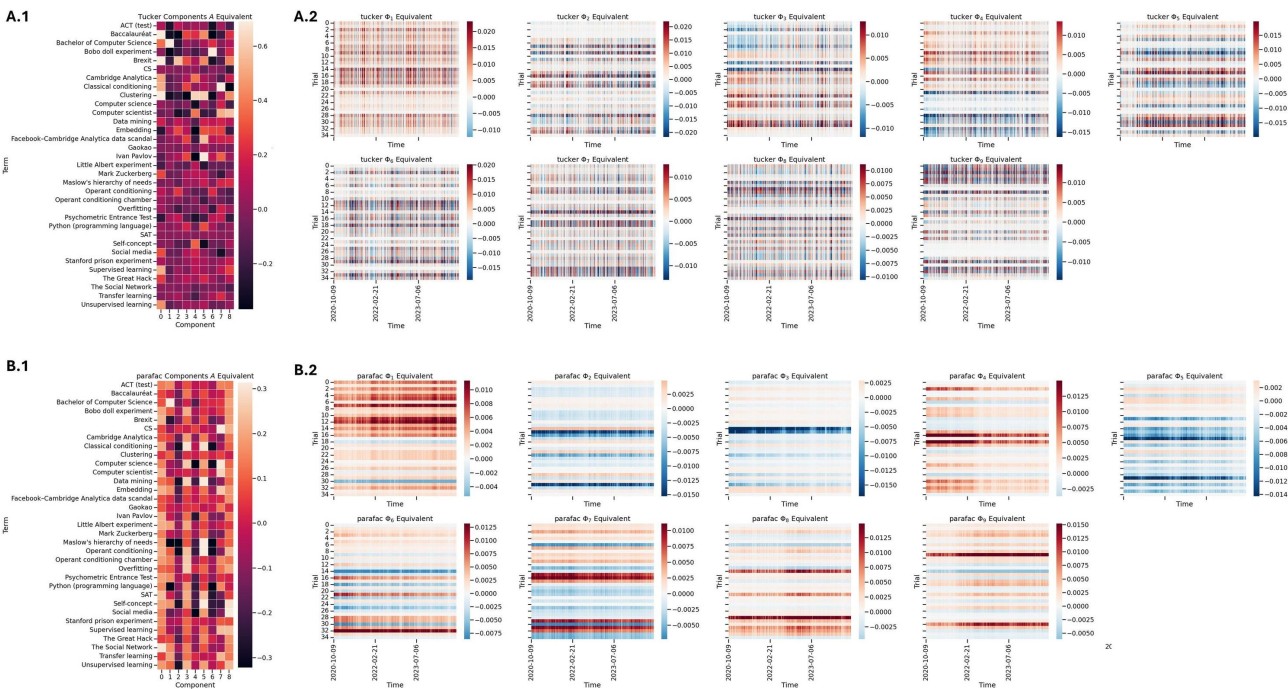

*Figure 26.* Components Identified by Tucker (**A.1**, **A.2**) and PARAFAC (**B.1**, **B.2**) for the Wikipedia Experiment, see App. H

| Abbreviation | State | Abbreviation | State |
|---|---|---|---|
| AL | ALABAMA | AK | ALASKA |
| AZ | ARIZONA | AR | ARKANSAS |
| CA | CALIFORNIA | CO | COLORADO |
| CT | CONNECTICUT | DE | DELAWARE |
| DC | DISTRICT OF COLUMBIA | FL | FLORIDA |
| GA | GEORGIA | HI | HAWAII |
| ID | IDAHO | IL | ILLINOIS |
| IN | INDIANA | IA | IOWA |
| KS | KANSAS | KY | KENTUCKY |
| LA | LOUISIANA | ME | MAINE |
| MD | MARYLAND | MA | MASSACHUSETTS |
| MI | MICHIGAN | MN | MINNESOTA |
| MS | MISSISSIPPI | MO | MISSOURI |
| MT | MONTANA | NE | NEBRASKA |
| NV | NEVADA | NH | NEW HAMPSHIRE |
| NJ | NEW JERSEY | NM | NEW MEXICO |
| NY | NEW YORK | NC | NORTH CAROLINA |
| ND | NORTH DAKOTA | OH | OHIO |
| OK | OKLAHOMA | OR | OREGON |
| PA | PENNSYLVANIA | RI | RHODE ISLAND |
| SC | SOUTH CAROLINA | SD | SOUTH DAKOTA |
| TN | TENNESSEE | TX | TEXAS |
| UT | UTAH | VT | VERMONT |
| VA | VIRGINIA | WA | WASHINGTON |
| WV | WEST VIRGINIA | WI | WISCONSIN |
| WY | WYOMING | | |

*Table 2.* List of US States and Their Abbreviations

and standard deviation as the original data. This tests against pure statistical noise baseline.

**Shuffle Each Component** (Fig. 10D, right):
For each component dimension, we randomly permute the assignment of states within that component while we preserve component-wise statistics. This tests whether the specific coordination between components within each state matters.

Each permutation test runs 1000 iterations, with p-values calculated as the fraction of permuted reconstructions that achieve equal or better performance than the original.

Component-specific permutation results (Fig. 10E) demonstrate that discovered patterns are robust across individual voting components. We store intermediate reconstruction results for each component, which allows examination of component-specific robustness to randomization. Fig. 10F provides the state abbreviation key for reference.

All statistical tests yield $p < 0.001$, which provides evidence that MILCCI's discovered voting patterns represent genuine structure.

## F. Additional Information–Wikipedia Experiment

### F.1. Wikipedia Pageview Data Pre-Processing

We extracted daily Wikipedia Pageview data from October 9, 2020, to October 29, 2024 ($T = 1482$ time points) for 32 diverse pages ("terms") related to college, computer science, machine learning, and psychology majors ( (Meta, 2022)). Notably, we intentionally chose topics with both corollaries and co-variates. For each term, we collected data separately for three access platforms: (1) desktop, (2) mobile web, and (3) mobile app. We also distinguished the agent accessing the data: 1) a user or 2) a spider (for spider data was extracted only for web and desktop platforms due to extreme sparsity of app +

*Table 3.* Parties Detailed vs. Simplified Versions. 'Other' includes only parties with at least 10 instances over all years & states.

| Democrat | Republican | Other | Libertarian |
|---|---|---|---|
| Democrat; Democratic-Farmer-Labor; Democratic-Nonpartisan League; Democratic-Npl; Democrat (Not Identified On Ballot) | Republican | Prohibition; Independent; American Independent; U.S. Labor; Socialist Workers; American; Conservative; Socialist Labor; Independent American; Constitution; Socialist; Liberty Union; Statesman; Citizens; New Alliance; Workers World; Workers League; Independence; Populist; Nominated By Petition; Grassroots; No Party Affiliation; Green; Natural Law; Unaffiliated; Other; Working Families; Alliance; Non-Affiliated; Constitution Party; American Independent Party; Communist Party Use; Peace & Freedom; Taxpayers Party; Reform Party; U.S. Taxpayers Party; Socialism And Liberation Party; American Delta Party; American Solidarity Party; Party For Socialism And Liberation; Becoming One Nation | Libertarian |

spider combination).

We focused on seven languages representing diverse world regions: English (en), Chinese (zh), Spanish (es), Hindi (hi), Arabic (ar), French (fr), and Hebrew (he). To ensure comparability, data for each language were range-normalized across all terms and time points using the 99th percentile to reduce outlier influence: $Y_{:,:,l} \leftarrow (Y_{:,:,l} - \min(Y_{:,:,l}))/\mathrm{perc}(Y_{:,:,l}, 99)$, where $Y_{:,:,l}$ is the full dataset for language $l$.

Each term was then normalized across all languages and time points using the same 99th percentile procedure: $Y_{k,:,:} \leftarrow (Y_{k,:,:} - \min(Y_{k,:,:}))/\mathrm{perc}(Y_{k,:,:}, 99)$, where $Y_{k,:,:}$ denotes the full time-course of term $k$ across languages. This results in overall three categories candidate for compositional adjustments in the data: (1) agent (user or spider), (2) platform (desktop / mobile web / mobile app), and (3) language (one of the seven listed).

### F.2. Clarification on Findings—Wikipedia Data

Lists of terms of components mentioned in main text (Sec. 4):

- $\mathcal{A}_{:,1,:}^{(\text{agent})}$ **(Psychology)**: Classical conditioning; Bobo doll experiment; Operant conditioning; Self-concept; Little Albert experiment; Unsupervised learning; Embedding;

- $\mathcal{A}_{:,2,:}^{(\text{agent})}$: **(Social Media)**: The Social Network (movie); Social media; Ivan Pavlov;Mark Zuckerberg.

- $\mathcal{A}_{:,4,:}^{(\text{platform})}$: **(Computer Science)**: Data mining; Computer science; Supervised learning; Unsupervised learning; Computer scientist; Social media;

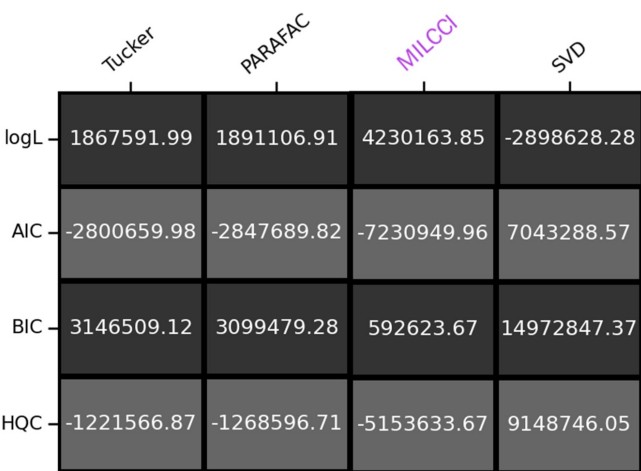

*Figure 27.* Information Criteria for Wikipedia Experiment, MILCCI vs. baselines. PARAFAC and Tucker did not converge of 12 components due to SVD instability. Notably, PARAFAC and Tucker encountered instability issues, and hence the results presented for Tucker and PARAFAC here are for rank 9 and added noise with $\sigma = 0.1$ (Tucker) and $\sigma = 0.2$ (PARAFAC).

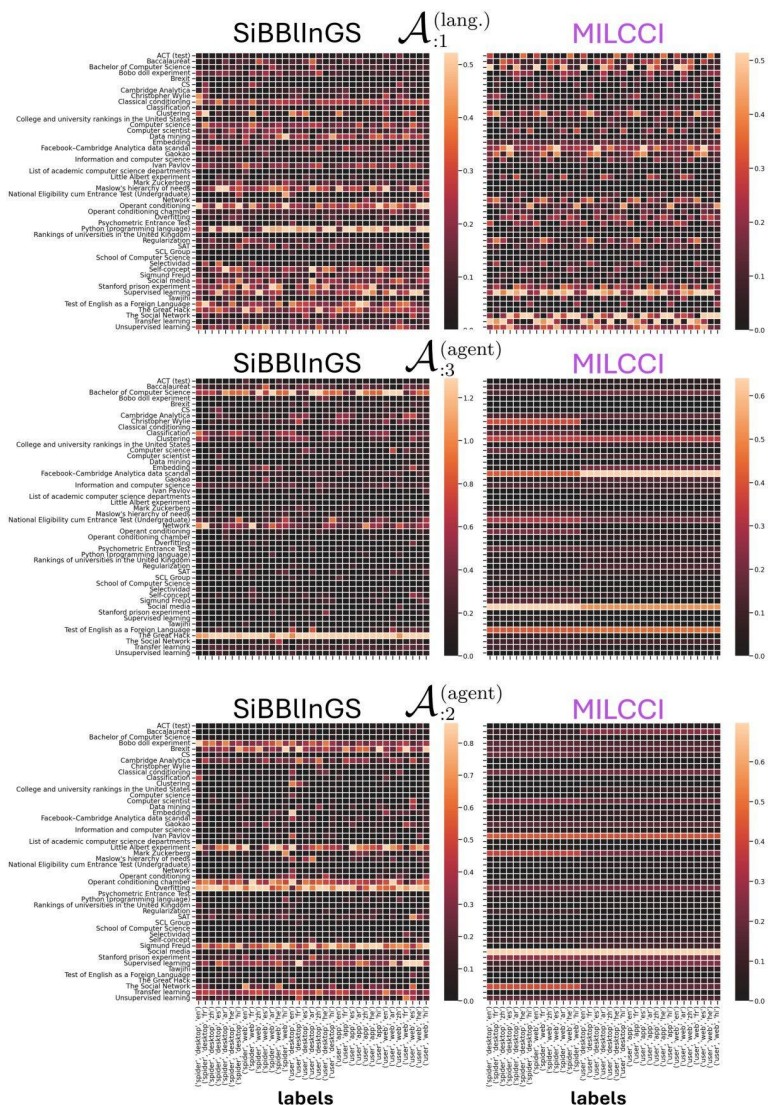

*Figure 28.* **Wikipedia Components, MILCCI vs. SiBBlInGS**. SiBBlInGS components display compositional changes scattered across labels, rather than the category-specific adjustments captured by MILCCI. *Note:* MILCCI components are shown here with duplicate columns to align with SiBBlInGS components for visualization.

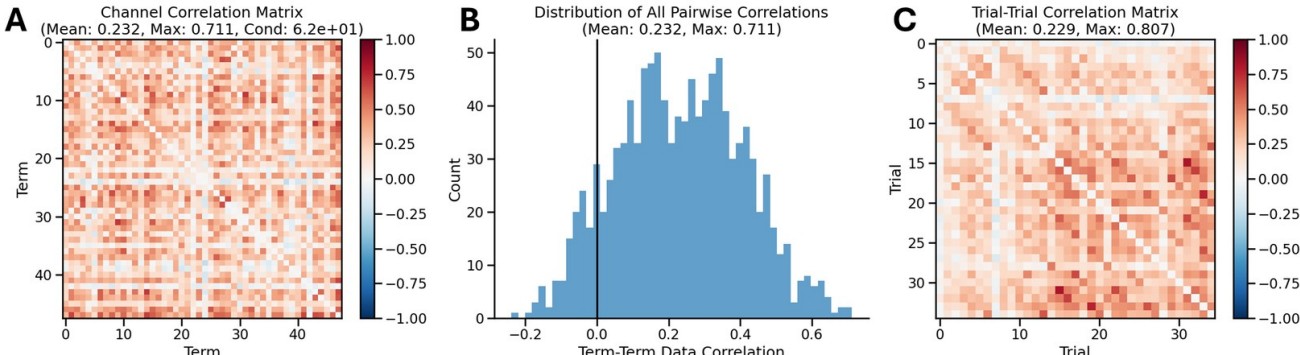

*Figure 29.* Correlation analysis of Wikipedia dataset reveals moderate correlations that do not explain baseline convergence failures. (A) Channel (term-term) correlation matrix shows moderate correlations with maximum of 0.711 and good condition number (6.2e+01). (B) Distribution of all pairwise correlations demonstrates that most correlations are moderate, with no high correlations (>0.8). (C) Trial-trial correlation matrix shows similar moderate correlation patterns (max 0.807). The data has full rank (48/48) and reasonable correlation structure, indicating that baseline convergence failures (Tucker, PARAFAC) stem from algorithmic limitations rather than problematic data characteristics.

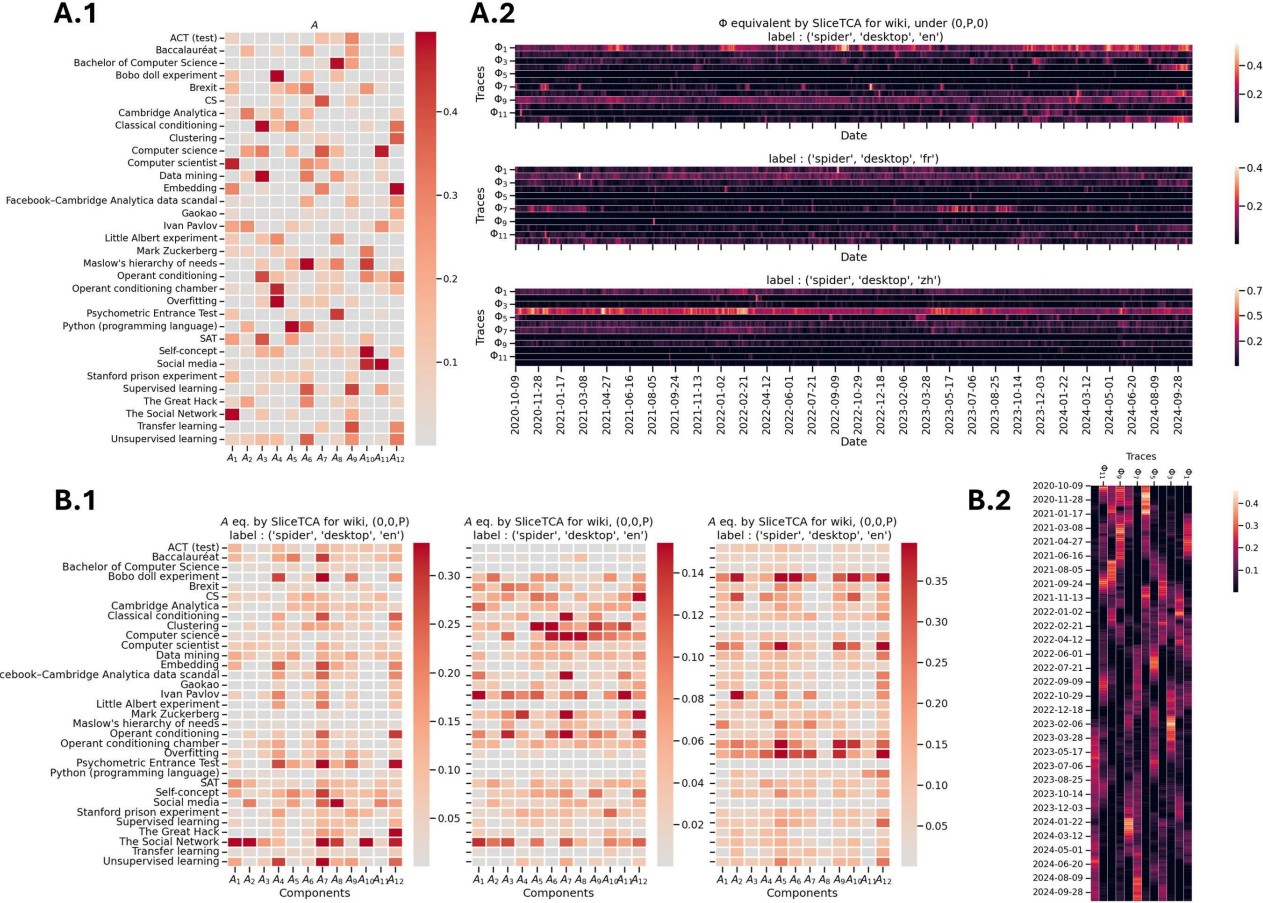

*Figure 30.* **Wikipedia experiment: SliceTCA comparison.** Components equivalent to MILCCI's were extracted from SliceTCA's configuration 1 (see Sec. H.2), i.e., components extracted from sliceTCA's $u$ vector (**A.1**) and temporal traces from sliceTCA's $A$ matrix (**A.2**). For the second configuration (**B** panels), components (MILCCI's $\mathcal{A}$) were extracted from SliceTCA's $C$ (**B.1**) and traces from SliceTCA's $v$ (**B.2**). See details in Section H.2.

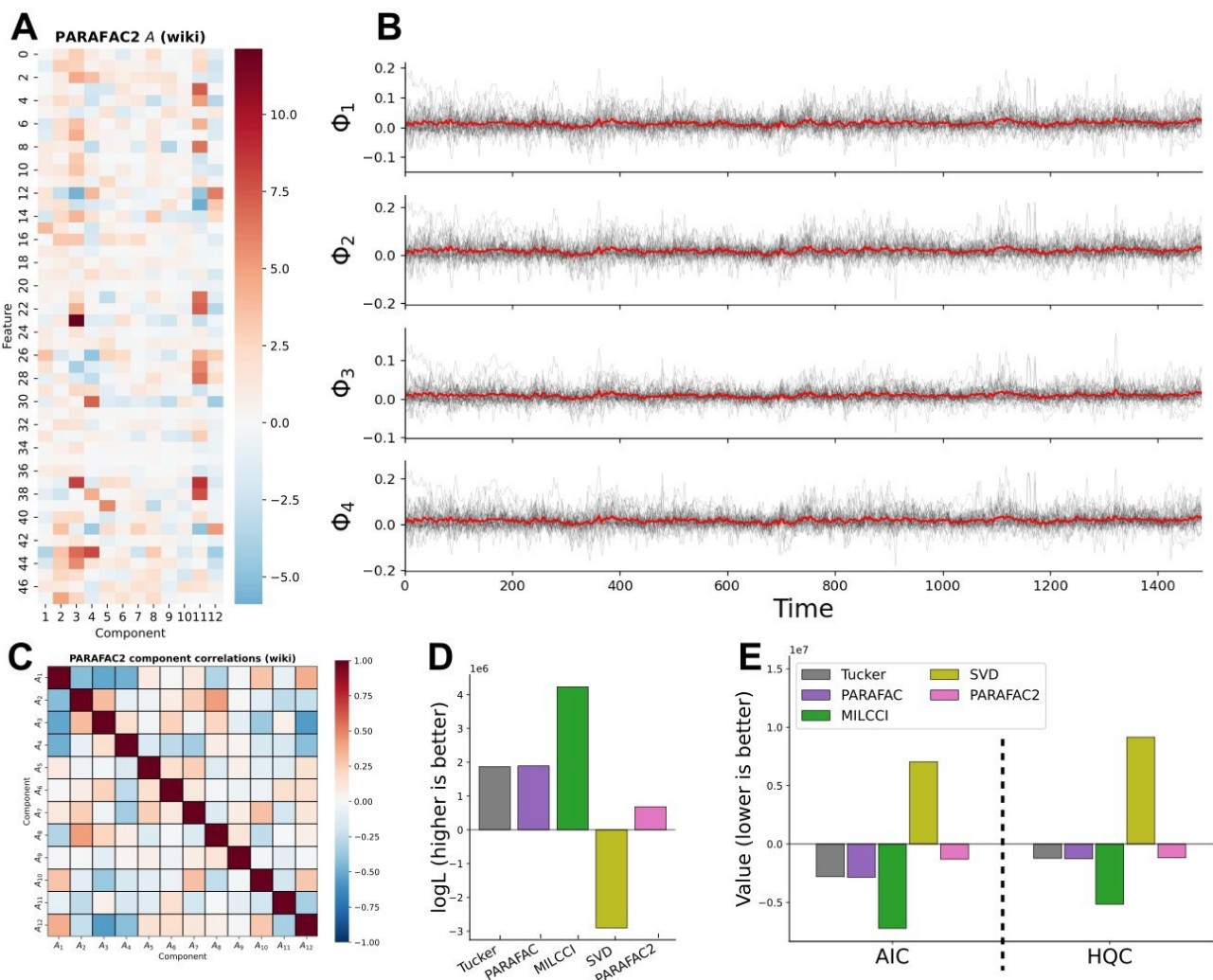

*Figure 31.* **PARAFAC2 for Wikipedia Data. A:** Component matrix $A$ identified by PARAFAC2 (rank 12). **B:** Temporal traces across all trials (gray) with mean in red, shown for 4 of 12 components. All components display nearly identical temporal patterns, indicating severe degeneracy at rank 12. **C:** Cross-component correlation matrix confirms high pairwise correlations (most $> 0.9$), showing that PARAFAC2 fails to identify distinct temporal structure at this rank. **D:** Log-likelihood comparison; MILCCI achieves the highest value. **E:** AIC and HQC comparison; MILCCI achieves the lowest (best) values across both criteria.

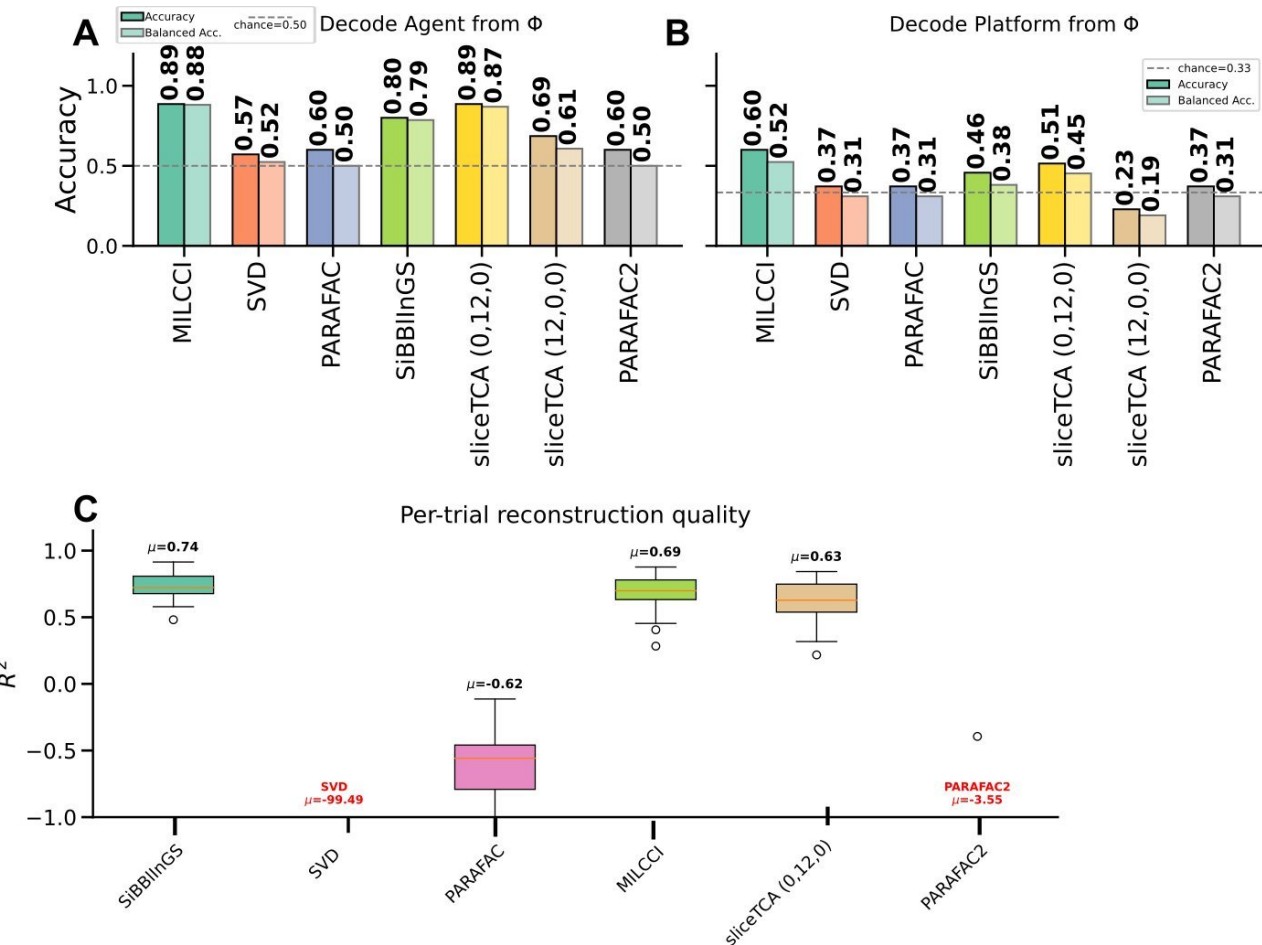

*Figure 32.* **Cross-validated decoding and reconstruction quality for the Wikipedia experiment. A–B:** Five-fold cross-validated decoding accuracy from temporal traces ($\mathbf{\Phi}$) for agent (**A**) and platform (**B**) categories across all methods. Dark bars indicate standard accuracy; light bars indicate balanced accuracy. Dashed line indicates chance level. **C:** Per-trial reconstruction quality ($R^2 = 1 - |\mathbf{Y}^{(m)} - A^{(m)}\mathbf{\Phi}^{(m)}|^2/|\mathbf{Y}^{(m)} - \bar{\mathbf{Y}}^{(m)}|^2$). Methods with $R^2 \ll 0$ (SVD, PARAFAC2) are annotated in red. Features were extracted by normalizing each trace to unit mean and computing the average activation within 6 equal-length temporal windows. MILCCI achieves the highest decoding accuracy across both categories.

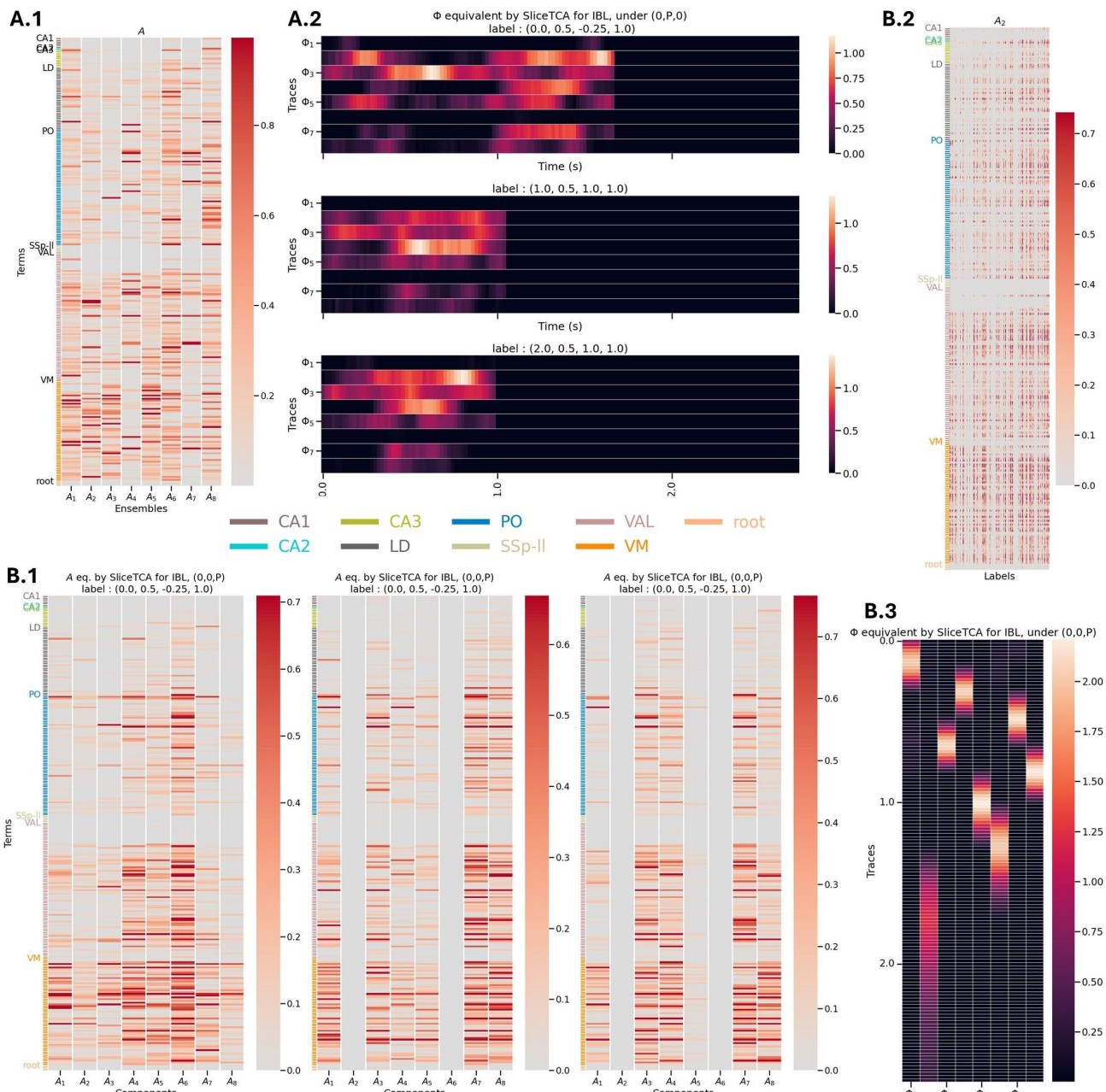

*Figure 33.* **IBL experiment: SliceTCA components. A:** sliceTCA's configuration 1 (see Sec. H.2). **A.1:** Components from the fixed component case (i.e., from sliceTCA's $u$). **A.2:** Traces from the fixed component case (i.e., from sliceTCA's $A$).
**B:** sliceTCA's configuration 2: **B.1:** Components from the varying component case. **B.2:** Components showing how they vary over labels (example: component 2). **B.3:** Traces from the sliceTCA's $v$ vector. See details in Section H.2.

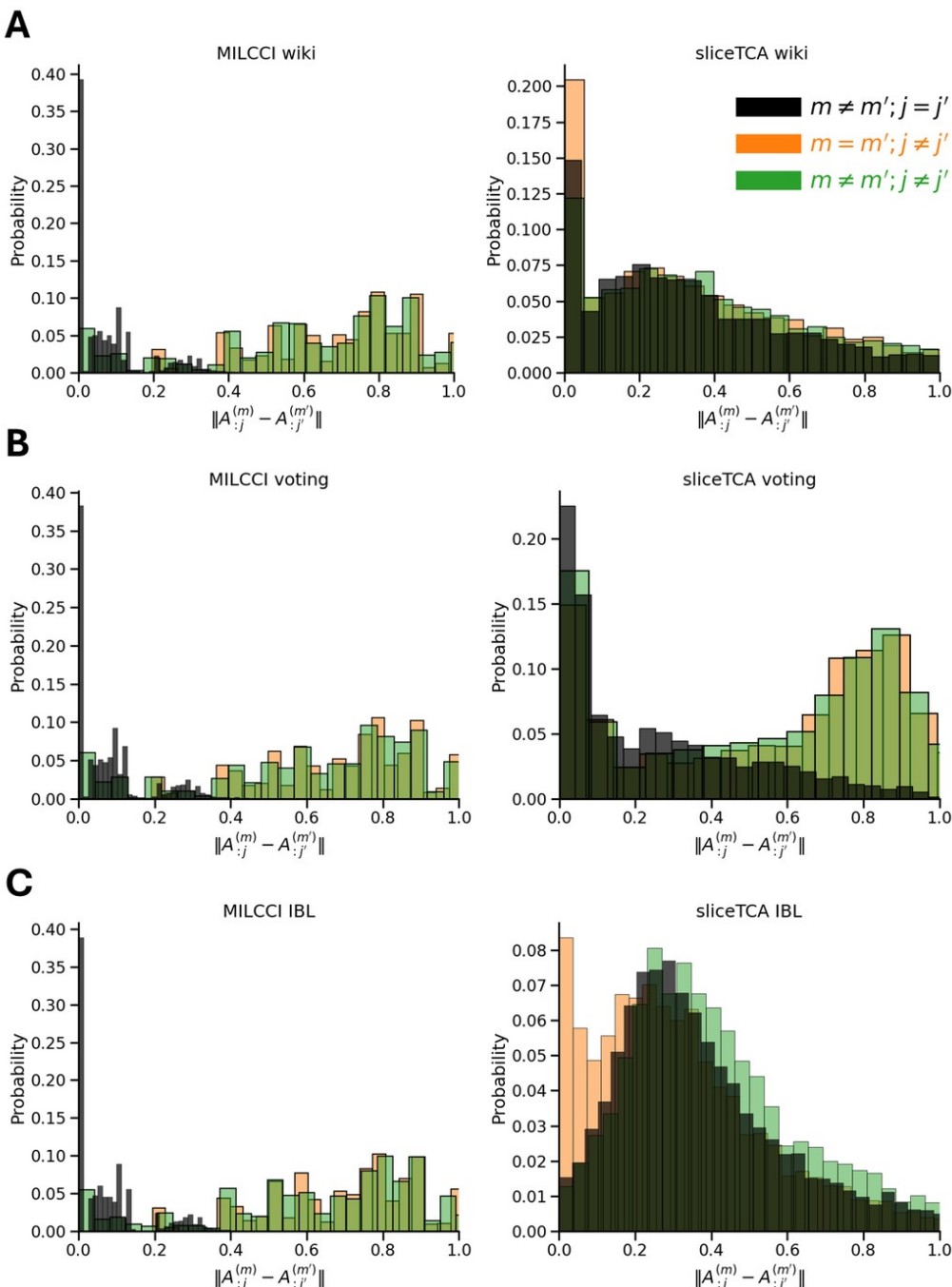

*Figure 34.* **Distribution of distances between component pairs.** Comparison of distances between same components (black) versus different component pairs across three datasets and two methods, presented for sliceTCA configuration 2 that allows components change (Sec. H.2). **A** Wiki dataset, **B** Voting dataset, **C** IBL dataset. Left column shows MILCCI results, right column shows sliceTCA results. Black bars represent distances between the same component across different conditions ($m \neq m'; j = j'$), orange bars represent distances between different components in the same condition ($m = m'; j \neq j'$), and green bars represent distances between different components in different conditions ($m \neq m'; j \neq j'$).

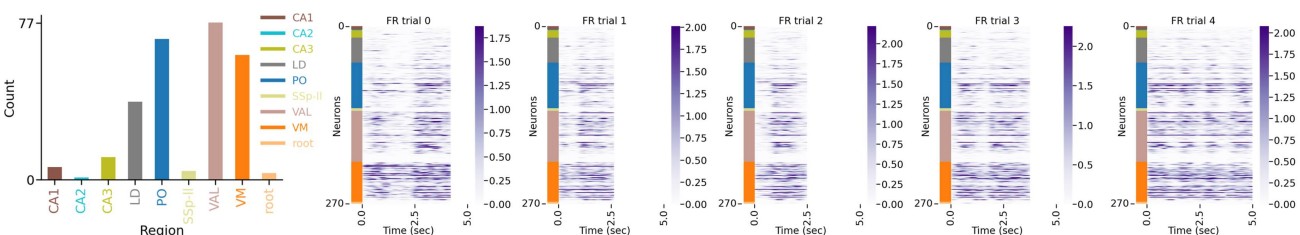

*Figure 35.* IBL data following our pre-processing steps.

$$\Phi_{:\mathcal{G}_2^{(b)}:}$$

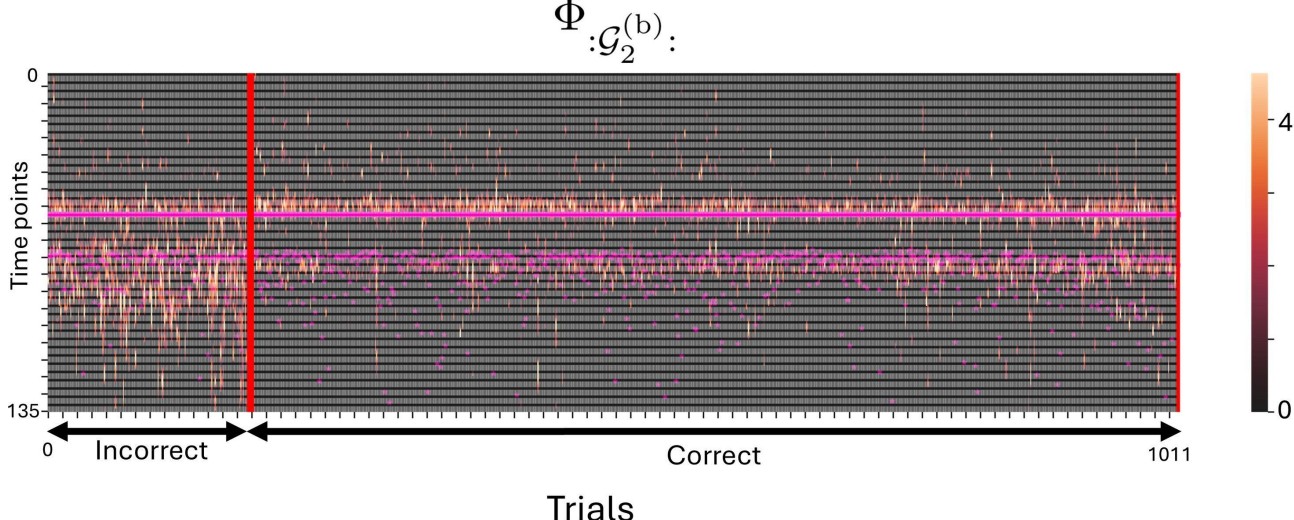

*Figure 36.* Traces of $\Phi_{:\mathcal{G}_2^{(b)}:}$ across trials, separated by decision correctness. Solid pink: stimulus on. Dashed pink: stimulus off. The stimulus appears for a median duration of 1.45 s across trials.

$$\Phi_{:\mathcal{G}_1^{(c)}:}^{(m)}$$

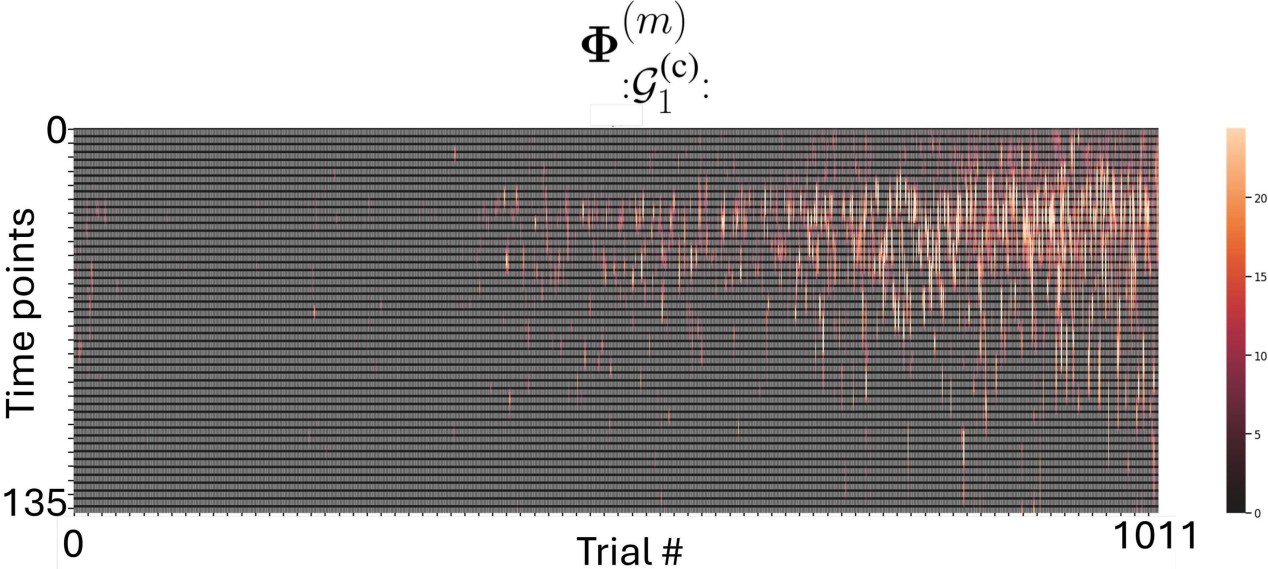

*Figure 37.* $\Phi_1^{(c)}$ presents an increasing temporal drift over trials.

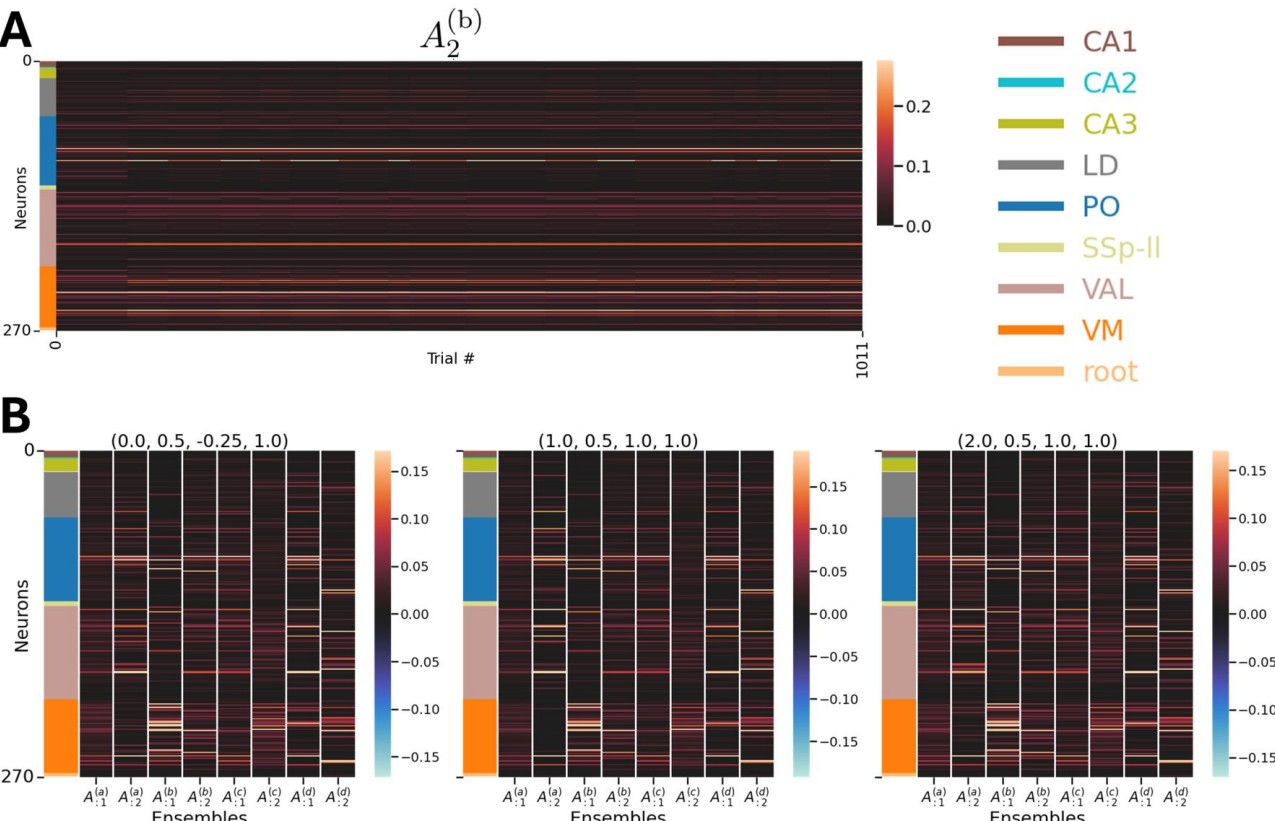

*Figure 38.* **Components identified by MILCCI in the IBL experiment. A:** Example ensemble (with its trace discussed in the main text) and its adjustments across trials. **B:** Example ensemble matrices reconstructed for three random trials. Each subplot shows all ensembles present in that trial under the unique set of labels indicated.

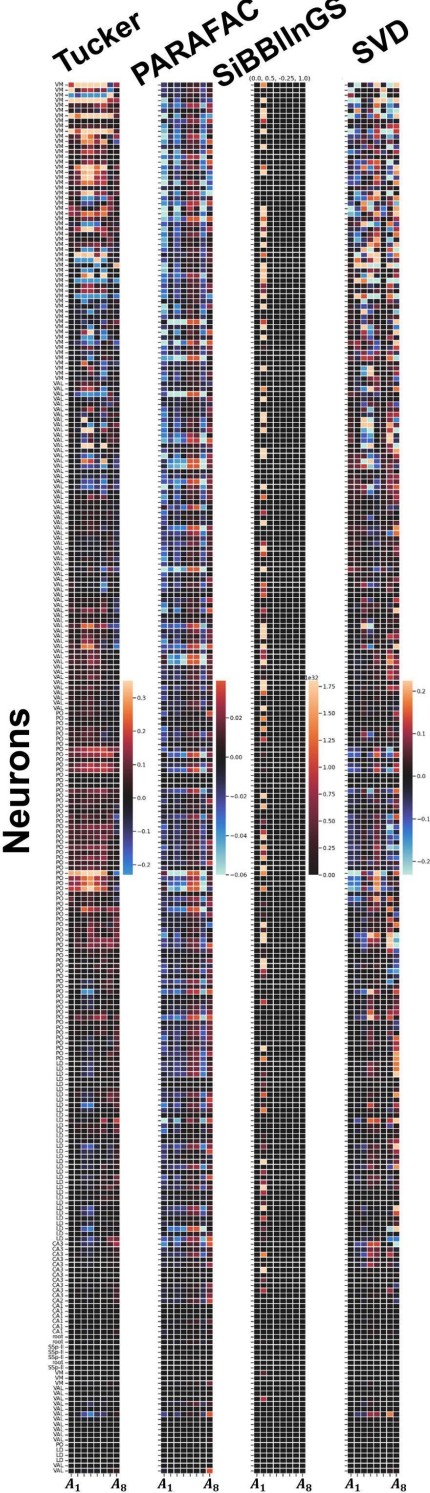

*Figure 39.* **IBL Neuronal Ensembles Components Identified by Baselines.**

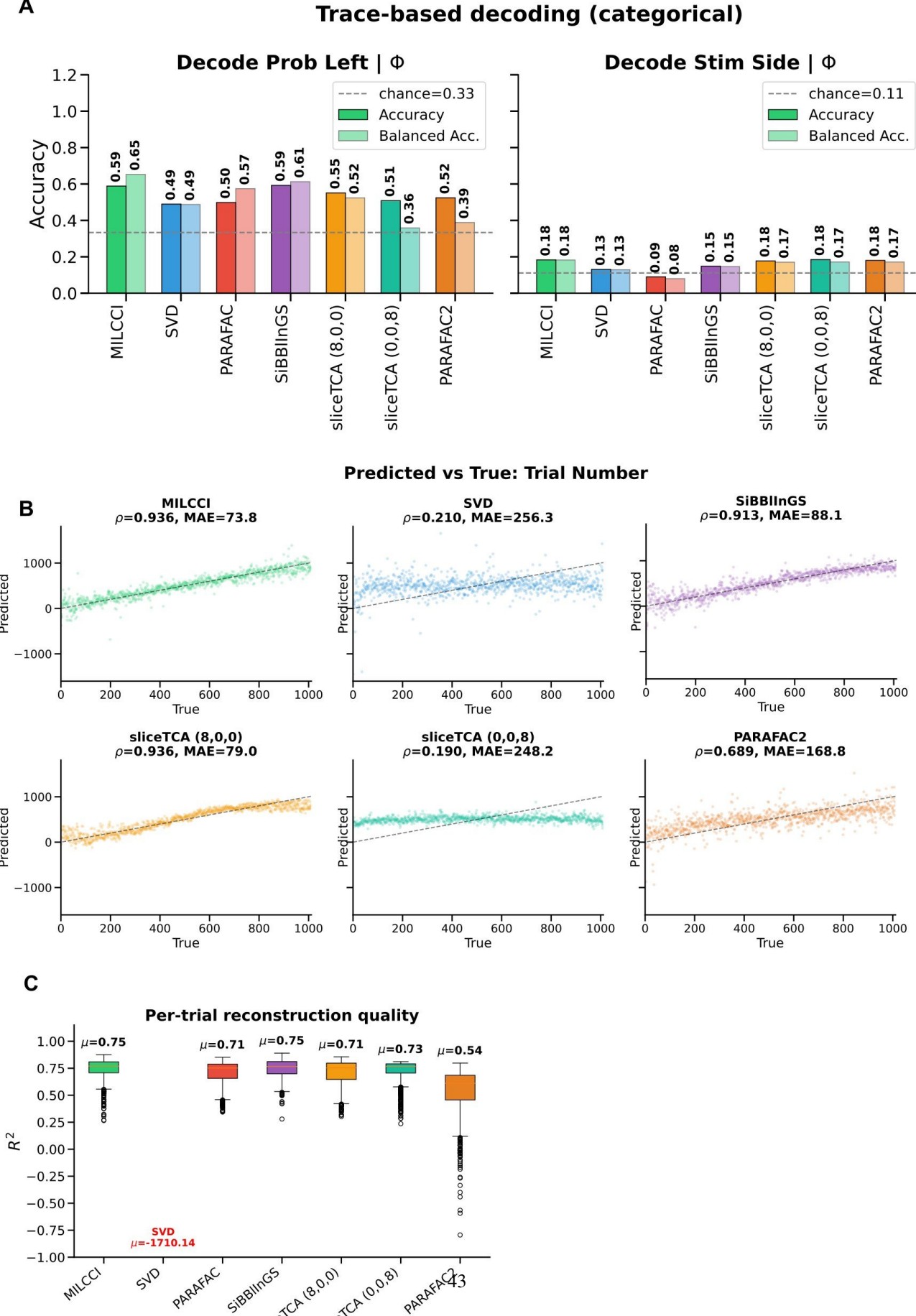

*Figure 40.* **Cross-validated decoding and reconstruction quality for the IBL neuroscience experiment. A:** Five-fold cross-validated

## G. Additional Information about Neuronal Ensembles Experiment

The IBL dataset is part of the International Brain Laboratory (IBL) effort to map neural activity underlying decision-making in mice across the whole brain. We accessed the IBL's data via the Dandi archive, in an NWB (Rübel et al., 2022; Laboratory et al., 2025) format.

The randomly selected session was recorded on February 11, 2020 in the Churchland Lab at CSHL (currently at UCLA). In the IBL task, mice view a grating stimulus on the left or right side of a screen (or no stimulus) and report its location by turning a wheel. The task includes block-wise priors, where one side is more likely than the other, requiring mice to combine sensory evidence with prior expectations. Stimulus contrast varies across trials to manipulate difficulty, enabling precise measurement of perceptual decision-making and neural correlates during electrophysiology recordings. Electrophysiological recordings were collected using Neuropixels probes from diverse brain areas. These recordings provide single-spike resolution activity during a decision-making task and include additional data such as sensory stimuli presented to the mouse, behavioral responses and response times. The subject (ID: CSHL052) was a female C57BL/6 mouse (Mus musculus), 6 months old and 22g at the time of recording. The full description of the session and task protocol is provided in (Angelaki et al., 2025).

## H. Information About Baseline Calculation and Execution

We compared MILCCI to matrix (SVD), tensor (PARAFAC, HOSVD, sliceTCA (Pellegrino et al., 2024), PARAFAC2 (Harshman, 1972)), and multi-array (SiBBlInGS (Mudrik et al., 2024)) decompositions.

### H.1. Comparison to SVD, Tucker, PARAFAC

We compared these methods to MILCCI both quantitatively and qualitatively. For the qualitative comparison, we provide example figures in the main text and appendix that emphasize MILCCI's superior performance. Notably, in all these methods the component matrix is fixed, as defined by the first mode; therefore, unlike MILCCI, they cannot (1) reveal structural adjustments over trials or disentangle category effects via the components, and (2) capture free trial-to-trial variability without tensor constraints (except for SVD). Consequently, their ability to capture such effects is inherently limited, though they represent the closest methods to MILCCI that we can reasonably compare to (in the sense that they provide comparable components and traces). Thus, the comparison is also limited in that we cannot show structural variability that these methods fundamentally do not support. Quantitatively, we calculated information criteria (AIC, BIC, HQC; lower values indicate better fit) using the degrees of freedom of each method. As seen in the appendix figures, these methods struggle to capture the data when constrained to the same dimensionality as MILCCI, which we attribute to (1) the need for small structural adjustments, and (2) their inability to capture free trial-to-trial variability.

**See below running details for these methods:**

- **SVD**: We used NumPy's 'linalg.svd' package, using the same number of components as in MILCCI for each experiment. The SVD was applied to the data from all trials concatenated horizontally.

  For experiments containing missing values (e.g., the voting experiment), NaNs were filled with zeros. Components were extracted from the left singular vectors ($U$), and the corresponding traces were obtained by multiplying the singular values matrix ($\Sigma$) with the right singular vectors ($V^\top$).

- **PARAFAC** (Harshman et al., 1970): We used the PyLops (Ravasi & Vasconcelos, 2020) PARAFAC implementation, with the same rank and number of components as MILCCI. For experiments with trials of varying durations (e.g., the IBL), we used the 90-th percentile trial length to prevent outliers from dominating, stacking trials along a third dimension and zero-padding shorter trials.

  Components ($\mathcal{A}$) were extracted from the first tensor mode (first factor), and traces were obtained by multiplying the second mode and the third mode according to the trial and component count.

- **Tucker** (Tucker, 1966) (HOSVD): Also for Tucker, we used the PyLops (Ravasi & Vasconcelos, 2020) PARAFAC implementation, with the same dimensions and number of components as MILCCI. For the 3-rd mode, we used the minimum between the number of time points and the number of trials. Again, for experiments with trials of varying durations (e.g., the IBL), we used the 90-th percentile trial length to prevent outliers from dominating, stacking trials along a third dimension and zero-padding shorter trials.

Components ($\mathcal{A}$) were extracted from the first tensor mode (first factor), and traces were obtained by multiplying the second mode, the core matrix, and the trial and component count. Notably, the component matrix in these methods is fixed, as defined by the core tensor, and therefore cannot adjust over time or disentangle label variability via the components.

For some datasets (e.g., Wikipedia), PARAFAC and Tucker (PyLops (Ravasi & Vasconcelos, 2020) implementation) could not converge at the same MILCCI dimensionality ($p = 12$), even with SVD initialization, various normalization schemes, high tolerance (1e-2), $\ell_2 2$ regularization, and maximum iterations of 10,000, due to least-squares optimization instability. These errors often occur due to (1) many missing values (though no NaNs exist in our data), or (2) high-resolution (daily) measurements introducing highly correlated (Fig. 29) or nearly linearly dependent structures in the data. Hence, for the Wikipedia comparison, in addition to comparisons to SVD, SiBBlInGS, and sliceTCA, we tested Tucker and PARAFAC under lower ranks with increasing added noise. We found that these models converge at rank 9 with added i.i.d. Gaussian noise ($\sigma = 0.1$ for Tucker and $\sigma = 0.2$ for PARAFAC), and the results presented here are under these conditions.

## H.2. Comparison to SliceTCA

We note that a direct comparison between our method and SliceTCA (Pellegrino et al., 2024) is limited since SliceTCA is not tailored to find subtle label-driven supervised reorganization patterns in neural ensembles. SliceTCA performs an unsupervised decomposition $\hat{\boldsymbol{X}}_{n,t,k} = \sum_{r=1}^{R_{\text{neuron}}} u_n^{(r)} A_{t,k}^{(r)} + \sum_{r=1}^{R_{\text{time}}} v_t^{(r)} B_{n,k}^{(r)} + \sum_{r=1}^{R_{\text{trial}}} w_k^{(r)} C_{n,t}^{(r)}$ that finds $R$ components. In their notation, $\boldsymbol{u}^{(r)}$ (neural loading vector) is equivalent to one column of our $\mathcal{A}$ (i.e., $\mathcal{A}_{:,j}$), while their slice $\boldsymbol{A}^{(r)}$ (time-by-trial matrix) would correspond to our $\boldsymbol{\Phi}$ for all trials. Similarly, $\boldsymbol{v}^{(r)}$ (time loading vector) combined with $\boldsymbol{B}^{(r)}$ (neuron-by-trial slice) provides an alternative decomposition. SliceTCA's $R_{\text{neuron}}$ corresponds to our $P$.

We compared MILCCI to SliceTCA (Pellegrino et al., 2024) with $R_{\text{neuron}} = P$ using their publicly available implementation at https://github.com/arthur-pe/slicetca. We followed the parameters outlined in their Google Colab notebook: positive=True, learning_rate=$5 \times 10^{-3}$, min_std=$10^{-5}$, max_iter=1,000 for Wiki and Synth, 5,000 for IBL, and seed=0.

We tested two separate configurations as baselines:

1. **Configuration (1):**
   considers one neural component (e.g., one ensemble) that varies its traces over trials without additions, which is the closest to MILCCI in terms of formulation, but not enabling small changes in ensembles.

2. **Configuration (2):**
   uses a matrix of neurons $\times$ trials (i.e., captures how neurons can change over trials) via the matrix $\boldsymbol{B}$, however each matrix of neurons by trial has one trace. This captures mainly the ensemble adjustment to trial in an unconstrained way (unlike MILCCI) and enables more flexibility in that, but on the other hand restricts the traces more. Any other combination of ranks would not be interpretable in terms of comparison to MILCCI.

We extracted MILCCI-equivalent $\{\mathcal{A}\}$ and $\boldsymbol{\Phi}$ as follows:

1. **Configuration (1)**:
   The $\boldsymbol{u}^{(r)}$ vectors form the ensemble matrix (fixed across trials), while temporal traces are extracted from the $\boldsymbol{A}^{(r)}$ matrices by breaking to trials.

2. **Configuration (2)**:
   The ensembles are captured by the rows of $\boldsymbol{B}^{(r)}$ and their variation over trials by the columns. Traces are given by $\boldsymbol{v}^{(r)}$.

## H.3. Comparison to PARAFAC2

We also compared MILCCI to PARAFAC2 (Harshman, 1972), which allows one mode to vary across slices and can therefore handle variable-length trials. However, PARAFAC2 does not incorporate label information, does not enforce sparsity on components, and does not disentangle category-specific effects. We ran PARAFAC2 across all main experiments using the

`tensorly` implementation. As shown in Figs. 8, 31, and 19, PARAFAC2 exhibits high cross-component correlations and achieves lower reconstruction quality and information criteria values compared to MILCCI. We also included PARAFAC2 in the cross-validated decoding analyses (Figs. 32 and 40).

### H.4. SiBBlInGS

:

MILCCI's main advantage over SiBBlInGS (Mudrik et al., 2024) is interpretability for multi-way, multi-label data. Particularly, SiBBlInGS cannot disentangle the effects of co- or separately-varying labels, making it difficult to understand their individual contributions. SiBBlInGS, by modeling each unique label tuple as a distinct label would further increase tensor size and computational complexity. This is especially pronounced in experiments where some label categories vary across many unique values (e.g., IBL trial number with over 1000 values). In contrast, MILCCI can handle all of these without requiring additional dimensions, and can also account for the ordinal nature of each category, whereas SiBBlInGS requires choosing a single sorting across all categories. While we acknowledge quantitative metrics would also be valuable against SiBBlInGS, SiBBlInGS' graph-driven sparsity makes standard model comparison metrics (AIC, BIC) irrelevant. Particularly, SiBBlInGS inference includes a graph-based reweighting sparsity mechanism that hinders accurate estimation of degrees-of-freedom needed for information criteria calculations, making information criteria comparison intractable. Hence, we limited quantitative comparisons against SiBBlInGS to synthetic data (Fig. 2) and, for the real-world experiments, we focused on qualitative comparisons that emphasize MILCCI's interpretability advantages.

# I. Alternative Inference of traces via Dynamic Prior

In the main text, we regularize the temporal traces $\mathbf{\Phi}^{(m)}$ using a smoothness penalty. Here, we present an alternative formulation where the temporal traces evolve according to a Linear Dynamical System (LDS), suitable for data with non-stationary dynamics.

### I.1. Linear Dynamical System Prior

We assume that for each trial $m$, the temporal traces follow:

$$\phi_t^{(m)} = \mathbf{W}^{(m)}\phi_{t-1}^{(m)} + \boldsymbol{\eta}_t, \quad t = 2, \ldots, T^{(m)} \tag{4}$$

where $\mathbf{W}^{(m)} \in \mathbb{R}^{P \times P}$ is a trial-specific transition matrix and $\boldsymbol{\eta}_t \sim \mathcal{N}(\mathbf{0}, \sigma^2 \mathbf{I})$.

### I.2. Modified Objective and Inference

We modify the optimization to jointly learn traces and dynamics. The algorithm alternates between:

**Step 1: Update $\mathcal{A}^{(k)}$ (for each category $k$)**

Same as main text, using Equation 2.

**Step 2: Update $\{\mathbf{\Phi}^{(m)}, \mathbf{W}^{(m)}\}$ (for each trial $m$)**

Given a fixed loading matrix for a general trial $m$, $\mathbf{A}^{(L^{(m)})}$, we perform inner iterations (3-5 times):

**Step 2a: Update $\mathbf{\Phi}^{(m)}$ given $\mathbf{W}^{(m)}$**

$$\hat{\mathbf{\Phi}}^{(m)} = \arg\min_{\mathbf{\Phi}^{(m)}} \left\| \mathbf{Y}^{(m)} - \mathbf{A}^{(L^{(m)})}\mathbf{\Phi}^{(m)} \right\|_F^2 + \gamma_3 \sum_{t=2}^{T^{(m)}} \left\| \phi_t^{(m)} - \mathbf{W}^{(m)}\phi_{t-1}^{(m)} \right\|_2^2 + \gamma_4 \left\| (\mathbf{C} \odot (\mathbf{1} - \mathbf{I}_P)) \odot \mathbf{D} \right\|_{1,1} \tag{5}$$

**Step 2b: Update $\mathbf{W}^{(m)}$ given $\mathbf{\Phi}^{(m)}$**

With regularization $R(\boldsymbol{W}^{(m)}) = \|\boldsymbol{W}^{(m)} - \boldsymbol{I}\|_F^2$ to encourage stability:

$$\hat{\boldsymbol{W}}^{(m)} = \left( \sum_{t=2}^{T^{(m)}} \phi_t^{(m)} (\phi_{t-1}^{(m)})^T + \gamma_5 \boldsymbol{I} \right) \left( \sum_{t=2}^{T^{(m)}} \phi_{t-1}^{(m)} (\phi_{t-1}^{(m)})^T + \gamma_5 \boldsymbol{I} \right)^{-1} \tag{6}$$

The inner iterations stabilize both $\boldsymbol{\Phi}^{(m)}$ and $\boldsymbol{W}^{(m)}$ before updating $\mathcal{A}$, preventing noise amplification.

### I.3. Initialization

1. Initialize $\boldsymbol{\Phi}^{(m)}$ using the original smoothness-based objective (Equation 3)

2. Initialize $\boldsymbol{W}^{(m)} = \left( \sum_t \phi_{t,\text{init}}^{(m)} (\phi_{t-1,\text{init}}^{(m)})^T \right) \left( \sum_t \phi_{t-1,\text{init}}^{(m)} (\phi_{t-1,\text{init}}^{(m)})^T \right)^{-1}$

This dynamic prior is suitable for cases that are assumed to be stationary and governed by a single LDS that does not change over time. Notably, real-world data is often non-stationary, and hence this dynamic prior would benefit from extensions in future work, such as learning dynamics like those exemplified in (Chen et al., 2024; Mudrik et al., 2025)).

*Table 4.* Timing results (in seconds) for different methods across the three main experiments. Dash (-): method did not converge for the same dimension.

| Experiment | SVD | Tucker | parafac | parafac_scaled | SliceTCA | SiBBlInGS | MILCCI |
|---|---|---|---|---|---|---|---|
| Voting | 20.01 | 20.12 | 20.26 | 20.35 | 27.58 | 27.19 | 28.16 |
| Wiki | 20.37 | - | - | - | 34.05 | 29.48 | 31.12 |
| IBL | 24.03 | 268.94 | 1056.20 | 1256.74 | 1000 | 1004.32 | 1000.32 |

## J. Fourth real-world experiment: MILCCI identifies neural ensembles that adjust to arousal level and stimulation frequency and evolve via dynamical rules

### J.1. Data and Pre-processing

We used one experimental session from (Papadopoulos et al., 2024) in NWB format from the DANDI archive (Rübel et al., 2022) (DANDI:000986), which the McCormick laboratory at the University of Oregon collected. The dataset contains recordings from mouse auditory cortex during passive exposure to auditory stimuli. The subject was a male mouse (Mus musculus), aged P79D from birth (see experimental illustration taken from (Papadopoulos et al., 2024) in Fig. 41A).

We processed spike trains from the NWB file (raster plot in Figure 41E) to estimate firing rates (FR) using Gaussian convolution with the following parameters: 5 ms time bins, 50 ms Gaussian window, and 5 ms $\sigma$ (Fig. 41F). For this demonstration, we used the first 500 trials. The processed dataset thus contains 235 neurons across 15 unique experimental conditions that varied throughout the first 500 trials.

We leverage this example to demonstrate MILCCI's robustness and capacity for extensions, including: 1) pre-processing via non-linearity, 2) modifying the inference of $\Phi$ to evolve via dynamical priors, rather than via Eq. 3 (App. I for details), and 3) robustness to hyperparameters (App. J.3).

### J.2. Demonstration of Extended MILCCI With Non-Linear Transformation and Dynamics Prior Over Traces Evolution

This experiment provides a glimpse of MILCCI's extensibility by showcasing two key advances: (1) dynamical evolution of neural ensembles, and (2) nonlinear transformation of FR via tanh (normalizing to range [-1, 1]). This serves as a proof of concept for modeling dynamics alongside MILCCI, paving the way for future extensions with more complex non-stationary dynamics (see App. I for details on dynamics inference).

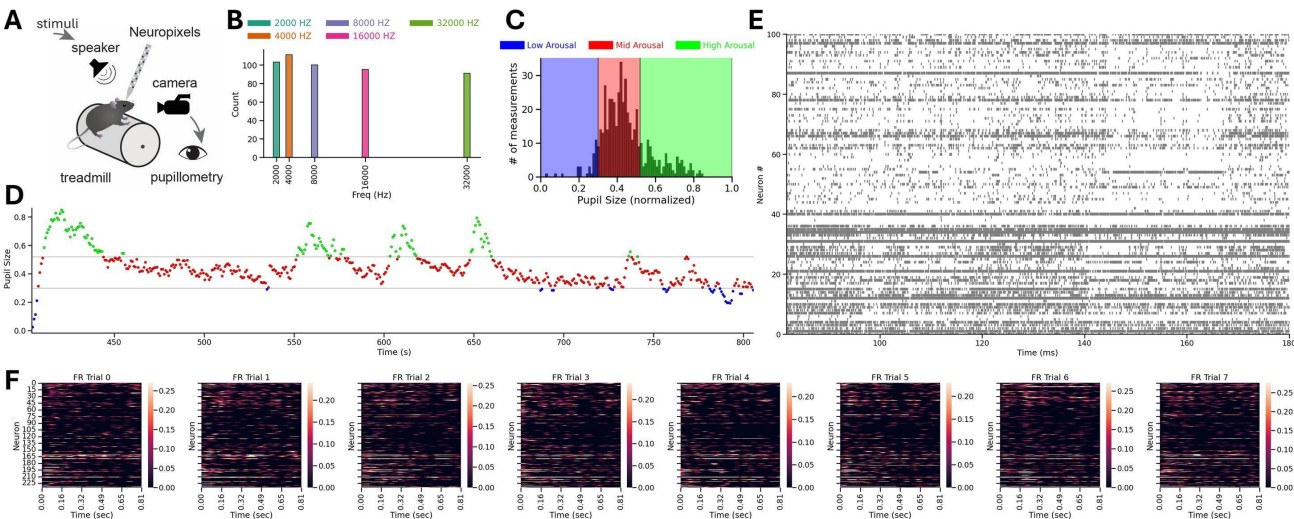

*Figure 41.* **Additional demonstration of MILCCI on data from (Papadopoulos et al., 2024).** **A:** Experiment illustration, adapted from (Papadopoulos et al., 2024). **B:** Trial counts per stimulation frequency among the first 500 trials analyzed. **C:** Distribution of normalized pupil size (a.u.) across the first 500 trials. Colored background represents the three arousal states considered. (cutoffs 0.3, 0.52) **D:** Average pupil size per trial. Each marker represents one trial; x-axis shows trial midpoint time from recording start. **E:** Raster plot of the first 180 ms of neural activity. **F:** Example firing rate estimation for the first 8 trials from spike train data.

We defined the following categories: (a) arousal level (based on binarized pupil diameter, Fig. 41C); and (b) stimulation frequency (distribution among the used trials in Fig. 41B).

We ran extended MILCCI with $p_j = 3$ ensembles per category.

Here, MILCCI reveals interpretable structure across multiple experimental variables (Fig. 42). The learned transition networks $\boldsymbol{W}^{(m)}$ (Fig. 42A) show condition-dependent ensemble interactions, with varying positive and negative coupling strengths across arousal levels and stimulation frequencies. The heatmap of all transition matrices across trials (Fig. 42B) reveals systematic organization aligned with both arousal and frequency conditions. The temporal traces (Fig. 42C) capture joint structure: when trials are sorted by arousal (left color bar), $\boldsymbol{\Phi}_2$ and $\boldsymbol{\Phi}_3$ show distinct activation patterns that align with arousal levels, while the stimulation frequency (top color bar) reveals additional frequency-specific structure within arousal groups. This is further supported by the averaged traces (Fig. 42D), where conditioning on specific frequencies (4, 8, 32 kHz) reveals distinct trace shapes that vary with arousal level. The identified ensembles (Fig. 42F) show subtle compositional adjustments to both arousal (top) and stimulus frequency (bottom), demonstrating MILCCI's ability to capture category-specific structural effects. The model achieves low reconstruction error across trials (Fig. 42E; mean relative MSE $\approx 0.07$).

### J.3. Hyperparameter Sensitivity Analysis

To empirically demonstrate MILCCI's robustness to hyperparameter choices, we conducted a comprehensive sensitivity analysis across a wide range of values. We tested 20 values for $\gamma_1 \in [0.002, 0.5]$ and 40 values for $\gamma_2 \in [0.002, 0.5]$, yielding 800 total hyperparameter combinations across multiple ensemble instances.

To quantify the differences of learned components $\mathcal{A}$ between iterations with different parameters vs. same parameters, we employed the normalized Frobenius distance:

$$d_{\mathrm{norm}}(\widetilde{\mathcal{A}}^{(1)}, \widetilde{\mathcal{A}}^{(2)}) := \frac{\|\widetilde{\mathcal{A}}^{(1)} - \widetilde{\mathcal{A}}^{(2)}\|_F}{\sqrt{\|\widetilde{\mathcal{A}}^{(1)}\|_F^2 + \|\widetilde{\mathcal{A}}^{(2)}\|_F^2}} \tag{7}$$

where $\widetilde{\mathcal{A}}^{(1)}$ and $\widetilde{\mathcal{A}}^{(2)}$ are two general components for a pair of iterations. This metric enables us to compare the degree of change of the same ensemble under hyperparameter change to the degree of difference between 2 distinct ensembles. If a single ensemble remains more consistent under hyperparameter change compared to cross-ensemble differences, that means that the degree of hyperparameter sensitivity is slight, which suggests MILCCI's robustness.

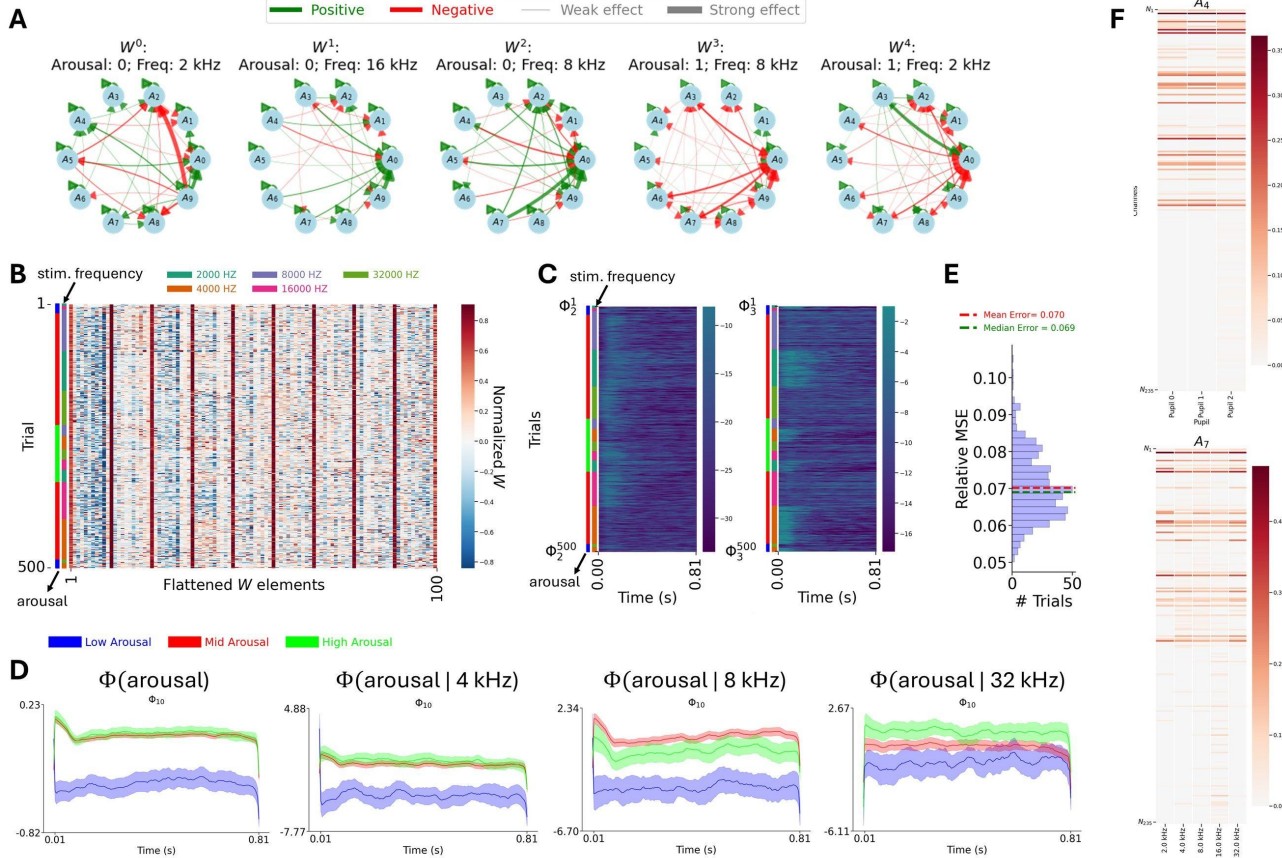

*Figure 42.* **Demonstrating MILCCI with nonlinear tanh data transformation** $(\tanh{(\boldsymbol{y}^{(m)})} \approx \sum_{(\mathbf{k})\in C} \mathcal{A}^{(\mathbf{k})}_{::L_{\mathbf{k}}^{(m)}} \boldsymbol{\Phi}^{(m)}_{\mathcal{G}^{(\mathbf{k})}:})$ **and dynamical constraints over the trace evolution** $(\boldsymbol{\Phi}_t^{(m)} \approx \boldsymbol{W}^{(m)} \boldsymbol{\Phi}_{t-1}^{(m)})$ **, via recent neural data from (Papadopoulos et al., 2024). (A)** Transition networks $\boldsymbol{W}^{(m)}$ for five example trials from different arousal and stimulus frequency conditions, showing learned interactions between ensembles. **(B)** Heatmap of all learned transition matrices (each $\boldsymbol{W}^{(m)}$ flattened) across trials (rows). Scatters on the left indicate trial conditions: arousal level (left side) and stimulation frequency (right side). Weights are normalized by column maximum. **(C)** Temporal traces for two example ensembles ($\boldsymbol{\Phi}^1$ and $\boldsymbol{\Phi}^3$) across trials and time, organized by arousal level, demonstrating condition-dependent temporal dynamics. Scatters on the left are the same as in panel **B**. **(D)** Average temporal traces with stderr grouped by arousal level: without frequency conditioning (left), or conditioned on specific stimulus frequencies (three rightmost panels: 4 kHz, 8 kHz, and 32 kHz). **(E)** Distribution of reconstruction error (relative MSE) across trials. Dashed lines indicate mean and median. **(F)** Example neuronal ensembles and how they subtly adjust to labels, showing condition-specific loadings across channels. Top panels show components varying with arousal level; bottom panels show components adjusting to stimulus frequency.

We thereby computed two types of distances to assess hyperparameter sensitivity relative to ensemble variability (Fig. 43B,C,F):

- **Within-ensemble distances**: For each ensemble, we computed pairwise distances between component matrices obtained with different hyperparameter settings. These distances quantify how much the learned components vary due to hyperparameter choices while holding the data fixed.

- **Cross-ensemble distances**: For each hyperparameter setting, we computed pairwise distances between component matrices from different ensemble instances. These distances quantify how much the learned components vary due to data sampling and stochastic initialization while holding hyperparameters fixed.

As seen in Fig. 43, B-F, the within-ensemble distances are substantially smaller than cross-ensemble distances. This would indicate that the learned components are more sensitive to the underlying data structure than to hyperparameter tuning, demonstrating that the model reliably captures meaningful patterns across reasonable hyperparameter ranges.

Interestingly, the component matrices (Fig. 43A) show that higher $\gamma_1$ values produce sparser components as expected, while the overall structure remains stable. Within-ensemble distances are consistently lower than cross-ensemble distances, with a mean ratio of 0.17 (Fig. 43C), indicating that hyperparameter variation introduces less variability than ensemble stochasticity. The distribution of distances (Fig. 43D) reveals separation between the two types of variation. Per-ensemble analyses (Fig. 43B,E) show this pattern holds across all individual ensemble instances, with ratios below 1. These results demonstrate model robustness, as hyperparameter variation contributes minimally compared to stochastic ensemble variation. We additionally evaluated sensitivity to random initialization across 100 seeds (Fig. 44), to $\gamma_3$ and $\gamma_4$ across five orders of magnitude (Fig. 45), and to the number of components $P$ from 2 to 30 (Fig. 46). All analyses confirm that the learned components remain stable.

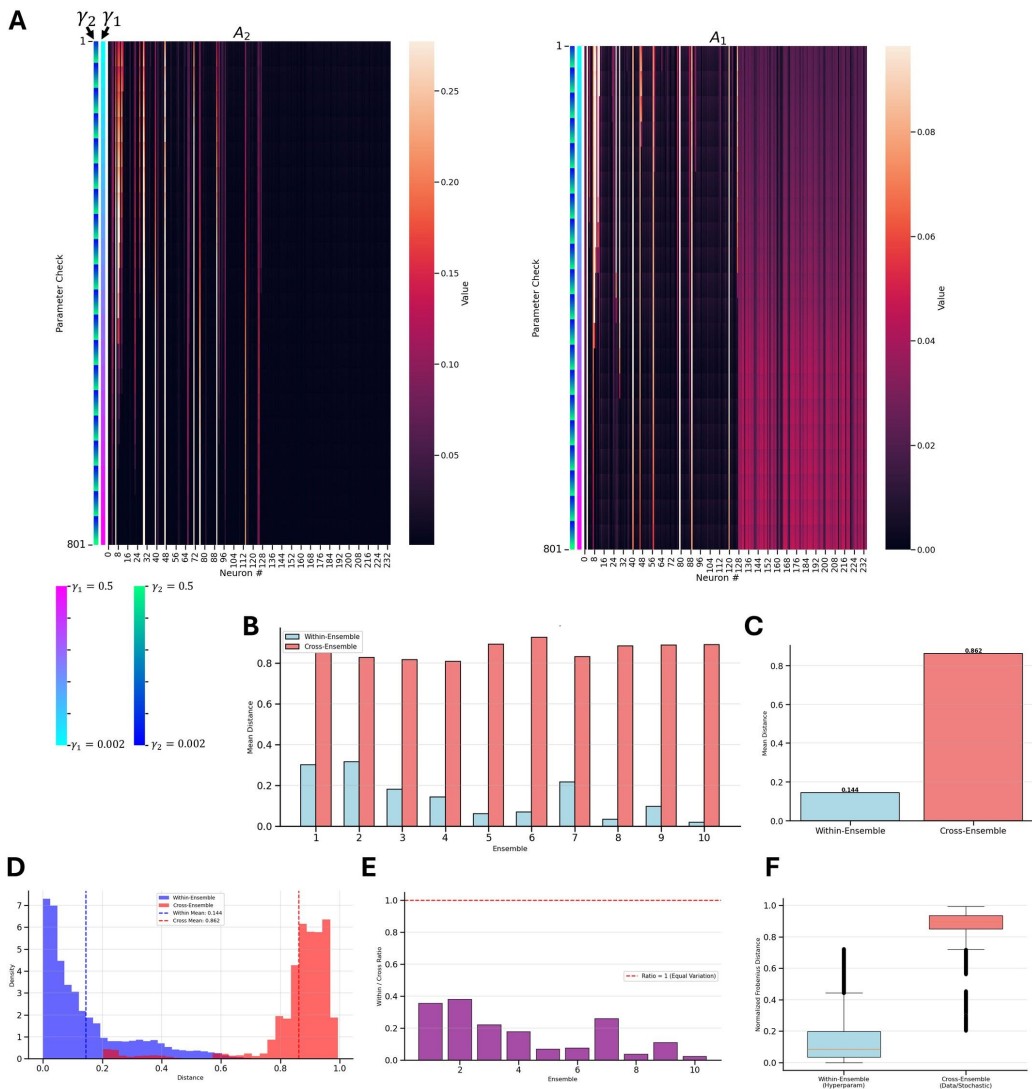

*Figure 43.* **Hyperparameter sensitivity analysis across 800 parameter combinations (App. J.3). A:** Example component matrices from two ensembles across all hyperparameter settings, ordered by $\gamma_1$ and $\gamma_2$ (colorbars at the bottom). For example, higher $\gamma_1$ values yield sparser components. **B:** Per-ensemble comparison of within-ensemble (blue) versus cross-ensemble (red) distances. **C:** Overall mean distances showing within-ensemble variation is smaller than cross-ensemble variation. **D:** Distribution of normalized Frobenius distances reveals clear separation between within-ensemble and cross-ensemble variation. **E:** Robustness ratios for each ensemble, all below 1 (dashed line indicates equal variation). **F:** Overall boxplot comparison confirms systematic difference between within-ensemble and cross-ensemble distances across all 10 ensembles.

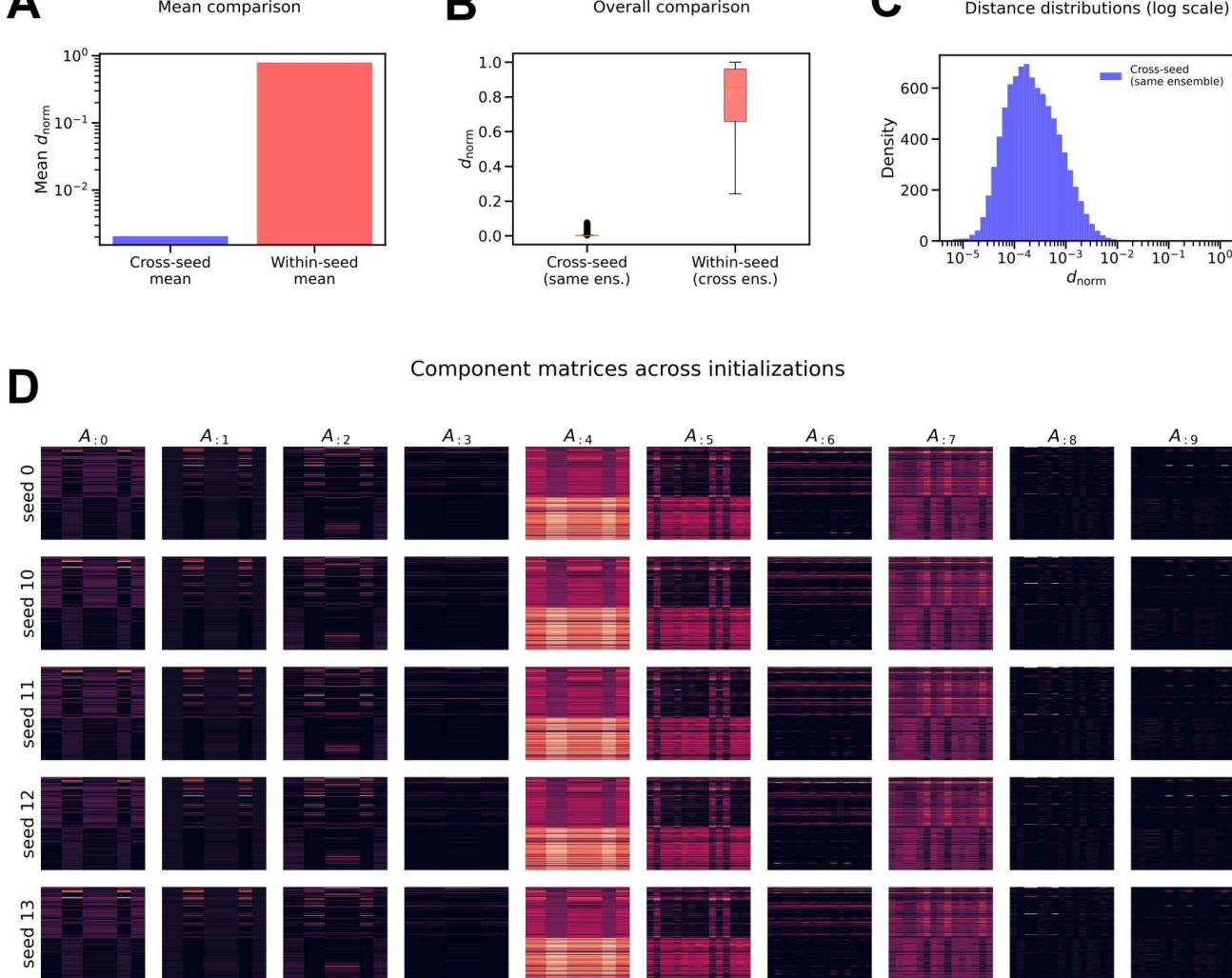

*Figure 44.* **Initialization sensitivity. A:** Mean $d_{\mathrm{norm}}$ for cross-seed (same ensemble, different initialization) vs. within-seed (different ensembles, same initialization) comparisons across 100 random seeds, shown on log scale. Cross-seed variation is smaller than within-seed variation. **B:** Boxplot comparison of the two distributions. **C:** Distribution of cross-seed $d_{\mathrm{norm}}$ values (log scale), centered around $10^{-4}$. **D:** Component matrices $A$ for five example seeds, showing visually identical structure across initializations.

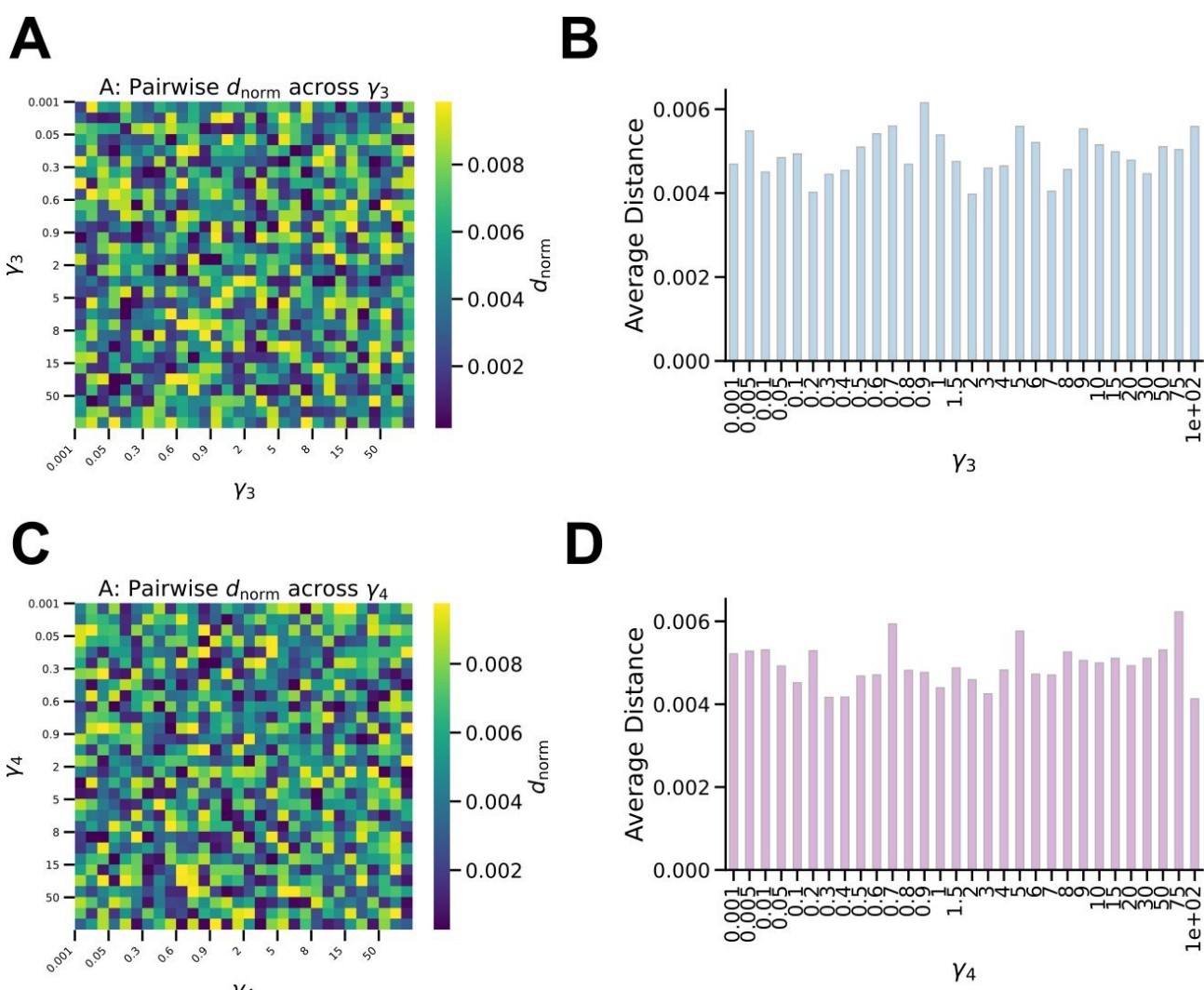

*Figure 45.* **Sensitivity to $\gamma_3$ (temporal smoothness) and $\gamma_4$ (trace decorrelation). A:** Pairwise $d_{\text{norm}}$ heatmap across 30 values of $\gamma_3 \in [0.001, 100]$, showing uniformly low distances ($< 0.01$). **B:** Average $d_{\text{norm}}$ per $\gamma_3$ value, confirming flat profile. **C:** Pairwise $d_{\text{norm}}$ heatmap across 30 values of $\gamma_4 \in [0.001, 100]$, similarly showing uniformly low distances. **D:** Average $d_{\text{norm}}$ per $\gamma_4$ value, confirming robustness. The learned component matrices are largely invariant to both hyperparameters across five orders of magnitude.

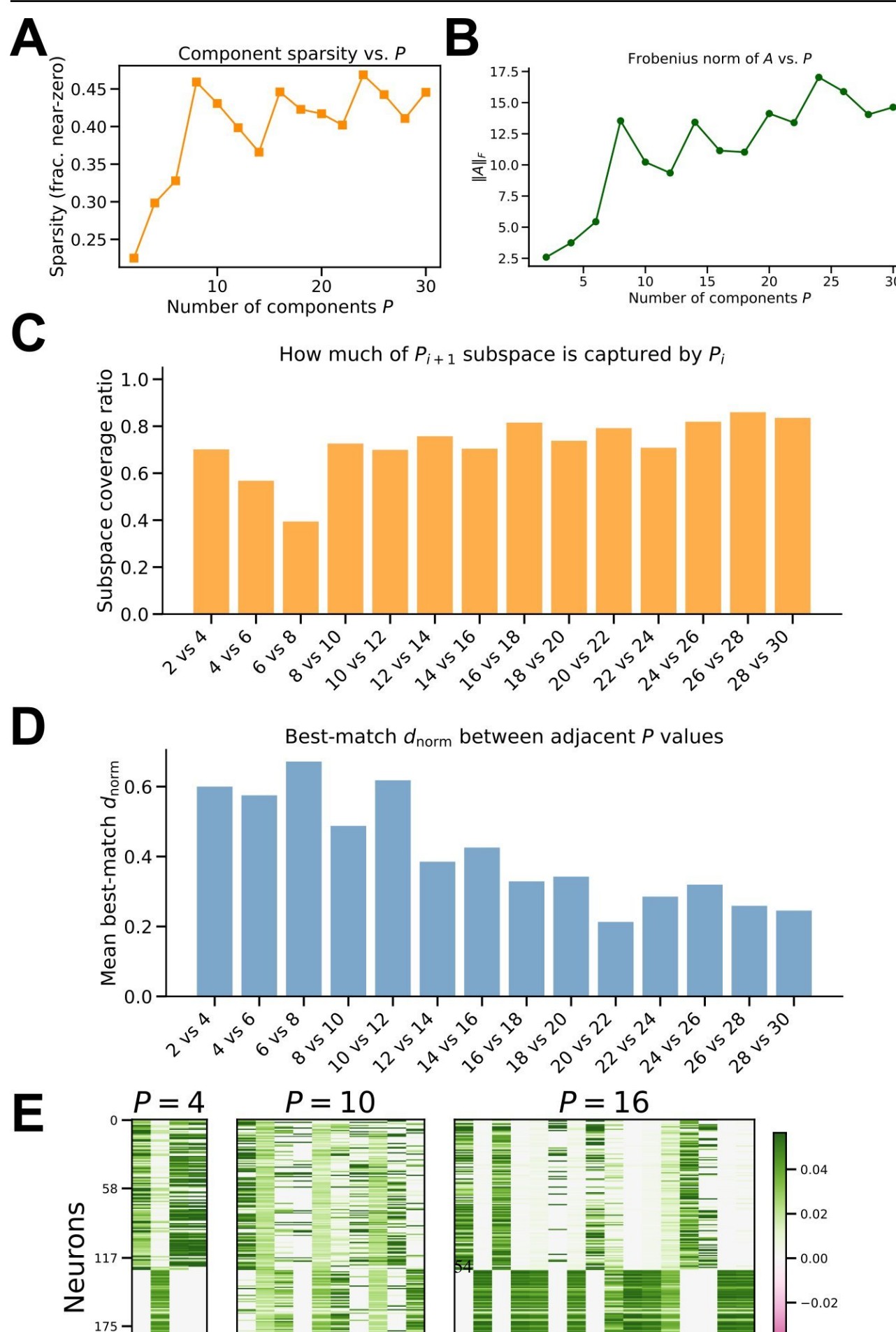

# K. Ethics Statement and LLM Usage

Our work does not raise any ethical concerns. Large language models were used only at the word or sentence level during manuscript writing to improve the language and catch grammar mistakes, with no influence on the scientific content or analysis.

