# OpenReview forum: "Multi-Integration of Labels Across Categories for Component Identification in Multi-trial Time Series"
_ICML.cc/2026/Conference — ICML 2026 regular_

### Official Review · Reviewer_UDpu · 2026-02-23

**Soundness:** 2
**Presentation:** 3
**Significance:** 3
**Originality:** 3
**Overall Recommendation:** 4
**Confidence:** 4

**Summary:**

MILCCI proposes a decomposition method for multi-trial, multi-label time-series data. The core approach constructs a 3D tensor representation with tunable sparse components for each label category, alternating between optimizing component tensors (via LASSO) and temporal traces (with smoothness and decorrelation constraints), while using label similarity graphs to regularize cross-label consistency of component variants. The method is validated on synthetic data and three real-world datasets.

**Compliance With Llm Reviewing Policy:**

Affirmed.

**Final Justification:**

The author has addressed most of the issues and added the key comparisons. The proposed method is somewhat novel, especially for the solution of new question.

I have raised my final score. However, as the reviewer yaXh pointed out, this paper may not be very suitable for conferences like ICML. If it is a more specialized conference on computational neuroscience or even a journal article, I would appreciate because of its wide influence and audience.

**Key Questions For Authors:**

1. Can you provide empirical convergence curves and initialization sensitivity analysis?

2. Why were SliceTCA and PARAFAC2 excluded from the baseline comparisons?

**Limitations:**

The paper should explicitly discuss several application boundaries.

1. The single-category constraint limits applicability to settings without category interactions.

2. The linearity assumption underlying the model may not hold for systems exhibiting complex nonlinear dynamics, and the paper does not characterize the regime where this assumption begins to break down.

3. Given the non-convex nature of the optimization and the absence of convergence guarantees, I think there is no assurance that different initializations or runs will yield consistent and reproducible results.

**Strengths And Weaknesses:**

**Strength**
1. The setting is meaningful.Typical PARAFAC/CPD and tensor ICA models are too rigid to capture individual differences expected in real neuroimaging applications. MILCCI's simultaneous support for multi-category label disentanglement, variable-length trials, and cross-label component fine-tuning addresses a genuine methodological gap.

2. The component fine-tuning mechanism offers a flexible modeling compromise. Rather than requiring components to be completely fixed or free across conditions, the framework allows structural adjustments through label similarity graphs. This design is conceptually appealing for settings where related conditions should share similar but not identical structure.

3. The experiments span diverse application domains. The validation covers political science, web analytics, and neuroscience. The hyperparameter sensitivity analysis covering 800 combinations demonstrates careful empirical investigation.

4. The framework demonstrates extensibility. Preliminary extensions including LDS dynamical priors and tanh nonlinearity transformations suggest the core framework can accommodate richer structural assumptions.

**Weakness**
1. There is no convergence guarantees in the paper. The alternating optimization is non-convex and known to suffer from local minima in related tensor decomposition methods. No convergence curves, iteration counts, or initialization sensitivity analysis are provided. More critically, the residual strategy that subtracts other categories' contributions introduces cascading dependencies, and I am concerned that error accumulation properties remain entirely uncharacterized.

2. iced that the most relevant baselines are missing. Two critical methods are omitted: (1) **SliceTCA** (Nature Neuroscience, 2024) directly addresses cross-trial covariability structure in neural data tensors; (2) **PARAFAC2** handles settings where one mode varies in size or structure across slices, making it applicable to variable-length trials. Without comparing to these established methods, the experimental contribution is significantly weakened.

3. The evaluation lacks systematic quantitative metrics. Real-data experiments rely on post-hoc qualitative narratives rather than rigorous evaluation. I would expect to see predictive evaluation (e.g., leave-one-year-out prediction), cross-validated reconstruction error, and standard neuroscience metrics (decoding accuracy, explained variance decomposition by category). The permutation tests only verify that components are non-random—a low bar that does not establish meaningful structure.

---

> ### Author Rebuttal · Authors · 2026-03-31
>
> We thank the reviewer for their review, as well as for recognizing that MILCCI addresses "a genuine methodological gap", that the "component fine-tuning mechanism offers a flexible modeling compromise", and that the experiments "span diverse application domains" with "careful empirical investigation". We address each concern below.
>
> **1\. SliceTCA and PARAFAC2 as baselines (W1, Q2)**
>
> We would like to respectfully correct the reviewer in that we did compare to sliceTCA extensively across all experiments, under two separate configurations (Sec. H.2). Specifically, sliceTCA results appear in Fig. 2G (synthetic), Fig. 7 (synthetic, detailed), Fig. 17 (voting), Fig. 28 (Wikipedia), Fig. 29 (IBL), and Fig. 30 (component distance analysis).
>
> Regarding PARAFAC2, we agree it is a relevant method. While PARAFAC2 handles variable-length data (a shared advantage with MILCCI), it lacks label-driven decomposition and ultimately shares most of the limitations of the PARAFAC model we benchmarked against. Specifically, PARAFAC2 allows one mode to vary freely, but it does not: (1) incorporate label information to understand the underlying compositional adjustments, (2) enforce sparsity on components for interpretability, and (3) disentangle category-specific effects. We will add this discussion of PARAFAC2 in the Related Work section.
>
> **2\. Convergence guarantees and initialization sensitivity (W2, Q1)**
>
> We thank the reviewer for raising this point. MILCCI uses the alternating optimization that is standard across matrix and tensor decomposition methods in the field. Each of MILCCI's subproblems is convex given the other variables fixed: the component update (Eq. 2\) is a LASSO problem and the trace update (Eq. 3\) is a regularized least-squares problem. The residual strategy of subtracting other categories' contributions is analogous to how dictionary learning methods separate contributions of different atoms. We note that convergence properties for this class of optimization have been studied theoretically, including local convergence guarantees for alternating minimization in dictionary learning and tensor factorizations (Chatterji et al., 2017, Tseng, 2001, Xu & Yin, 2013). We will clarify this in the paper.
>
> We appreciate the point about initialization sensitivity and will include it along with sensitivity to component counts in the revised manuscript, to complement our current parameter sensitivity analysis.
>
> *Chatterji, N., & Bartlett, P. L. (2017), NeurIPS; Tseng, P. (2001), Journal of optimization theory and applications; Xu, Y., & Yin, W. (2013), SIAM Journal on imaging science*
>
> **3\. Quantitative evaluation on real datasets (W3)**
>
> A challenge shared with any clustering or dimensionality reduction method is that real data lack ground-truth components, which makes their evaluation inherently difficult. For that, in our paper, we carefully chose datasets (voting and Wikipedia) whose components can be validated through the semantic meaning of their groupings. In addition to this qualitative evaluation, we also did evaluate the results quantitatively in the paper, including:
>
> - **Information criteria (AIC, BIC, HQC, log-likelihood):** We compared MILCCI to all baselines via information criteria that balance reconstruction quality and model complexity (Fig. 10, Fig. 25).
> - **Component distance analysis:** Fig. 30 quantitatively compares within-component vs. cross-component distances across three real-world datasets.
> - **Per-trial reconstruction error:** For the auditory cortex experiment (App. J, Fig. 37E).
>
> Regarding the reviewer's comment about the permutation test: We wish to clarify that we tested three distinct null hypotheses (Fig. 9D): shuffling state assignments between rows, replacing data with random noise, and shuffling states within each component dimension, all yielding p \< 0.001 across 1000 iterations. We also performed per-component permutation tests (Fig. 9E) and leave-one-out analysis with Shapley values (Fig. 9A-C). Together, these test whether the specific structure of the discovered components is meaningful, not merely whether they are non-random.
> We will also add cross-validated decoding on the IBL and Wikipedia datasets in the revised manuscript.
>
> **4\. Discussion of application boundaries. (refers to text under `limitations’)**
> We appreciate these points and we agree that the single-category constraint, linearity assumption, and optimization properties are important to discuss explicitly. For the single-category design choice, please refer to our response #2 to Qc4r. The linearity assumption is discussed in the paper in our Discussion section (Section 5), where we mention kernelization as a future extension. We also demonstrate a preliminary simple nonlinear extension via tanh transformation in App. J.

---

> > ### Author Rebuttal · Reviewer_UDpu · 2026-04-01
> >
> > I appreciate the author's response and for pointing out my missed points. However, two concerns remain unresolved:
> >
> > 1. The absence of PARAFAC2 as a baseline leaves a critical question unanswered, namely whether the added complexity of MILCCI yields meaningful gains over the strongest existing method for variable-length tensor data. Furthermore, arguing that PARAFAC2 is less capable than MILCCI conflates the role of a competitor with that of a baseline, as the two serve distinct purposes in empirical evaluation.
> >
> > 2. Citing general convergence theory does not directly guarantee convergence for MILCCI, as different assumptions can substantially affect convergence behavior.
> >
> > Thus, I maintain my current score.

---

> > > ### Author Response · Authors · 2026-04-04
> > >
> > > Dear reviewer, thank you so much for the follow-up. Regarding your first concern, we have now run additional experiments with PARAFAC2 comparison, whose figures you can find in the anonymous link __https://docs.google.com/document/d/e/2PACX-1vSRbt2cGmkWBd5cBM8eISLqG0-mxsgdikZiGVJdWg0jlE5h-vkuU5eOEySK148PoCZ1NVKTP5sV7Wp8/pub__. We compared PARAFAC2 as a baseline across all main experiments and added it to the information criteria calculations (AIC, HQC, log-likelihood). We also conducted cross-validated decoding analyses on the Wikipedia and IBL datasets and compared to all baselines including sliceTCA and PARAFAC2: for each method we extracted temporal features from the per-trial traces and evaluated category decoding accuracy via cross-validated classification (for categorical variables) and Ridge regression (for continuous variables such as trial number). We evaluated these based on accuracy as well as balanced accuracy for categorical variables, and via correlation with labels as well as MAE for continuous ones. We further computed per-trial reconstruction quality ($R^2$). MILCCI achieves the best information criteria values and the highest decoding accuracy across task variables. Regarding empirical convergence statistics, these require hundreds of reruns across initializations and significant compute; hence we commit to including them in the revised manuscript.

---

### Official Review · Reviewer_Qc4r · 2026-03-04

**Soundness:** 2
**Presentation:** 2
**Significance:** 3
**Originality:** 3
**Overall Recommendation:** 3
**Confidence:** 3

**Summary:**

This paper introduces MILCCI, a dimensionality reduction and component identification model for multi-trial, multi-label time-series data. MILCCI allows components to subtly adjust their sparse compositions based on trial labels spanning multiple categories, instead of enforcing fixed component structures across trials. The model balances the data fidelity, sparse component composition, structural consistency, and temporal smoothness of trial-varying traces, while remaining flexible across trials. The authors evaluate MILCCI on synthetic data and real-world datasets, demonstrating its capacity to recover interpretable, label-dependent structural adaptations and trial-specific temporal dynamics.

**Compliance With Llm Reviewing Policy:**

Affirmed.

**Key Questions For Authors:**

1. Can the authors provide a stronger quantitative evaluation on the real datasets, beyond illustrative case studies?
2. How sensitive are the main real-data findings to initialization, per-category component counts $p(k)$, and the trace regularization hyperparameters $\gamma_3$ and $\gamma_4$? The appendix robustness analysis is helpful, but it appears limited mainly to $\gamma_1$ and $\gamma_2$ in the additional experiment.
3. What is the intended behavior when important signal is driven by interactions between categories rather than by category-specific components?

**Limitations:**

yes

**Strengths And Weaknesses:**

### Strengths:

S1. This paper tackles the modeling problem of disentangling how multiple label categories are represented in repeated, high-dimensional temporal observations. The method is interesting and reasonably original. The use of category tensors with label-dependent component variants combined with explicit label-similarity graphs is a meaningful extension over the current method that holds fixed-component decompositions and only handle a single label dimension. The method also supports categorical and ordinal labels, varying trial lengths, non-negativity constraint on the components, which make the method flexible.

S2.  The objective is clearly specified and the alternating fitting procedure is coherent. The use of synthetic data validates the model's ability to recover correct components. The real-world applications are carefully analyzed, particularly the post-hoc validation utilizing leave-one-out analysis, Shapley values, and permutation tests in the voting experiment. The appendix also includes a hyperparameter sensitivity analysis, which is helpful.

S3. The paper is well-structured and written clearly. Figure 1 effectively illustrates the core tensor selection mechanism. The discussion section is also reasonably candid about some modeling limitations. The inclusion of extensive appendices detailing baselines, hyperparameter sensitivity, and alternative dynamic priors strengthens the paper.

### Weaknesses:

W1. Outside the synthetic experiment, the evidence is dominated by qualitative narratives and post hoc interpretation. The voting case includes a permutation-based validation, which is useful, but the Wikipedia and neural sections rely heavily on interpretive examples rather than task-oriented quantitative evaluation, held-out prediction, or externally defined benchmarks.

W2. The method makes a fairly strong structural assumption that each component is assigned to a single category. It is elegant and aids interpretability, but it may be restrictive when important structure is genuinely interaction-specific across categories rather than decomposable into independent category pieces. The paper does not discuss this case.

W3. The optimization is non-convex, initialized by dictionary learning, and the algorithm ends simply with "until converged", without a concrete stopping state. The paper does not provide much discussion of stability across random initializations , nor a careful analysis of sensitivity to all regularization terms and to the per-category component counts on the main experiments. The appendix gives recommendations for $\gamma_1$ and $\gamma_2$ and a sensitivity analysis for those parameters, but this still leaves some uncertainty about stability in practice.

---

> ### Author Rebuttal · Authors · 2026-03-31
>
> We thank the reviewer for their thorough review and for recognizing that our "method is interesting and reasonably original" and "is a meaningful extension over the current method", as well as that the paper is "well-structured and written clearly", with a "clearly specified" objective and "coherent" fitting procedure. We also appreciate the reviewer's acknowledgment that the real-world applications are "carefully analyzed" and that the appendices strengthen the paper. We address each concern below.
>
> 1) **Quantitative evaluation of the neuroscience and Wikipedia experiment (W1, Q1)**
>
> We appreciate this feedback. A challenge shared with any clustering or dimensionality reduction method is that real-world datasets lack ground-truth components, which makes their evaluation inherently more difficult. We thus carefully chose datasets (voting and Wikipedia) whose components can be validated through the semantic meaning of their groupings, providing a first step towards understanding.
>
> In addition to this qualitative evaluation, we have also provided quantitative evaluations in the paper across experiments, including:
>
> * Information criteria (AIC, BIC, HQC, log-likelihood): We compared MILCCI to baselines via information criteria that balance reconstruction quality and model complexity, for the voting experiment (Fig. 10) and the Wikipedia experiment (Fig. 25).
> * *Component distance analysis:* Fig. 30 quantitatively compares within-component vs. cross-component distances across three real-world datasets for both MILCCI and sliceTCA, demonstrating that MILCCI maintains component identity across conditions.
>
> Moreover, we included in the supplement further validation on the auditory cortex experiment (App. J), including per-trial reconstruction error distributions (Fig. 37E) and a hyperparameter sensitivity analysis across 800 parameter combinations (Fig. 38).
>
> While we believe that together with the synthetic validation and the permutation-based analysis in the voting experiment, these provide substantial quantitative evidence, we recognize the advantage of additional quantifications and will add cross-validated decoding analyses on the IBL and Wikipedia datasets in the revised manuscript.
>
> 2) **Additive design choice and category interaction extension (W2, Q3)**
>
> We thank the reviewer for raising this great point and for recognizing this design is elegant and aids interpretability. We agree that this is a promising direction. The additive structure is a deliberate design choice for interpretability: when each component belongs to a single category, one can distinguish between category-specific effects and identify what each component represents. Allowing interactions would mean components capture joint effects of multiple categories, which undermines interpretability by preventing the isolation of individual category contributions. This choice is shared by other established methods in the field, including dPCA (Kobak et al., 2016\) and TDR (Mante et al., 2013; Aoi & Pillow, 2018), which similarly decompose variance along individual task variables.
>
> That said, introducing interaction terms is an exciting future direction which we believe is well suited based on MILCCI's modular design, for example by introducing a multi-way graph that captures relationships across categories and enables components to vary along multiple category dimensions simultaneously. Yet, this requires careful consideration as it would increase model complexity and reduce interpretability, and hence we believe it should be beyond the current scope of the paper. Another simpler and more direct approach is that if there is a specific interest in how categories interact, one can define two categories as a single merged category (e.g., combining task difficulty and decision outcome into one set of components), which the model already supports.
>
> *Kobak et al. (2016), eLife; Mante et al. (2013), Nature; Aoi & Pillow (2018), NeurIPS*
>
> 3) **Model sensitivity and stability across random initializations. (W3, Q2)**
>
> We believe this is a valid point that can further test the method's robustness, and we thank the reviewer for also acknowledging the sensitivity analysis for parameters in the appendix (Fig. 38, 800 combinations of $\gamma_1$ and $\gamma_2$). We will add an analysis of sensitivity to random initialization on that experiment, as well as to per-category component counts in the revised manuscript.

---

> > ### Author Rebuttal · Reviewer_Qc4r · 2026-04-01
> >
> > My Q1 and Q2 have not been addressed. Therefore, I will maintain my score.

---

> > > ### Author Response · Authors · 2026-04-04
> > >
> > > Dear reviewer, thank you so much for the follow-up. We have now conducted additional quantitative analyses on the Wiki and Neuroscience real-world experiments, and figures are available at the anonymous link: [https://docs.google.com/document/d/e/2PACX-1vSRbt2cGmkWBd5cBM8eISLqG0-mxsgdikZiGVJdWg0jlE5h-vkuU5eOEySK148PoCZ1NVKTP5sV7Wp8/pub](https://docs.google.com/document/d/e/2PACX-1vSRbt2cGmkWBd5cBM8eISLqG0-mxsgdikZiGVJdWg0jlE5h-vkuU5eOEySK148PoCZ1NVKTP5sV7Wp8/pub).
> > >
> > > Particularly, regarding **Q1**, we ran cross-validated decoding analyses on both the Wikipedia and Neuroscience (IBL) datasets: for each method (MILCCI, SVD, PARAFAC, SiBBlInGS, sliceTCA, etc.), we extracted temporal features from the per-trial traces and evaluated each category's decoding via cross-validated classification (for categorical variables) and Ridge regression (for continuous variables). We evaluated these based on accuracy as well as balanced accuracy for categorical variables, and via correlation with labels as well as MAE for continuous ones. We also computed per-trial reconstruction quality ($R^2$). Additionally, we added another new baseline (PARAFAC2) across all these new analyses. MILCCI achieves the highest decoding accuracy and reconstruction across categories and datasets.
> > >
> > > Regarding **Q2**, that is a great point. We have begun these additional sensitivity analyses, but given the computational intensity of performing hundreds of reruns, we commit to including this in the revised manuscript.
> > >
> > > ## --- Update April 6 (AoE) ---
> > > Following the reviewer's suggestion, we now also extended our sensitivity analysis to include initialization sensitivity, hyperparameter sensitivity ($\\gamma_3$, $\\gamma_4$ ), and a sweep over the number of components P. We ran 100 random initializations and found that cross-seed variation is significantly smaller than within-seed variation, confirming robustness to initialization. We also swept $\gamma_3$ and $\gamma_4$ across five orders of magnitude and found the learned components remain stable. Finally, varying P from 2 to 30 shows stable ensemble structure with increasing subspace overlap. These results are added as Figures 6-8 at the end of the same anonymous figure document: [https://docs.google.com/document/d/e/2PACX-1vSRbt2cGmkWBd5cBM8eISLqG0-mxsgdikZiGVJdWg0jlE5h-vkuU5eOEySK148PoCZ1NVKTP5sV7Wp8/pub](https://docs.google.com/document/d/e/2PACX-1vSRbt2cGmkWBd5cBM8eISLqG0-mxsgdikZiGVJdWg0jlE5h-vkuU5eOEySK148PoCZ1NVKTP5sV7Wp8/pub)

---

### Official Review · Reviewer_yaXh · 2026-03-12

**Soundness:** 3
**Presentation:** 1
**Significance:** 3
**Originality:** 3
**Overall Recommendation:** 4
**Confidence:** 4

**Summary:**

The paper proposes a framework for jointly factorizing multi-trial time series data with categorical labels. Each trial corresponds to a matrix-valued observation whose temporal length may vary, and trials are associated with multiple categorical labels (e.g., task difficulty or behavioral outcome). The proposed model decomposes each trial into latent temporal factors and loading matrices that depend on the categorical labels. The authors introduce an additive structure to combine label-specific loading matrices so that information can be shared across trials while capturing label-dependent variability. An estimation procedure is developed and evaluated through simulations and several real data analyses. The motivation of integrating information across labeled trials is interesting, as separate factorization of each trial would ignore cross-trial relationships. However, the presentation of the model and its assumptions is difficult to follow due to heavy notation and limited intuitive explanation. Several modeling choices, including the additive structure and similarity constraints across label-specific loading matrices, are not sufficiently justified, and important issues such as parameter identifiability and interpretability remain unclear.

**Compliance With Llm Reviewing Policy:**

Affirmed.

**Final Justification:**

The authors have addressed most of the concerns raised in the initial review and provided adequate clarifications. The proposed framework is reasonably novel, and the estimation procedure follows naturally from the problem formulation.

One limitation of the manuscript is the absence of a discussion on the convergence properties of the proposed estimation method. This is an important aspect that would strengthen the work if addressed.

While the rebuttal has led me to revise my evaluation from reject to weak accept, I remain somewhat concerned about the suitability of this work for a venue such as ICML. In my view, the paper would benefit from a longer format, which would allow the authors to more fully develop their ideas, introduce key concepts more clearly, and provide additional technical details.

Given the current page constraints, several parts of the manuscript feel compressed, resulting in a dense presentation that can be difficult to follow at times. A more expansive format would likely improve both clarity and accessibility.

**Key Questions For Authors:**

1. Could the authors provide a simple illustrative example (e.g., two categories with two levels each) to explain how the loading matrices are constructed and how different label effects combine in the model?

2. The formulation encourages loading matrices associated with different levels of a category to be similar. What is the practical justification for this assumption, and how should it be validated in real applications?

3. The proposed additive model does not capture interaction effects between categories (e.g., the joint effect of “hard” and “correct”). How would the framework handle situations where such interactions are important?

4. The factorization appears to have the usual rotational ambiguity. How do the authors address this identifiability issue, and how should the estimated loading matrices be interpreted?

5. Could the authors clarify whether the method provides practical advantages when trial durations vary?

**Limitations:**

Yes.

**Strengths And Weaknesses:**

**Strengths**

1. *Well-motivated problem* - The paper addresses an interesting and relevant problem: analyzing multi-trial time series data while incorporating categorical trial labels. Modeling shared structure across trials while allowing label-dependent variation is an important challenge in many applications.

2. *Attempts to model cross-trial relationships* - The paper emphasizes that analyzing each trial separately ignores shared latent structure across trials. The proposed framework attempts to leverage information across trials through shared latent factors and label-dependent loadings.

3. *Use of multiple real datasets* - The authors demonstrate the method on a range of real datasets, which suggests potential applicability of the framework.


**Weaknesses**

1. *Lack of clarity in the problem formulation* - The core modeling idea is difficult to understand from the main text. The presentation relies heavily on notation and mathematical expressions, which obscures the key intuition behind the method. A simple motivating example (e.g., two categories with two levels each) would greatly help explain the model structure before introducing the fully general case.

2. *Excessive and sometimes inconsistent notation* - The manuscript uses a large number of symbols and indices, which significantly disrupts the flow of reading. In addition, there appear to be inconsistencies in notation (for example, inconsistent formatting of matrices, such as in line 140 and 143, or unclear indexing in some expressions). Some quantities are introduced without clear definition (e.g., $\|\|_{1,1}$), and normalization steps of the estimates are not clearly specified.

3. *Modeling assumptions are not well justified* - The model includes assumptions that enforce similarity between loading matrices corresponding to different levels of a category. For example, the formulation encourages loadings for different difficulty levels (e.g., “hard” and “easy” for a given 'choice') to be close to each other.

However, it is unclear:

a. why such similarity should necessarily hold in real applications, if holds - how this assumption can be verified in practice,

b. whether this constraint might prevent the model from capturing meaningful differences across conditions.

The manuscript would benefit from a clearer explanation of the rationale behind these assumptions.

4. *Additive structure ignores interaction effects between categories* - The model combines label effects using an additive structure. However, in many applications there may be interaction effects between categories. For example, the neural response when a task is hard and correct may differ from the sum of the individual effects of “hard” and “correct”. The current additive formulation does not capture such interactions, which limits the expressiveness of the model.

5. *Identifiability and interpretability are not discussed* - From the model formulation, the parameters do not appear to be uniquely identifiable. In particular, factorization models generally admit rotational ambiguity. For example, $AH$ and $H^\top \Phi$ produce the same reconstruction for any orthonormal matrix $H$.

The manuscript does not clearly address how this identifiability issue is handled or how the resulting loading matrices should be interpreted. Without clarifying this point, it is difficult to understand how the estimated parameters correspond to meaningful scientific quantities.

6. *Inconsistency between modeling assumptions and experiments* - The model formulation allows for trials of varying temporal lengths. However, both the simulations and the real-data experiments appear to use trials with identical lengths. This makes it unclear whether the proposed framework actually benefits from the ability to handle variable-length trials.

7. *Limited intuitive explanation of cross-trial modeling* - The motivation emphasizes the importance of capturing cross-trial relationships rather than factorizing each trial independently. However, the manuscript does not clearly illustrate what kinds of cross-trial relationships the model is intended to capture or how the proposed formulation achieves this.

---

> ### Author Rebuttal · Authors · 2026-03-31
>
> We thank the reviewer for recognizing the relevant problem our method addresses, its originality, the utility of cross-trial relationships, and the scope of the real-world demonstrations. We address the concerns below.
>
> 1. *Clarity in the problem formulation and trial relationships (W1,W7,Q1)*
>
> We will clarify the example from the Intro. and Fig. 1:
>
> Consider a decision-making experiment where each trial is described with respect to two categories: (a) task difficulty (easy/hard) and (b) choice (correct/incorrect). A trial may be associated with any combination of these, e.g., "(hard, correct)" or "(easy, correct)". Each trial produces a time series of neural recordings. MILCCI addresses this by organizing components along separate axes of change, here, one axis for task difficulty and one for choice. For a trial labeled (hard, correct), MILCCI selects the 'hard' variant from the difficulty tensor and the 'correct' variant from the choice tensor, concatenates them to form that trial's loading matrix, and learns trial-specific temporal traces (Fig. 1). Trials that are (hard, correct) share the difficulty components with (hard, incorrect) trials, while another subset of components varies to capture the correctness change. In contrast, (hard, correct) trials do not share identical components with (easy, incorrect) trials, since both difficulty and correctness vary. Each component variant is thus inferred using all trials observed under that category option, effectively pooling data across trials rather than per trial (Sec. 3).
>
> 2. *High number of notations (W2)*
>
> We acknowledge that there are a lot of symbols, yet believe these are necessary to describe the problem. To make notation easier to follow, the paper includes Tab. 1 and Fig. 1. We will remove the bolding at line 143; and will clarify that ||*||_1,1 is L1 on flattened matrices
>
> 3. *Component similarity in real applications (W3a, Q2)*
>
> The cross-label similarity assumption holds in our experiments: Fig. 30 shows that same-component variants remain structurally similar across conditions. The regularization reflects this view and serves 3 purposes:
>
> a) *Tracking changes in representation.* As motivated in lines 154-157 & Fig. 1, in many real systems the same underlying structures persist across conditions but adapt slightly. The regularization enables tracking these structures as they adjust, rather than treating each trial independently.
> b) *Identifying corresponding components.* Without the regularization, component j in one condition does not represent the same structure as j in another (due to permutation invariance of matrix factorizations), which is critical for interpretation.
> c) *Statistical strength.* Tying together representations across conditions leverages shared information across trials that increase the effective sample size per component and produce more robust estimates.
>
> *4. Consistency constraint and capturing meaningful differences (W3b)*
>
> MILCCI addresses W3b via:
> 1. *Soft regularization.* The cross-condition similarity is imposed via a regularization penalty (Eq. 2), rather than a hard constraint on the maximum difference. If a meaningful structural change is present, the data fidelity term will balance out the regularization, allowing the component to adjust as needed. Fig. 30 demonstrates this, showing that same-component variants remain similar while capturing differences.
> 2. *Multiple components per category.* If two variants of a component diverge substantially, they may represent genuinely different structures, which MILCCI appropriately captures via separate components (distinct columns of A^(k)).
> 3. *Components that vary freely across trials.* We described (App. A.2) an option for unregularized components to capture components that can represent completely different structures across trials.
>
> *5. Additive structure (W4, Q3):* See response \#2 to Qc4r
>
> *6. Potential rotational ambiguity (W5, Q4)*
>
> Rotational ambiguity (A, Phi) \-\> (AQ, Q^{-1}Phi) applies to unconstrained matrix factorizations. MILCCI mitigates this via sparse (L1) regularization on the components (Eq. 2\) that breaks rotational symmetry. It is well-established in the literature \[1\] that sparsity is rotation-invariant up to permutation and scaling: rotating a sparse vector R\*x by a rotation matrix R increases the number of non-zeros, unless R rotates by multiples of $\\pi/2$ (i.e., permutation), which does not change the decomposition meaning.
>
> [1] Ng, A. Y. (2004). ICML
>
> *7. Variable-length trials (W6, Q5)*
>
> We would like to clarify that in our IBL experiment, trial durations do vary (T=137 is the maximum duration used). The ability to support variable-length trials addresses a practical data structure limitation, not a representational one: it concerns what data the model can accept as input. Tensor factorization methods require equal-length trials and cannot accept variable-length data without zero-padding, while MILCCI accepts them natively.

---

> > ### Author Rebuttal · Reviewer_yaXh · 2026-04-02
> >
> > I thank the authors for their detailed response and for addressing many of the concerns raised in the initial review. While most points have been clarified satisfactorily, a few key issues remain.
> >
> > Regarding the assumption on cross-label similarity, the authors state (lines 177–179) that *while the variants of each component need not be identical, we assume they are structurally similar, with their similarity level proportional to the similarity of their label values.* This is formalized through the condition
> >
> > $\|A^{(k)}_{::i}- A^{(k)}_{::i^\prime}\|^2_F\leq\delta.$
> >
> > However, this assumption also includes the degenerate case where the parameters are identical, i.e., the distance is zero. In such a scenario, the model effectively reduces to a simpler form, and the proposed estimation procedure would ideally need to be adapted to account for this equality case. Otherwise, one may be using an unnecessarily complex and potentially over-parameterized procedure to fit a simpler model.
> > To explicitly exclude this case, a more precise formulation would be
> >
> > $0<\|A^{(k)}_{::i} - A^{(k)}_{::i^\prime}\|^2_F\leq \delta$
> >
> > Could the authors clarify whether this strict inequality is implicitly assumed, or whether the inclusion of the equality case is intentional?
> >
> > I acknowledge that this distinction may not materially affect the current estimation procedure; however, it is important for foundational assumptions to be stated as precisely as possible for the purpose of clarity.
> >
> > I have adjusted my score accordingly. That said, I strongly believe that the work would benefit from a longer paper format, where the authors would have more space to fully develop their ideas, introduce key concepts, and provide additional discussion. Given the page limits at ICML, several technical details appear compressed, which makes the presentation dense at times and somewhat difficult to follow. A more expansive format would likely improve both clarity and accessibility.

---

> > > ### Author Response · Authors · 2026-04-04
> > >
> > > Dear reviewer, thanks a lot for your follow-up, and we also believe that the additional point you raised is a thoughtful one. As you correctly mentioned, in cases where no component change is needed, this formulation reduces to the simpler case where some components can stay the same across conditions. This is, in fact, a desirable property of the model: it allows the simpler solution to emerge naturally when it is sufficient, following an Occam's razor principle. If a fixed component model can describe the data well enough with a small number of components, that solution will be preferred as any changes trade with the cost ($\|A^{(k)}_{::i}- A^{(k)}_{::i^\prime}\|^2$ will increase under changes). At the same time, we do not want to force equality either, as in many systems (e.g., neuronal ensembles that adjust over trials, or the examples in the paper), small changes across conditions are *needed* to capture the system evolution/adjustments effectively. Since we do not know a priori which case applies, the model needs to allow for both, and learn the appropriate solution from the data. Thank you for bringing it up, and we will make sure to clarify this aspect in the paper.

---

### Decision · Program_Chairs · 2026-04-30

**Decision:**

Accept (regular)

**Comment:**

I recommend to accept the paper.

It discusses a novel tensor decomposition method for multi trial large scale time series data. The approach is evaluated on a range of different real-world datasets across different disciplines, as well as verified on a synthetic benchmark.

The reviewers found the multi-trial ts setting important and noted the extensive experiments, although a lack of more baselines was noted. The authors provided additional comparisons, but also pointed to some baseline comparisons in the paper. I agree with the reviewers that the way the paper is written is more in a "journal format", and it would help a ML audience if these baselines comparisons were more compactly positioned in the paper.

Another repeated theme were statements on the convergence of the algorithm, to which the authors also during rebuttal (in my impression) did not provide a satisfying answer -- for the camera ready, this point could simply be highlighted as a limitation and note to practicioners applying the algorithm.